# Improved Sample Complexity for Multiclass PAC Learning

**Steve Hanneke**
Purdue University
`steve.hanneke@gmail.com`

**Shay Moran**
Technion and Google Research
`smoran@technion.ac.il`

**Qian Zhang**
Purdue University
`zhan3761@purdue.edu`

## Abstract

We aim to understand the optimal PAC sample complexity in multiclass learning. While finiteness of the Daniely-Shalev-Shwartz (DS) dimension has been shown to characterize the PAC learnability of a concept class [Brukhim, Carmon, Dinur, Moran, and Yehudayoff, 2022], there exist polylog factor gaps in the leading term of the sample complexity. In this paper, we reduce the gap in terms of the dependence on the error parameter to a single log factor and also propose two possible routes towards completely resolving the optimal sample complexity, each based on a key open question we formulate: one concerning list learning with bounded list size, the other concerning a new type of shifting for multiclass concept classes. We prove that a positive answer to either of the two questions would completely resolve the optimal sample complexity up to log factors of the DS dimension.

## 1 Introduction

Multiclass learning refers to the problem of classifying an input feature from a set (feature space) $\mathcal{X}$ to a label in a set (label space) $\mathcal{Y}$ with $|\mathcal{Y}| > 2$ ($\mathcal{Y}$ can be infinite) [Natarajan, 1989, Ben-David et al., 1995, Daniely and Shalev-Shwartz, 2014, Brukhim et al., 2022]. When $|\mathcal{Y}| = 2$, the problem is known as binary classification. Multiclass learning has wide applications to various tasks in machine learning including image classification [Rawat and Wang, 2017], natural language processing [Kowsari et al., 2019], tissue classification [Li et al., 2004], etc. For theoretical analysis of multiclass learning, a probabilistic setting is typically assumed, where all the feature-label pairs in the training sequence are assumed to be independent and identically distributed (iid) samples from some distribution $P$ over $\mathcal{X} \times \mathcal{Y}$. Then, the objective of the learner is to minimize the error rate of the output classifier under the distribution $P$. A basic framework in the probabilistic setting is Probably Approximately Correct (PAC) learning [Valiant, 1984]. Though the characterization of PAC learnability of a binary concept class with the finiteness of its Vapnik-Chervonenkis (VC) dimension has been proved by Vapnik and Chervonenkis [1968], the characterization of the multiclass PAC learnability remained open until Brukhim et al. [2022] showed the equivalence between the PAC learnability of a concept class and the finiteness of its Daniely-Shalev-Shwartz (DS) dimension (dim, see Definition 1.4) [Daniely and Shalev-Shwartz, 2014] instead of Natarajan dimension or graph dimension.

However, the problem of establishing the optimal sample complexity or error rate (see Section 1.1 for formal definitions) for multiclass learning remains unsolved. For binary concept classes, Hanneke [2016] showed that the sample complexity is in $\Theta((d + \log(1/\delta))/\varepsilon)$ where $d$ is the VC dimension of the concept class. Since DS dimension and VC dimension coincide for binary concept classes, it is natural to ask if the sample complexity of a concept class $\mathcal{H} \subseteq \mathcal{Y}^{\mathcal{X}}$ for $|\mathcal{Y}| > 2$ is also in

38th Conference on Neural Information Processing Systems (NeurIPS 2024).

$\Theta((d + \log(1/\delta))/\varepsilon)$ where $d = \dim(\mathcal{H})$ is the DS dimension of $\mathcal{H}$. In terms of the upper bound, it asks if there exists a multiclass learner whose worst case error rate is in $O((d + \log(1/\delta))/n)$ with probability at least $1 - \delta$, where $n$ denotes the size of the training sequence. However, on the one hand, an explicit proof of the $\Omega((d + \log(1/\delta))/\varepsilon)$ lower bound on the sample complexity is still missing in the literature. On the other hand, the current best upper bound on worst case error rate to our knowledge is $O\big(\frac{(d^{3/2} \log(d) + d \log(\log(n))) \log^2(n) + \log(1/\delta)}{n}\big)$ [Brukhim et al., 2022], which differs from the conjectured rate by the factor $(\log(\log(n)) + \sqrt{d} \log(d)) \log^2(n)$.

In this paper, we step forward towards improved sample complexity and error rate in multiclass learning. As the concept class is fixed and the sample size increases during an online learning process, we mainly focus on improving the error rate in terms of the sample size $n$. Specifically, for a concept class $\mathcal{H} \subseteq \mathcal{Y}^{\mathcal{X}}$ with $\dim(\mathcal{H}) = d$, we prove an $\Omega((d + \log(1/\delta))/\varepsilon)$ lower bound on its sample complexity and construct a multiclass learner whose worst case error rate is $O((d^{3/2} \log(d) \log(n) + \log(1/\delta))/n)$ with probability at least $1 - \delta$, which implies that the sample complexity of $\mathcal{H}$ is $O\big(\frac{d^{3/2} \log(d) \log(d/\varepsilon) + \log(1/\delta)}{\varepsilon}\big)$. Our results greatly narrow the gap between the upper and lower bounds of the sample complexity and the error rate. The dependence of the upper bound of the error rate on the sample size has also been improved from $O(\log(\log(n)) \log^2(n)/n)$ to $O(\log(n)/n)$ (treating $d$ as a constant). The multiclass learner we construct builds upon a list learner which predicts a list of labels for the test point (see Section 2.1 for a detailed introduction to list learning). Actually, we prove a reduction from multiclass learning to list learning and upper bound the error rate of the constructed multiclass learner with some function of the list size and the expected error rate of the list learner (the probability of excluding the true label in the predicted list). Moreover, the upper bound indicates that a list learner with size independent of $n$ and expected error rate scaling linearly in $1/n$ in terms of the sample size $n$ would imply an $O(1/n)$ error rate (treating $d$ as a constant). We leave the construction of such list learners an open question.

Furthermore, we also explore an alternative combinatorial approach towards improved sample complexity in multiclass learning. For a concept class, we can define a hypergraph called the one-inclusion graph [Haussler et al., 1994] on its projection to a finite sequence of features (see Section 1.1 for definitions). Then, informally speaking, the "density" (defined through the average degree of the one-inclusion graph) of a concept class can be used to upper bound the error rate of multiclass learning [Daniely and Shalev-Shwartz, 2014, Aden-Ali et al., 2023]. Specifically, if we can upper bound the density of any concept class $\mathcal{H}$ by a multiple of its DS dimension, then the sample complexity is in $O((\dim(\mathcal{H}) + \log(1/\delta))/\varepsilon)$, which matches the lower bound we prove. Thus, a proof of the above upper bound directly leads to a $\Theta((\dim(\mathcal{H}) + \log(1/\delta))/\varepsilon)$ sample complexity for multiclass learning. When $\dim(\mathcal{H}) = 1$, we successfully prove the $\Theta(\log(1/\delta)/\varepsilon)$ sample complexity in Theorem 3.2. For general concept classes, we develop a technique named "pivot shifting" similar to the shifting operator [Haussler, 1995] on concept classes. We show that if a pivot shifting does not increase the DS dimension of a concept class, then its density is upper bounded by twice the DS dimension. We leave the impact of pivot shifting on DS dimension as another open question.

Throughout the paper, we use $\mathbb{N}$ to denote the set of positive integers. For any $n \in \mathbb{N}$, we define $[n] := \{1, \ldots, n\}$. For any sets $\mathcal{X}, \mathcal{Y}$, sequence $\mathbf{x} = (x_1, \ldots, x_n) \in \mathcal{X}^n$, and function $f \in \mathcal{Y}^{\mathcal{X}}$, we define the subsequence $\mathbf{x}_{-i} := (x_1, \ldots, x_{i-1}, x_{i+1}, \ldots, x_n)$ for $i \in [n]$ and $f|_{\mathbf{x}} := (f(x_1), \ldots, f(x_n))$. The projection of a set $F \subseteq \mathcal{Y}^{\mathcal{X}}$ to $\mathbf{x}$ is defined as $F|_{\mathbf{x}} := \{f|_{\mathbf{x}} : f \in F\} \subseteq \mathcal{Y}^n$.

**Outline** In Section 1.1, we introduce the problem of multiclass learning and review some existing results. In Section 1.2, we summarize the key points of our theoretical results. In Section 2, we introduce list learning, present the reduction from multiclass learning to list learning, and improve the sample complexity upper bound of multiclass learning via this reduction together with a boosting technique for list learners. In Section 3, we prove the optimal sample complexity for classes of DS dimension 1, introduce the intuition and the definition of "pivot shifting", and demonstrate its potential application to the proof of the optimal sample complexity of multiclass learning.

## 1.1 Multiclass learning

In this section, we formally introduce the problem of multiclass learning [Valiant, 1984]. For any distribution $P$ over $\mathcal{X} \times \mathcal{Y}$, the error rate of a classifier $h \in \mathcal{Y}^{\mathcal{X}}$ under $P$ is defined as

$$\mathrm{er}_P(h) := P(\{(x, y) \in \mathcal{X} \times \mathcal{Y} : y \neq h(x)\}).$$

In this paper, we focus on **realizable** distributions: for a concept class $\mathcal{H} \subseteq \mathcal{Y}^{\mathcal{X}}$, a distribution $P$ over $\mathcal{X} \times \mathcal{Y}$ is $\mathcal{H}$-realizable if $\inf_{h \in \mathcal{H}} \mathrm{er}_P(h) = 0$. Let $\mathrm{RE}(\mathcal{H})$ denote the set of $\mathcal{H}$-realizable distributions. Besides, $((x_i, y_i))_{i=1}^n \in (\mathcal{X} \times \mathcal{Y})^n$ is $\mathcal{H}$-realizable if $\exists h \in \mathcal{H}$ such that $y_i = h(x_i), \forall i \in [n]$.

**Definition 1.1** (Multiclass learner). *A **multiclass learner** (or a learner) $\mathcal{A}$ is an algorithm which given a sequence $\mathbf{s} \in \cup_{n=0}^{\infty} (\mathcal{X} \times \mathcal{Y})^n$ and a concept class $\mathcal{H} \subseteq \mathcal{Y}^{\mathcal{X}}$, outputs a classifier $\mathcal{A}(\mathbf{s}, \mathcal{H}) \in \mathcal{Y}^{\mathcal{X}}$.*

Then, we can define multiclass PAC learning as follows.

**Definition 1.2** (Multiclass PAC learning). *For any concept class $\mathcal{H} \subseteq \mathcal{Y}^{\mathcal{X}}$, the **(PAC) sample complexity** $\mathcal{M}_{\mathcal{A},\mathcal{H}} : (0,1)^2 \to \mathbb{N}$ of a multiclass learner $\mathcal{A}$ is a mapping from $(\varepsilon, \delta) \in (0,1)^2$ to the smallest positive integer such that for any $m \geq \mathcal{M}_{\mathcal{A},\mathcal{H}}(\varepsilon, \delta)$ and any distribution $P \in \mathrm{RE}(\mathcal{H})$, $\mathbb{P}_{S \sim P^n}(\mathrm{er}_P(\mathcal{A}(S, \mathcal{H})) > \varepsilon) \leq \delta$, and we define $\mathcal{M}_{\mathcal{A},\mathcal{H}}(\varepsilon, \delta) = \infty$ if no such integer exists. We say $\mathcal{H}$ is **PAC learnable** by $\mathcal{A}$ if $\mathcal{M}_{\mathcal{A},\mathcal{H}}(\varepsilon, \delta) < \infty$ for all $(\varepsilon, \delta) \in (0,1)^2$. The **(PAC) sample complexity** of $\mathcal{H}$ is defined as $\mathcal{M}_{\mathcal{H}}(\varepsilon, \delta) := \inf_{\mathcal{A}} \mathcal{M}_{\mathcal{A},\mathcal{H}}(\varepsilon, \delta)$ for any $(\varepsilon, \delta) \in (0,1)^2$.*

Sometimes it is easier to analyze the expected error rate

$$\varepsilon_{\mathcal{A},\mathcal{H},P} : \mathbb{N} \to [0,1], \quad n \mapsto \mathbb{E}_{S \sim P^n}[\mathrm{er}_P(\mathcal{A}(S, \mathcal{H}))] = \mathbb{P}_{(S,(X,Y)) \sim P^{n+1}}(Y \neq \mathcal{A}(S, \mathcal{H})(X))$$

for a learner $\mathcal{A}$ and distribution $P$ over $\mathcal{X} \times \mathcal{Y}$, or transductive error rate

$$\varepsilon_{\mathcal{A},\mathcal{H},\mathrm{trans}} : \mathbb{N} \to [0,1], \quad n \mapsto \sup_{\mathbf{s} = ((x_1, h(x_1)), \dots, (x_n, h(x_n))) \in (\mathcal{X} \times \mathcal{Y})^n : h \in \mathcal{H}} \frac{1}{n} \sum_{i=1}^n \mathbb{1}_{h(x_i) \neq \mathcal{A}(\mathbf{s}_{-i}, \mathcal{H})(x_i)}.$$

We further define $\varepsilon_{\mathcal{A},\mathcal{H}} := \sup_{P \in \mathrm{RE}(\mathcal{H})} \varepsilon_{\mathcal{A},\mathcal{H},P}$, $\varepsilon_{\mathcal{H}} := \inf_{\mathcal{A}} \varepsilon_{\mathcal{A},\mathcal{H}}$, and $\varepsilon_{\mathcal{H},\mathrm{trans}} := \inf_{\mathcal{A}} \varepsilon_{\mathcal{A},\mathcal{H},\mathrm{trans}}$. By a leave-one-out argument [Brukhim et al., 2022, Fact 14], we observe that $\varepsilon_{\mathcal{A},\mathcal{H}} \leq \varepsilon_{\mathcal{A},\mathcal{H},\mathrm{trans}}$. Aden-Ali et al. [2023, Theorem 2.1] upper bounded the high probability error rate using the transductive error rate, which leads to a guarantee on PAC sample complexity. Based on their result, we prove the same upper bound up to a multiplicative constant on the high probability error rate using the expected error rate in Theorem 2.6.

Next, we define pseudo-cubes and DS dimensions of concept classes. Here, we also present their extensions to the setting of $k$-list learning for future reference in Section 2.1.

**Definition 1.3** (Pseudo-cube and $k$-pseudo-cube). *For any $d, k \in \mathbb{N}$, a class $H \subseteq \mathcal{Y}^d$ is called a $k$-**pseudo-cube** of dimension $d$ if it is non-empty, finite, and for every $h \in H$ and $i \in [d]$, there exist at least $k$ $i$-neighbors of $h$ in $H$, where $g$ is an $i$-neighbor of $h$ if $g(i) \neq h(i)$ and $g(j) = h(j)$ for all $j \in [d] \backslash \{i\}$. A **pseudo-cube** of dimension $d$ is a 1-pseudo-cube of dimension $d$.*

**Definition 1.4** (DS dimension and $k$-DS dimension, Charikar and Pabbaraju 2023). *For any $d, k \in \mathbb{N}$, we say $\mathbf{x} \in \mathcal{X}^d$ is $k$-**DS shattered** by $\mathcal{H} \subseteq \mathcal{Y}^{\mathcal{X}}$ if $\mathcal{H}|_{\mathbf{x}}$ contains a $d$-dimensional $k$-pseudo-cube. The $k$-**DS dimension** $\dim_k(\mathcal{H})$ of $\mathcal{H}$ is the maximum size of a $k$-DS shattered sequence. We say $\mathbf{x}$ is **DS shattered** by $\mathcal{H}$ if it is 1-DS shattered by $\mathcal{H}$. The **DS dimension** $\dim(\mathcal{H})$ of $\mathcal{H}$ is defined as $\dim_1(\mathcal{H})$.*

Now, we introduce some existing results in multiclass learning. Brukhim et al. [2022] proved that a class $\mathcal{H} \subseteq \mathcal{Y}^{\mathcal{X}}$ is PAC learnable if and only if $d := \dim(\mathcal{H}) < \infty$, and there exists a multiclass learner $\mathcal{A}$ which for any $P \in \mathrm{RE}(\mathcal{H})$, $\delta \in (0,1)$, $n \in \mathbb{N}$, and $S \sim P^n$, satisfies that with probability at least $1 - \delta$,

$$\mathrm{er}_P(\mathcal{A}(S, \mathcal{H})) = O\left(\frac{(d^{3/2} \log(d) + d \log(\log(n))) \log^2(n) + \log(1/\delta)}{n}\right), \tag{1}$$

which is also the best upper bound before this paper to our knowledge. In terms of lower bound, it follows from Charikar and Pabbaraju [2023, Theorem 6] that $\varepsilon_{\mathcal{H}}(n) = \Omega(d/n)$. Thus, the current upper and lower bounds of the expected error rate does not match. Moreover, a potentially sharp lower bound on the sample complexity $\mathcal{M}_{\mathcal{H}}$ is still missing.

The learner $\mathcal{A}$ in Brukhim et al. [2022] relies on orienting the one-inclusion graphs defined below as a building block.

**Definition 1.5** (One-inclusion graph, Haussler et al. 1994). *The **one-inclusion graph** (OIG) of $H \subseteq \mathcal{Y}^n$ for $n \in \mathbb{N}$ is a hypergraph $\mathcal{G}(H) = (H, E)$ where $H$ is the vertex-set and $E$ denotes the edge-set defined as follows. For any $i \in [n]$ and $f : [n] \backslash \{i\} \to \mathcal{Y}$, we define the set $e_{i,f} := \{h \in H : h(j) = f(j), \forall j \in [n] \backslash \{i\}\}$. Then, the edge-set is defined as*

$$E := \{(e_{i,f}, i) : i \in [n], f : [n] \backslash \{i\} \to \mathcal{Y}, e_{i,f} \neq \varnothing\}.$$

*For any $(e_{i,f}, i) \in E$ and $h \in H$, we say $h \in (e_{i,f}, i)$ if $h \in e_{i,f}$ and the size of the edge is $|(e_{i,f}, i)| := |e_{i,f}|$.*

Typically, we consider the one-inclusion graph of the projection of a concept class $\mathcal{H} \subseteq \mathcal{Y}^{\mathcal{X}}$ to a sequence $\mathbf{x} \in \mathcal{X}^n$ with $n \in \mathbb{N}$, i.e., $\mathcal{G}(\mathcal{H}|_{\mathbf{x}})$. The "density" of $\mathcal{H}$ discussed in Section 3 is defined via the "maximal average degree" (defined below) of the hypergraph $\mathcal{G}(\mathcal{H}|_{\mathbf{x}})$.

**Definition 1.6** (Degree and average degree). *For any hypergraph $G = (V, E)$ and $v \in V$, we define the **degree** of $v$ in $G$ to be $\deg(v; G) := |\{e \in E : v \in e, |e| \geqslant 2\}|$. When the underlying graph is clear in the context, we simply write $\deg(v)$ in abbreviation. If $|V| < \infty$, we can define the **average degree** and **average out-degree** of $G$ to be*

$$\mathsf{avgdeg}(G) := \frac{1}{|V|}\sum_{v \in V} \deg(v; G) = \frac{1}{|V|}\sum_{e \in E:|e|\geqslant 2} |e| \;\text{ and }\; \mathsf{avgoutdeg}(G) := \frac{1}{|V|}\sum_{e \in E}(|e| - 1).$$

*For general $V$, we can define the **maximal average degree** of $G$ to be $\mathsf{md}(G) := \sup_{U \subseteq V:|U|<\infty} \mathsf{avgdeg}(G[U])$, where $G[U] = (U, E[U])$ denotes the induced hypergraph of $G$ on $U \subseteq V$ with $E[U] := \{e \cap U : e \in E, e \cap U \neq \varnothing\}$.*

Note that for finite graphs, the average out-degree does not depend on the choice of orientation on $G$. Moreover, since $|e| \geqslant 2$ for all $e \in E_n$, we have

$$\mathsf{avgdeg}(G) = \frac{1}{|V|}\sum_{e \in E:|e|\geqslant 2} |e| \leqslant \frac{1}{|V|}\sum_{e \in E} 2(|e| - 1) = 2\mathsf{avgoutdeg}(G). \tag{2}$$

Now, we can define the density of a concept class as follows.

**Definition 1.7.** *The **density** of $\mathcal{H} \subseteq \mathcal{Y}^{\mathcal{X}}$ is defined as $\mu_{\mathcal{H}}(m) := \sup_{\mathbf{x} \in \mathcal{X}^m} \mathsf{md}(\mathcal{G}(\mathcal{H}|_{\mathbf{x}})), \; \forall m \in \mathbb{N}$.*

## 1.2 Main results

In this section, we summarize the key points of our theoretical results. The full versions of some results are stated in Section 2 and 3. We first need the following definition to rule out trivial concept classes for which one training point suffices to achieve zero error rate under any realizable distribution.

**Definition 1.8** (Nondegenerate concept class, Hanneke et al. 2023). *A concept class $\mathcal{H} \in \mathcal{Y}^{\mathcal{X}}$ is called **nondegenerate** if there exist $h_1, h_2 \in \mathcal{H}$ and $x_0, x_1 \in \mathcal{X}$ such that $h_1(x_0) = h_2(x_0)$ and $h_1(x_1) \neq h_2(x_1)$. $\mathcal{H}$ is called **degenerate** if it is not nondegenerate.*

Our main result on the multiclass PAC sample complexity is as follows.

**Theorem 1.9** (Partial summary of Theorem 2.5 and 2.11). *For any nondegenerate concept class $\mathcal{H} \subseteq \mathcal{Y}^{\mathcal{X}}$ with $\dim(\mathcal{H}) = d$ and any $(\varepsilon, \delta) \in (0, 1)^2$, we have*

$$\Omega((d + \log(1/\delta))/\varepsilon) \leqslant \mathcal{M}_{\mathcal{H}}(\varepsilon, \delta) \leqslant O((d^{3/2}\log(d)\log(d/\varepsilon) + \log(1/\delta))/\varepsilon). \tag{3}$$

Our upper bound follows from a reduction to list learning and an improved sample complexity for list learning summarized below.

**Theorem 1.10** (Informal summary of Theorem 2.7 and 2.10). *Assume that there exists a list learner which given a concept class $\mathcal{H}$ with $\dim(\mathcal{H}) = d$ and training sequence of size $n \in \mathbb{N}$ outputs a menu of size $p(\mathcal{H}, n)$ with expected error rate upper bounded by $\beta(\mathcal{H}, n)/n$ for some functions $p$ and $\beta$ nondecreasing in $n$. Then, there exists a multiclass learner whose error rate is*

$$O((\beta(\mathcal{H}, n) + d\log(p(\mathcal{H}, n)) + \log(1/\delta))/n) \text{ with probability at least } 1 - \delta.$$

*Moreover, there exists a list learner satisfying $p(\mathcal{H}, n) = O\big((e\sqrt{d})^{\sqrt{d}}\log(n)\big)$ and $\beta(\mathcal{H}, n) = O\big(d^{3/2}\log(d)\log(n)\big)$.*

We refer readers to Section 2.1 for detailed definitions regarding list learning. Note that if $p(\mathcal{H}, n)$ and $\beta(\mathcal{H}, n)$ of some list learner is independent of $n$, there exists a multiclass learner with error rate linear in $1/n$. We leave the establishment of such list learners as Open Question 1.

In addition to the above approach, we propose an alternative route toward obtaining the conjectured $\Theta((d + \log(1/\delta))/\varepsilon)$ sample complexity, by directly bounding the average degrees of one-inclusion graphs. In particular, we show in Theorem 3.2 that any $\mathcal{H}$ with $\dim(\mathcal{H}) = 1$ has $\mathcal{M}_{\mathcal{H}}(\varepsilon, \delta) = \Theta(\log(1/\delta)/\varepsilon)$, which was not previously known. Moreover, we approach the general case via a new technique we call "pivot shifting". Specifically, we obtain the following result, which relies on an assumption on such pivot shifting. The verification of this assumption is left as Open Question 2.

**Proposition 1.11** (Informal summary of the results in Section 3). *Assume that for any finite concept class, there exists a pivot shifting such that the DS dimension of the concept class does not increase after the pivot shifting, then we have $\mathcal{M}_{\mathcal{H}}(\varepsilon, \delta) = \Theta\big((\dim(\mathcal{H}) + \log(1/\delta))/\varepsilon\big)$ for any $\mathcal{H} \subseteq \mathcal{Y}^{\mathcal{X}}$.*

## 2 Multiclass learning via list learning

In this section, we prove a reduction from multiclass learning to list learning in Section 2.2. We improve the existing list learners using boosting in Section 2.3. Then, using a boosted list learner and the reduction, we improve the multiclass learning sample complexity upper bound in Section 2.4. We first present some definitions and results of list learning in Section 2.1.

### 2.1 List learning

In list learning, the menus defined below serve as classifiers in multiclass learning.

**Definition 2.1** ($k$-menu, Brukhim et al. 2022). *A **menu** of size $k \in \mathbb{N}$ is a function $\mu : \mathcal{X} \to \{\mathsf{Y} \subseteq \mathcal{Y} : |\mathsf{Y}| \leqslant k\}$. A 1-menu can be viewed as a classifier in $\mathcal{Y}^{\mathcal{X}}$, and vice versa.*

For any distribution $P$ over $\mathcal{X} \times \mathcal{Y}$, the error rate of a $k$-menu $\mu$ under $P$ is defined as $\mathrm{er}_P(\mu) := P(\{(x, y) \in \mathcal{X} \times \mathcal{Y} : y \notin \mu(x)\})$ which agrees with the definition of the error rate of classifiers when the size of the menu is 1.

**Definition 2.2** ($k$-list learner). *A **list learner** $\mathcal{A}$ of size $k \in \mathbb{N}$ is an algorithm which given a sequence $\mathbf{s} \in \cup_{n=0}^{\infty}(\mathcal{X} \times \mathcal{Y})^n$ and a concept class $\mathcal{H} \subseteq \mathcal{Y}^{\mathcal{X}}$, outputs a $k$-menu $\mathcal{A}(\mathbf{s}, \mathcal{H})$. A 1-list learner can be viewed as a multiclass learner, and vice versa.*

Similar to multiclass learners, the expected error rate of a list learner $\mathcal{A}$ is defined as

$$\varepsilon_{\mathcal{A},\mathcal{H},P} : \mathbb{N} \to [0,1], \quad n \mapsto \mathbb{E}_{S \sim P^n}[\mathrm{er}_P(\mathcal{A}(S, \mathcal{H}))] = \mathbb{P}_{(S,(X,Y)) \sim P^{n+1}}(Y \notin \mathcal{A}(S, \mathcal{H})(X))$$

for any concept class $\mathcal{H} \in \mathcal{Y}^{\mathcal{X}}$ and distribution $P$ over $\mathcal{X} \times \mathcal{Y}$. Restricting to realizable distributions, we can define

$$\varepsilon_{\mathcal{A},\mathcal{H}} := \sup_{P \in \mathrm{RE}(\mathcal{H})} \varepsilon_{\mathcal{A},\mathcal{H},P} \quad \text{and} \quad \varepsilon_{\mathcal{H}}^k := \inf_{k\text{-list learners } \mathcal{A}} \varepsilon_{\mathcal{A},\mathcal{H}}.$$

Next, we define list PAC learning.

**Definition 2.3** (List PAC learning, Charikar and Pabbaraju 2023). *For any concept class $\mathcal{H} \subseteq \mathcal{Y}^{\mathcal{X}}$ and $k \in \mathbb{N}$, the **(PAC) sample complexity** $\mathcal{M}_{\mathcal{A},\mathcal{H}} : (0,1)^2 \to \mathbb{N}$ of a $k$-list learner $\mathcal{A}$ is a mapping from $(\varepsilon, \delta) \in (0,1)^2$ to the smallest positive integer such that for every $m \geqslant \mathcal{M}_{\mathcal{A},\mathcal{H}}(\varepsilon, \delta)$ and every distribution $P \in \mathrm{RE}(\mathcal{H})$, $\mathbb{P}_{S \sim P^n}(\mathrm{er}_P(\mathcal{A}(S, \mathcal{H})) > \varepsilon) \leqslant \delta$, and we define $\mathcal{M}_{\mathcal{A},\mathcal{H}}(\varepsilon, \delta) = \infty$ if no such integer exists. We say $\mathcal{H}$ is $k$-**list PAC learnable** by $\mathcal{A}$ if $\mathcal{M}_{\mathcal{A},\mathcal{H}}(\varepsilon, \delta) < \infty$ for all $(\varepsilon, \delta) \in (0,1)^2$. The $k$-**list (PAC) sample complexity** of $\mathcal{H}$ is defined as $\mathcal{M}_{\mathcal{H}}^k(\varepsilon, \delta) := \inf_{k\text{-list learner } \mathcal{A}} \mathcal{M}_{\mathcal{A},\mathcal{H}}(\varepsilon, \delta)$ for any $(\varepsilon, \delta) \in (0,1)^2$.*

Note that $\mathcal{H}$ is PAC learnable by a learner $\mathcal{A}$ if and only if $\mathcal{H}$ is 1-list learnable by $\mathcal{A}$, and the PAC sample complexity of $\mathcal{H}$ is $\mathcal{M}_{\mathcal{H}} = \mathcal{M}_{\mathcal{H}}^1$. For list PAC learning, it was proved by Charikar and Pabbaraju [2023] that a concept class $\mathcal{H}$ is $k$-list learnabale if and only if $d_k := \dim_k(\mathcal{H}) < \infty$, and there exists a $k$-list learner $\mathcal{A}^k$ which for any $P \in \mathrm{RE}(\mathcal{H})$, $\delta \in (0,1)$, $n \in \mathbb{N}$, and $S \sim P^n$, satisfies that with probability at least $1 - \delta$,

$$\mathrm{er}_P(\mathcal{A}^k(S, \mathcal{H})) = O\left(\frac{k^6 d_k(\sqrt{d_k}\log(d_k) + \log(k\log(n)))\log^2(n) + \log(1/\delta)}{n}\right). \tag{4}$$

For the expected error rate, the lower bound $\varepsilon_{\mathcal{H}}^k(n) = \Omega\left(d_k/(kn)\right)$ has been proved in Charikar and Pabbaraju [2023, Theorem 6]. However, a lower bound of the same order on the $k$-list PAC sample complexity is still missing in the literature. To establish a lower bound, we also need to rule out trivial classes in list learning. In analogy to Definition 1.8, we define $k$-nondegeneracy as follows.

**Definition 2.4** ($k$-nondegenerate concept class). *A concept class $\mathcal{H} \in \mathcal{Y}^{\mathcal{X}}$ is called $k$-**nondegenerate** for $k \in \mathbb{N}$ if there exist $h_1, \ldots, h_{k+1} \in \mathcal{H}$ and $x_0, x_1 \in \mathcal{X}$ such that $|\{h_j(x_0) : j \in [k+1]\}| = 1$ and $|\{h_j(x_1) : j \in [k+1]\}| = k + 1$. $\mathcal{H}$ is called $k$-**degenerate** if it is not $k$-nondegenerate.*

We claim that only one training point is sufficient for the $k$-list learning of a $k$-degenerate concept class $\mathcal{H}$. Indeed, $\mathcal{H}$ is $k$-list learnable if $|\mathcal{H}| \leqslant k$. Now, suppose that $|\mathcal{H}| \geqslant k + 1$ and is $k$-degenerate. Upon observing any point $(x', y') \in \mathcal{X} \times \mathcal{Y}$ realizable by $\mathcal{H}$, if $|\{h \in \mathcal{H} : h(x') = y'\}| \leqslant k$, then $x \mapsto \{h(x) : h \in \mathcal{H}, h(x') = y'\}$ is a $k$-menu which always contains the correct label. If $|\{h \in \mathcal{H} : h(x') = y'\}| \geqslant k + 1$, then, for any $x \in \mathcal{X} \backslash \{x'\}$, we must have $|\{h(x) : h \in \mathcal{H}, h(x') = y'\}| \leqslant k$

because otherwise $\mathcal{H}$ is $k$-nondegenerate. Then, $x \mapsto \{h(x) : h \in \mathcal{H}, h(x') = y'\}$ is a $k$-list which always contains the correct label.

Now, we are ready to present the following lower bound on the $k$-list PAC sample complexity.

**Theorem 2.5.** *For any $k \in \mathbb{N}$, $k$-nondegenerate concept class $\mathcal{H} \subseteq \mathcal{Y}^{\mathcal{X}}$ with $\dim_k(\mathcal{H}) = d_k \in \mathbb{N}$, $\varepsilon \in \left(0, \frac{1}{8(k+1)}\right)$, and $\delta \in \left(0, \frac{1}{4(k+1)}\right)$, we have $\mathcal{M}_{\mathcal{H}}^k(\varepsilon, \delta) \geqslant \frac{(d_k - 1)\log(2) + 4\log(1/\delta)}{16(k+1)\varepsilon}$. In particular, when $k = 1$, for any $\varepsilon \in (0, 1/16)$ and $\delta \in (0, 1/8)$, we have*

$$\mathcal{M}_{\mathcal{H}}(\varepsilon, \delta) \geqslant \frac{(\dim(\mathcal{H}) - 1)\log(2) + 4\log(1/\delta)}{32\varepsilon}. \tag{5}$$

The proof of Theorem 2.5 is presented in Appendix A where we construct hard distributions based on properties of $k$-pseudo-cubes.

## 2.2 Reduction from multiclass learning to list learning

We first introduce the theorem that provides a guarantee on PAC sample complexity based on expected error rate, which will be used frequently in our analysis.

**Theorem 2.6.** *Fix a concept class $\mathcal{H} \subseteq \mathcal{Y}^{\mathcal{X}}$ and consider a learner $\mathcal{A}$ which satisfies $\varepsilon_{\mathcal{A},\mathcal{H},P}(n) \leqslant M_n/n$ for any $n \in \mathbb{N}$ and $P \in \mathrm{RE}(\mathcal{H})$ with $M_n$ nondecreasing in the sample size $n$. Then, there exists a learner $\mathcal{A}'$ such that for any $P \in \mathrm{RE}(\mathcal{H})$, $\delta \in (0, 1)$, $n \geqslant 4$, and the training sequence $S \sim P^n$, with probability at least $1 - \delta$, we have*

$$\mathrm{er}_P(\mathcal{A}'(S, \mathcal{H})) \leqslant 4.82 \cdot (8.34 M_{\lfloor n/2 \rfloor} + \log(2/\delta))/n.$$

The proof of Theorem 2.6 is provided in Appendix B. Now, we consider general list learners whose sizes may depend on the sample size. The theorem below states our reduction to list learning.

**Theorem 2.7.** *Assume that there exists a list learner $\mathcal{A}_{\mathrm{list}}$ which for any $\mathcal{H} \subseteq \mathcal{Y}^{\mathcal{X}}$, $\mathcal{D} \in \mathrm{RE}(\mathcal{H})$, $n \in \mathbb{N}$, and $S \sim \mathcal{D}^n$, outputs a menu $\mathcal{A}_{\mathrm{list}}(S, \mathcal{H})$ of size $p(\mathcal{H}, n)$ satisfying $\varepsilon_{\mathcal{A}_{\mathrm{list}}, \mathcal{H}, \mathcal{D}} \leqslant \beta(\mathcal{H}, n)/n$ for some function $\beta : 2^{\mathcal{Y}^{\mathcal{X}}} \times \mathbb{N} \to [0, \infty)$. Without loss of generality, we assume that $p(\mathcal{H}, n)$ and $\beta(\mathcal{H}, n)$ are nondecreasing in $n$. Then, there exist multiclass learners $\mathcal{A}_{\mathrm{red}}$ (see Algorithm 1) and $\mathcal{A}'_{\mathrm{red}}$ which for any concept class $\mathcal{H}$ of DS dimension $d$, $\mathcal{D} \in \mathrm{RE}(\mathcal{H})$, $\delta \in (0, 1)$, and $n \geqslant 4$, satisfy*

$$\varepsilon_{\mathcal{A}_{\mathrm{red}}, \mathcal{H}, \mathcal{D}}(n) = O\left((\beta(\mathcal{H}, n_1) + d\log p(\mathcal{H}, n_1))/n\right)$$

*where $n_1 := n - 2\lfloor n/3 \rfloor$, and for $S \sim \mathcal{D}^n$, with probability at least $1 - \delta$,*

$$\mathrm{er}_{\mathcal{D}}(\mathcal{A}'_{\mathrm{red}}(S, \mathcal{H})) = O\left((\beta(\mathcal{H}, n_1) + d\log p(\mathcal{H}, n_1) + \log(1/\delta))/n\right). \tag{6}$$

The proof of Theorem 2.7 is presented in Appendix C. Note that the order of the error rate upper bound of the constructed multiclass learner is not smaller than that of the original list learner in the above theorem. Thus, the list learner $\mathcal{A}^k$ of size $k \in \mathbb{N}$ developed in Charikar and Pabbaraju [2023] cannot lead to an improved error rate of multiclass learning using our current result. The construction of $\mathcal{A}_{\mathrm{red}}$ from $\mathcal{A}_{\mathrm{list}}$ is shown in Algorithm 1.

---

**Algorithm 1:** Multiclass learner $\mathcal{A}_{\mathrm{red}}$ using a list learner $\mathcal{A}_{\mathrm{list}}$

---

**Input:** List learner $\mathcal{A}_{\mathrm{list}}$, concept class $\mathcal{H} \subseteq \mathcal{Y}^{\mathcal{X}}$, training sequence
$\quad\quad S = ((x_1, y_1), \ldots, (x_n, y_n)) \in (\mathcal{X} \times \mathcal{Y})^n$ for $n \geqslant 3$, test feature $x_{n+1} \in \mathcal{X}$.
**Output:** A label $y \in \mathcal{Y}$ for the feature $x_{n+1}$.
1 $n_1 \leftarrow n - 2\lfloor n/3 \rfloor$, $n_2 \leftarrow \lfloor n/3 \rfloor$;
2 $S^1 \leftarrow ((x_i, y_i))_{i \in [n_1]}$, $S^2 \leftarrow ((x_i, y_i))_{i=n_1+1}^{n}$, $\mathbf{x}' \leftarrow (x_{n_1+1}, \ldots, x_n, x_{n+1})$;
3 $\hat{\mu} \leftarrow \mathcal{A}_{\mathrm{list}}(S^1, \mathcal{H})$, $N \leftarrow \sum_{(x,y) \in S^2} \mathbb{1}_{y \notin \hat{\mu}(x)}$;
4 $\mathcal{H}_{\mathbf{x}'} \leftarrow \{h|_{\mathbf{x}'} : h \in \mathcal{H}, |\{i \in [n+1] \backslash [n_1] : h(x_i) \notin \hat{\mu}(x_i)\}| \leqslant N + 1\}$;
5 Sample $(I_1, \ldots, I_{n_2}) \sim \mathrm{Unif}([2n_2])^{n_2}$;
6 $\hat{h} \leftarrow A_G(T, \mathcal{H}_{\mathbf{x}'})$ where $T \leftarrow ((I_j, y_{I_j+n_1}))_{j \in [n_2]}$;
7 **return** the label $\hat{h}(2n_2 + 1)$.

---

In step 6 of Algorithm 1, $A_G$ is a multiclass learner defined in Proposition H.5 in the appendix. Moreover, we prove in Proposition H.5 that for any $\mathcal{D} \in \mathrm{RE}(\mathcal{H})$, $n \in \mathbb{N}$, $\delta \in (0, 1)$, and $S \sim \mathcal{D}^n$, with probability at least $1 - \delta$, we have

$$\mathrm{er}_{\mathcal{D}}(A_G(S, \mathcal{H})) = O\left((\dim_G(\mathcal{H}) + \log(1/\delta))/n\right)$$

where $\dim_G(\mathcal{H})$ is the graph dimension [Natarajan and Tadepalli, 1988] of $\mathcal{H}$ (see Definition H.1 in the appendix). The above bound for classes of finite graph dimensions is also novel in the literature.

We briefly comment on the analysis of Algorithm 1. We first apply the list learner to the first third of the training samples to obtain the menu $\widehat{\mu}$. Then, we count the number of errors ($N$) made by $\widehat{\mu}$ in the last two thirds samples. Then, we consider $\mathcal{H}_{\mathbf{x}'}$ which is a subset of $\mathcal{H}|_{\mathbf{x}'}$ such that the number of errors on $\mathbf{x}'$ is bounded by $N + 1$. We show in Lemma H.9 that $\dim_G(\mathcal{H}_{\mathbf{x}'})$ is well controlled. However, as we do not observe the label of the test point, we can only consider resampling from elements in $S^2$ as the new training sequence fed to $A_G$ together with the concept class $\mathcal{H}_{\mathbf{x}'}$. Thus, there still exist great challenges of upper bounding the error probability for the test point that will never be sampled. We need to emphasize that the standard leave-one-out argument [Brukhim et al., 2022, Fact 14] cannot be directly applied as the definition of $N$ that determines $\mathcal{H}_{\mathbf{x}'}$ only depends on $S^2$ but not the test point $(X_{n+1}, Y_{n+1})$. We tackle this challenge by proving that some permutation of the error event together with the constraint on correctness of $\widehat{\mu}$ on the last two points in $\mathbf{x}'$ when leaving the last element (i.e., the test point) out is a subset of the error event when leaving the previous element (i.e., the point in $S^2$) out. The details of the proof is presented in Appendix C.

### 2.3 Sampled boosting of list learners

We now build a list learner whose invocation to Theorem 2.7 yields the upper bound in Theorem 1.9. Brukhim et al. [2022, Lemma 39] proposed a list sample compression scheme of size $r = O(d^{3/2} \log(n))$ for concept classes of DS dimension $d$ and sample size $n$. One can show that its error rate is $O\left((r \log(n/r) + \log(1/\delta))/n\right)$ using standard techniques for sample compression schemes [David et al., 2016], which however brings the extra log factor $\log(n/r)$. Recently, da Cunha et al. [2024] proposed stable randomized sample compression schemes for binary classification whose generalization does not induce the extra log factor in $n$ and used this framework to analyze a subsampling-based boosting algorithm for weak learners. Motivated by its success, we extend their boosting algorithm [da Cunha et al., 2024, Algorithm 1] for multiclass list learners in Algorithm 2. Before presenting the algorithm, we first need to define the majority vote of menus. For $K \in \mathbb{N}$ menus $\mu_1, \ldots, \mu_K$ each of size $p$, we define their majority vote to be $\mu = \mathrm{Maj}(\mu_1, \ldots, \mu_k)$ with

$$\mu(x) = \mathrm{Maj}(\mu_1, \ldots, \mu_k)(x) := \{y \in \mathcal{Y} : |\{k \in [K] : y \in \mu_k(x)\}| > K/2\}, \ \forall x \in \mathcal{X}.$$

Note that $\mu$ has size $2p - 1$. For $p = 1$, the above definition recovers the majority vote of classifiers.

---

**Algorithm 2:** Sampled boosting $\mathcal{A}_{\mathrm{boost}}$ of a list learner $\mathcal{A}_{\mathrm{list}}$

**Input:** List learner $\mathcal{A}_{\mathrm{list}}$, concept class $\mathcal{H} \subseteq \mathcal{Y}^{\mathcal{X}}$, training sequence
   $S = \{(x_1, y_1), \ldots, (x_n, y_n)\} \in (\mathcal{X} \times \mathcal{Y})^n, \gamma \in (0, 1/2), \nu \in (0, \gamma/18], \delta \in (0, 1)$.
**Output:** Menu $\mu$.

1 **for** $i = 1, \ldots, n$ **do**
2    $\mathcal{D}_1(\{(x_i, y_i)\}) \leftarrow 1/n$;
3 $\alpha \leftarrow \frac{1}{2} \log \left((1 + \gamma)/(1 - \gamma)\right), m \leftarrow \mathcal{M}_{\mathcal{A}_{\mathrm{list}}, \mathcal{H}}(1/2 - \gamma, \nu), K \leftarrow \lceil 4 \log(n/\delta)/\gamma \rceil$;
4 **for** $k = 1, \ldots, K$ **do**
5    Draw $m$ samples $S^k \sim \mathcal{D}_k^m$;
6    $\mu_k \leftarrow \mathcal{A}_{\mathrm{list}}(S^k, \mathcal{H})$;
7    **for** $i = 1, \ldots, n$ **do**
8      $\mathcal{D}_{k+1}(\{(x_i, y_i)\}) \leftarrow \mathcal{D}_k(\{(x_i, y_i)\}) \exp \left(-\alpha \left(2\mathbb{1}_{y_i \in \mu_k(x_i)} - 1\right)\right)$;
9    $\mathcal{D}_{k+1} \leftarrow \mathcal{D}_{k+1}/\left(\sum_{i=1}^n \mathcal{D}_k(\{(x_i, y_i)\}) \exp \left(-\alpha \left(2\mathbb{1}_{y_i \in \mu_k(x_i)} - 1\right)\right)\right)$;
10 **return** $\mu \leftarrow \mathrm{Maj}\left((\mu_k)_{k \in [K]}\right)$.

---

Here, $\gamma$ and $\nu$ are fixed constants, enabling us to invoke weak list learners (of constant error and confidence levels) to Algorithm 2. Next, we upper bound the error rate of the boosted list learner.

**Theorem 2.8.** *Assume that $\mathcal{A}_{\mathrm{list}}$ is a list learner with $\mathcal{M}_{\mathcal{A}_{\mathrm{list}}, \mathcal{H}}(1/2 - \gamma, \nu) < \infty$ for some $\gamma \in (0, 1/2)$ and $\nu \in (0, \gamma/18]$. Then, for any $\mathcal{D} \in \mathrm{RE}(\mathcal{H})$, $n \in \mathbb{N}$, and $\delta > 0$, sampling $S \sim \mathcal{D}^n$, with probability at least $1 - \delta$, the menu $\mu$ produced by $\mathcal{A}_{\mathrm{boost}}$ using $\mathcal{A}_{\mathrm{list}}$ in Algorithm 2 satisfies that*

$$\mathrm{er}_{\mathcal{D}}(\mu) = O\left(\frac{\mathcal{M}_{\mathcal{A}_{\mathrm{list}}, \mathcal{H}}(1/2 - \gamma, \nu) \log(n/\delta)}{\gamma n}\right).$$

The proof of Theorem 2.8 is a generalization of the proof of da Cunha et al. [2024, Theorem 1.1] and is presented in Appendix D together with the proofs of other results in this section. Since multiclass learners are list learners of size 1, we can also boost multiclass learners using Algorithm 2. For instance, invoking the multiclass learner in Brukhim et al. [2022, Theorem 1] to Algorithm 2 and applying Theorem 2.6, we achieve the following sample complexity in multiclass learning.

**Corollary 2.9.** *There exists a multiclass learner $\mathcal{A}$ with $\varepsilon_{\mathcal{A},\mathcal{H}}(n) = O\big(\frac{d^{3/2}\log^2(d)\log(n)}{n}\big)$ and $\mathcal{M}_{\mathcal{A},\mathcal{H}}(\varepsilon,\delta) = O\big(\frac{d^{3/2}\log^2(d)\log(d/\varepsilon)+\log(1/\delta)}{\varepsilon}\big)$ for any $n \in \mathbb{N}$, $(\varepsilon,\delta) \in (0,1)^2$, and $\mathcal{H} \subseteq \mathcal{Y}^{\mathcal{X}}$ with $\dim(\mathcal{H}) = d$.*

There is an extra $\log(d)$ factor in the above upper bound compared to that in Theorem 1.9, which explains the reason of routing through list learning with our reduction in Theorem 2.7: the list sample compression scheme in Brukhim et al. [2022] saves a $\log(d)$ factor compared to their sample compression scheme. Therefore, we invoke their list sample compression scheme as $\mathcal{A}_{\text{list}}$ to Algorithm 2 and build a list learner whose error rate depends on only one log factor in both $n$ and $d$.

**Theorem 2.10.** *There exists a list learner $\mathcal{A}_L$ which for any $\mathcal{H} \subseteq \mathcal{Y}^{\mathcal{X}}$ with $\dim(\mathcal{H}) = d$ and sample size $n \in \mathbb{N}$ outputs a menu of size $O\big((e\sqrt{d})^{\sqrt{d}}\log(n)\big)$ with $\varepsilon_{\mathcal{A}_L,\mathcal{H}}(n) = O\big(\frac{d^{3/2}\log(d)\log(n)}{n}\big)$.*

### 2.4 Improved upper bounds on sample complexity

Applying the list learner $\mathcal{A}_L$ in Theorem 2.10 to our reduction, Theorem 2.7 immediately implies the following result.

**Theorem 2.11.** *There exists a multiclass learner $\mathcal{A}_{\text{multi}}$ such that for any $\mathcal{H} \subseteq \mathcal{Y}^{\mathcal{X}}$ of DS dimension $d$, $\mathcal{D} \in \text{RE}(\mathcal{H})$, $\delta \in (0,1)$, $n \geq d+1$, and $S \sim \mathcal{D}^n$, with probability at least $1-\delta$, we have*

$$\text{er}_{\mathcal{D}}(\mathcal{A}_{\text{multi}}(S,\mathcal{H})) = O\left(\frac{d^{3/2}\log(d)\log(n)+\log(1/\delta)}{n}\right), \tag{7}$$

*which implies that*

$$\mathcal{M}_{\mathcal{A}_{\text{multi}},\mathcal{H}}(\varepsilon,\delta) = O\left(\frac{d^{3/2}\log(d)\log(d/\varepsilon)+\log(1/\delta)}{\varepsilon}\right), \ \forall \varepsilon,\delta \in (0,1). \tag{8}$$

*Furthermore, if there exists a list learner $\mathcal{A}_{\text{goodlist}}$ of size $f_1(d)$ and expected error rate $\varepsilon_{\mathcal{A}_{\text{goodlist}},\mathcal{H}}(n) \leq f_2(d)/n$ for some functions $f_1 : \mathbb{N} \to \mathbb{N}$ and $f_2 : \mathbb{N} \to [0,\infty)$, then, there exists a multiclass learner $\mathcal{A}_{\text{lin}}$ such that*

$$\mathcal{M}_{\mathcal{A}_{\text{lin}},\mathcal{H}}(\varepsilon,\delta) = O\left((d\log(f_1(d)) + f_2(d) + \log(1/\delta))/\varepsilon\right), \ \forall \varepsilon,\delta \in (0,1). \tag{9}$$

The proof of Theorem 2.11 follows directly from Theorem 2.7 and Theorem 2.10 and is provided in Appendix E. Moreover, observing that $\dim_k(\mathcal{H}) \geq \dim_{k'}(\mathcal{H})$ for $k < k'$, our requirement on $\mathcal{A}_{\text{goodlist}}$ does not violate the lower bound in Charikar and Pabbaraju [2023, Theorem 6].

Compared to the upper bound (1) by Brukhim et al. [2022], (7) improves the dependence of the error rate on the sample size $n$ from $O\left(\log(\log(n))\log^2(n)/n\right)$ to $O\left(\log(n)/n\right)$, which steps further towards the goal of $O(1/n)$ expected error rate (treating the DS dimension as a constant). Combining (5) and (8), we arrive at (3) where the gap has been improved to the factor $\sqrt{d}\log(d)\log(d/\varepsilon)$. However, we are not aware of any existing list learner satisfying the requirements of $\mathcal{A}_{\text{goodlist}}$ in Theorem 2.11. Thus, we leave the construction of $\mathcal{A}_{\text{goodlist}}$ as an open question.

**Open Question 1.** *Does there exist a list learner such that given a concept class $\mathcal{H} \subseteq \mathcal{Y}^{\mathcal{X}}$, its size is $f_1(\dim(\mathcal{H}))$ and its expected error rate is $\varepsilon_{\mathcal{A}_{\text{list}},\mathcal{H}}(n) = f_2(\dim(\mathcal{H}))/n$ for some functions $f_1 : \mathbb{N} \to \mathbb{N}$ and $f_2 : \mathbb{N} \to [0,\infty)$?*

Ideally, we would expect a list learner with size $O(\dim(\mathcal{H}))$ and expected error rate $O(\dim(\mathcal{H})/n)$ as it immediately implies an upper bound $\mathcal{M}_{\mathcal{H}}(\varepsilon,\delta) = O((\dim(\mathcal{H})\log(\dim(\mathcal{H})) + \log(1/\delta))/\varepsilon)$ which matches the lower bound in (5) up to the factor $\log(\dim(\mathcal{H}))$.

## 3 Density, DS dimension, and pivot shifting

We now introduce an alternative route toward proving the optimal sample complexity of multiclass PAC learning: bounding the density $\mu_{\mathcal{H}} : \mathbb{N} \to [0,\infty)$ (Definition 1.7) of concept classes $\mathcal{H}$. The

following proposition summarizes existing results that illustrate the role of density in multiclass learning.

**Proposition 3.1** (Daniely and Shalev-Shwartz 2014, Charikar and Pabbaraju 2023, Aden-Ali et al. 2023). *For any $\mathcal{H} \subseteq \mathcal{Y}^{\mathcal{X}}$ and $n \in \mathbb{N}$, we have*

$$\mu_{\mathcal{H}}(n)/(2en) \leqslant \varepsilon_{\mathcal{H}} \leqslant \varepsilon_{\mathcal{H},\text{trans}} \leqslant \mu_{\mathcal{H}}(n)/n. \tag{10}$$

*Assume that $\mu_{\mathcal{H}}(n) \leqslant f(\dim(\mathcal{H}))$ for some function $f : \mathbb{N} \to [0, \infty)$ and all $n \in \mathbb{N}$. Then, there exists a learner $\mathcal{A}$ based on orienting the one-clusion graph (Definition 1.5) of the projected concept class (see Aden-Ali et al. [2023, Appendix A] for the formal definition of the algorithm) with sample complexity $\mathcal{M}_{\mathcal{A},\mathcal{H}}(\varepsilon, \delta) = O\left(\frac{f(\dim(\mathcal{H})) + \log(1/\delta)}{\varepsilon}\right)$ for all $\varepsilon, \delta \in (0, 1)$.*[1]

In Proposition 3.1, the first inequality of (10) follows from Charikar and Pabbaraju [2023, Theorem 6], the last inequality of (10) follows from Daniely and Shalev-Shwartz [2014, Theorem 2], and the last paragraph follows from Aden-Ali et al. [2023, Theorem 2.2]. Thus, for sharper multiclass sample complexity, it suffices to bound the density of a concept class with some functions of its DS dimension. Furthermore, by Definition 1.7 and (2), it suffices to bound the average out-degree (Definition 1.6) of finite one-inclusion graphs. In fact, it has been conjectured that $\mu_{\mathcal{H}}(n) \leqslant c \cdot \dim(\mathcal{H})$ for some constant $c > 0$ [Daniely and Shalev-Shwartz, 2014] and the question remained open since then. A positive resolution of this conjecture would immediately imply that the $\mathcal{M}_{\mathcal{H}}(\varepsilon, \delta) = \Theta\left((\dim(\mathcal{H}) + \log(1/\delta))/n\right)$ by Proposition 3.1 and Theorem 2.5. It is worth mentioning that for $\mathcal{H} \subseteq \{0, 1\}^{\mathcal{X}}$, Haussler et al. [1994] proved that $\mu_{\mathcal{H}} \leqslant 2\dim(\mathcal{H})$ (for binary classes, the DS dimension is the VC dimension), which also motivates the above conjecture. In this paper, we confirm the above conjecture for concept classes of DS dimension 1.

**Theorem 3.2.** *For any $\mathcal{H} \subseteq \mathcal{Y}^{\mathcal{X}}$ with $\dim(\mathcal{H}) = 1$, we have $\mu_{\mathcal{H}}(n) \leqslant 2$, $\forall n \in \mathbb{N}$. Thus, $\mathcal{M}_{\mathcal{H}}(\varepsilon, \delta) = \Theta\left(\log(1/\delta)/\varepsilon\right)$ for any positive $\varepsilon, \delta \in O(1)$ and any $\mathcal{H}$ with $\dim(\mathcal{H}) = 1$.*

The above theorem follows from the following fact we prove for one-inclusion graphs of DS dimension 1 concept classes. The proofs of Theorem 3.2 and Proposition 3.3 are presented in Appendix F.

**Proposition 3.3.** *For any $n \in \mathbb{N}$ and $V_n \subseteq \mathcal{Y}^n$ with $|V_n| < \infty$ and $\dim(V_n) = 1$, there exists no cycle (see Definition 3.4) in the one-inclusion graph $\mathcal{G}(V_n)$ (see Definition 1.5).*

**Definition 3.4** (Cycle in finite hypergraph). *A **cycle** of length $m \in \mathbb{N}\backslash\{1\}$ in a finite hyergraph $G = (V, E)$ consists of pairwise different vertices $v^0, \ldots, v^{m-1} \in V$ and pairwise different edges $e^0, \ldots, e^{m-1} \in E$ such that $v^j, v^{(j+1) \mod m} \in e^j$ for all $0 \leqslant j \leqslant m - 1$.*

We prove Proposition 3.3 by contradiction and analyzing different cases of the cycle. However, it is hard to extend such result to classes of higher DS dimensions. For general concept classes, motivated by the proof for binary classes [Haussler et al., 1994, Lemma 2.4], we also consider proving by induction on the size of the sequence the class projects to. Though the analysis for binary classes does not apply to general concept classes, we discover that the analysis in the induction step proceeds seamlessly for some special concept classes where a common label which we call a "pivot" exists for each edge in the last dimension of size greater than 1 in its one-inclusion graph. Before summarizing this result in Lemma 3.6 below, we first introduce the definition of a "pivot" formally.

**Definition 3.5** (Pivot of finite concept class). *For any $n \in \mathbb{N}\backslash\{1\}$ and $V_n \subseteq \mathcal{Y}^n$, we define*

$$\mathfrak{P}(V_n) := \cup_{y \in \mathcal{Y}} \cup_{y' \in \mathcal{Y}\backslash\{y\}} \left\{(y_1, \ldots, y_{n-1}) \in \mathcal{Y}^{n-1} : (y_1, \ldots, y_{n-1}, y), (y_1, \ldots, y_{n-1}, y') \in V_n\right\}.$$

*Then, $a \in \mathcal{Y}$ is said to be a **pivot** of $V_n$ if $(y_1, \ldots, y_{n-1}, a) \in V_n$ for any $(y_1, \ldots, y_{n-1}) \in \mathfrak{P}(V_n)$. We emphasize that when $\mathfrak{P}(V_n) = \varnothing$, every $a \in \mathcal{Y}$ is a pivot of $V_n$.*

Then, we can present Lemma 3.6 whose proof is provided in Appendix G.

**Lemma 3.6.** *Assume that for some $n \in \mathbb{N}\backslash\{1\}$, any $d \in \mathbb{N}$, any $m \in [n - 1]$, and any $H \subseteq \mathcal{Y}^m$ with $\dim(H) \leqslant d$ and $|H| < \infty$, we have $\mathsf{avgoutdeg}(\mathcal{G}(H)) \leqslant d$. Consider an arbitrary set $V_n \subseteq \mathcal{Y}^n$ such that $|V_n| < \infty$ and $\dim(V_n) \leqslant d$. If $V_n$ has a pivot, then we have $\mathsf{avgoutdeg}(\mathcal{G}(V_n)) \leqslant d$.*

---

[1]In Aden-Ali et al. [2023, Section 2.4], the label space considered is finite. However, extending the compactness argument in Brukhim et al. [2022, Appendix B], we can prove that there exists an orientation of the hypergraph with its maximum out-degree upper bounded by the ceiling of the density even when the graph is infinite, which implies that the above sample complexity of the learner $\mathcal{A}$ still holds for infinite label spaces.

Though it only works for special classes, Lemma 3.6 can serve as a building block in the induction step for the proof of $\mathsf{avgoutdeg}(\mathcal{G}(\mathcal{H})) \leqslant \dim(\mathcal{H})$ for finite $\mathcal{H} \subseteq \cup_{n \in \mathbb{N}} \mathcal{Y}^n$. Moreover, the base case $n = d + 1$ has been verified in Brukhim et al. [2022, Lemma 13]. Consequently, it suffices to extend the induction step for concept classes without a pivot. With Lemma 3.6, it is natural to consider modifying the concept class to create a pivot for it while at the same time preserving the DS dimension of the modified class nonincreasing. The technique used here is similar to shifting [Haussler, 1995, Brukhim et al., 2022], though we do not shift the whole edge "downwards" but only shift the last label in some vertex of the edge to a candidate pivot. The difference is necessary, as it has already been shown that the DS dimension of a concept class can increase after the standard shifting [Brukhim et al., 2022, Example 19]. Thus, we name the technique used here "pivot shifting".

**Definition 3.7** (Pivot shifting). *For any $n \in \mathbb{N}\setminus\{1\}$, $a \in \mathcal{Y}$, and $V_n \subseteq \mathcal{Y}^n$ with $|V_n| < \infty$, we define*

$$\mathfrak{P}_a(V_n) := \cup_{y \in \mathcal{Y}} \left\{ (y_1, \ldots, y_{n-1}) \in \mathcal{Y}^{n-1} : (y_1, \ldots, y_{n-1}, y) \in V_n, (y_1, \ldots, y_{n-1}, a) \notin V_n \right\}.$$

*For any $\mathbf{y} = (y_1, \ldots, y_{n-1}) \in \mathfrak{P}_a(V_n)$ and the edge $(e_{n,\mathbf{y}}, n)$ in $\mathcal{G}(V_n)$, we define the set*

$$L_{\mathbf{y}} := \{y \in \mathcal{Y} : (y_1, \ldots, y_{n-1}, y) \in (e_{n,\mathbf{y}}, n)\}.$$

*A mapping $\gamma : \mathfrak{P}_a(V_n) \to \mathcal{Y}$ is called a **pivot shifting** on $V_n$ to $a$ if $\gamma(\mathbf{y}) \in L_{\mathbf{y}}$ for all $\mathbf{y} \in \mathfrak{P}_a(V_n)$. Let $\Gamma_{a,V_n}$ denote the set of all pivot shifting on $V_n$ to $a$. For any $\gamma \in \Gamma_{a,V_n}$, we define*

$$V_n^\gamma := (V_n \setminus \{(\mathbf{y}, \gamma(\mathbf{y})) : \mathbf{y} \in \mathfrak{P}_a(V_n)\}) \cup \{(\mathbf{y}, a) : \mathbf{y} \in \mathfrak{P}_a(V_n)\};$$

*i.e., $V_{n,\gamma}$ is obtained by replacing the label $\gamma(\mathbf{y})$ in $(\mathbf{y}, \gamma(\mathbf{y}))$ with $a$ for all $\mathbf{y} \in \mathfrak{P}_a(V_n)$.*

We prove that the average out-degree does not decrease after pivot shiftings in the following lemma.

**Lemma 3.8.** *For any $a \in \mathcal{Y}$, $V \subseteq \cup_{n=2}^{\infty} \mathcal{Y}^n$ with $|V| < \infty$, and $\gamma \in \Gamma_{a,V}$, we have*

$$\mathsf{avgoutdeg}(\mathcal{G}(V^\gamma)) \geqslant \mathsf{avgoutdeg}(\mathcal{G}(V)).$$

The proof is presented in Appendix G. A key observation for the proof is that by definition, only edges of sizes greater than one contribute to the average out-degree. However, we are not able to show that the DS dimension does not increase after some pivot shifting, which we leave as an open question. Thus, whether pivot shifting is applicable to upper bounding density with DS dimension remains open.

**Open Question 2.** *For any $d \in \mathbb{N}$ and any $V \subseteq \cup_{n=d+2}^{\infty} \mathcal{Y}^n$ with $|V| < \infty$ and $\dim(V) = d$, are there some $a \in \mathcal{Y}$ and $\gamma \in \Gamma_{a,V}$ such that $\dim(V^\gamma) \leqslant d$?*

Nevertheless, we have taken a further and specific step toward the verification of the conjecture that $\mu_{\mathcal{H}} \leqslant 2\dim(\mathcal{H})$: a positive resolution of the above question would lead to the conclusion that $\mu_{\mathcal{H}} \leqslant 2\dim(\mathcal{H})$ by Lemma 3.6, Lemma 3.8, and Brukhim et al. [2022, Lemma 13].

## Acknowledgments and Disclosure of Funding

Shay Moran is a Robert J. Shillman Fellow; he acknowledges support by ISF grant 1225/20, by BSF grant 2018385, by Israel PBC-VATAT, by the Technion Center for Machine Learning and Intelligent Systems (MLIS), and by the the European Union (ERC, GENERALIZATION, 101039692). Views and opinions expressed are however those of the author(s) only and do not necessarily reflect those of the European Union or the European Research Council Executive Agency. Neither the European Union nor the granting authority can be held responsible for them.

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

# A  Lower bound

Before proving Theorem 2.5, we first introduce two lemmas regarding $k$-pseudo-cubes that will be used in the proof.

**Lemma A.1.** *For any positive integers $k, d$, any label class $\mathcal{Y}$, any $k$-pseudo-cube $H \subseteq \mathcal{Y}^d$ of dimension $d$, any $j \in [d]$, and any label $y \in \mathcal{Y}$, define $H_y^j := \{h \in H : h(j) = y\}$. Then, we have*

$$|H_y^j| \leqslant \frac{|H|}{k+1}.$$

*Proof.* When $d = 1$, the result follows trivially from the definition of $k$-pseudo-cubes. We prove the result for $d \geqslant 2$ by contradiction. Suppose on the contrary that there exist some $j \in [d]$ and $y \in \mathcal{Y}$ such that $|H_y^j| > \frac{|H|}{k+1}$. The definition of pseudo-cubes implies that $|H| \geqslant k + 1$. Then, there exist $h, h' \in H_y^j$ with $h \neq h'$. Let $\{f_1, \ldots, f_k\}$ and $\{f_1', \ldots, f_k'\}$ denote the set of $j$-neighbors of $h$ and $h'$ in $H$ respectively. Since $h \neq h'$ and $h(j) = y = h'(j)$, there exists some $j' \in [d]\backslash\{j\}$ such that $h(j') \neq h'(j')$. It follows that $f_i(j') = h(j') \neq h'(j') = f_l'(j')$ and thus $f_i \neq f_l'$ for any $i, l \in [k]$. Then, we have

$$|\{h \in H : h(j) \neq y\}| \geqslant k|H_y^j| > \frac{k|H|}{k+1}$$

and

$$|H| = |\{h \in H : h(j) \neq y\}| + |H_y^j| > |H|,$$

which is a contradiction. Thus, we must have $|H_y^j| \leqslant \frac{|H|}{k+1}$. $\qquad\square$

**Lemma A.2.** *For any integer $k \geqslant 1$, $d \geqslant 2$, $n \in [d-1]$, and $1 \leqslant j_1 < \cdots < j_n \leqslant d$, any label class $\mathcal{Y}$, any $k$-pseudo-cube $H \subseteq \mathcal{Y}^d$ of dimension $d$, and any hypothesis $g \in H$, define $\mathsf{J} := (j_1, \ldots, j_n)$ and $\mathsf{K} = (k_1, \ldots, k_{d-n})$ such that $1 \leqslant k_1 < \cdots < k_{d-n} \leqslant d$ and $\{j_1, \ldots, j_n, k_1, \ldots, k_{d-n}\} = [d]$. Then, $H_{g, \mathsf{J}} := \{h|_{\mathsf{K}} : h \in H, h(j_i) = g(j_i), \forall i \in [n]\}$ is a $k$-pseudo-cube of dimension $d - n$.*

*Proof.* For any $f \in H_{g,J}$, there exists some $h \in H$ such that $f = h|_{\mathsf{K}}$. Then, for any $i \in [d-n]$, there exists $k$ distinct $h_1, \ldots, h_k \in H$ such that $h_m(k_i) \neq h(k_i)$ and $h_m(l) = h(l)$ for all $l \in [d]\backslash\{k_i\}$ and $m \in [k]$. Since $k_i \notin \{j_1, \ldots, j_n\}$, we have $h|_{\mathsf{J}} = h_m|_{\mathsf{J}} = g|_{\mathsf{J}}$ and thus $f_m := h_m|_{\mathsf{K}} \in H_{g,\mathsf{J}}$ for all $m \in [k]$. Then, we have $f(i) = h(k_i) \neq h_m(k_i) = f_m(i)$ and $f(l) = h(k_l) = h_m(k_l) = f_m(l)$ for any $l \in [d-n]\backslash\{i\}$ and $m \in [k]$, which implies that $H_{g,\mathsf{J}}$ is a $k$-pseudo-cube of dimension $d-n$. $\quad\square$

*Proof of Theorem 2.5.* Consider an arbitrary $k$-nondegenerate concept class $\mathcal{H} \subseteq \mathcal{Y}^{\mathcal{X}}$ for some $k \in \mathbb{N}$. Let $x_0, x_1 \in \mathcal{X}$ and $h_1, \ldots, h_{k+1} \in \mathcal{H}$ witness the $k$-nondegeneracy as specified in Definition 2.4. For any $\varepsilon \in (0, 1/2(k+1))$ and $\delta \in (0, 1)$, let $I \sim \mathrm{Bernoulli}((k+1)\varepsilon)$ and $J \sim \mathrm{Unif}([k+1])$. Then, for any $j \in [k+1]$, $(x_I, h_j(x_I))$ follows the $\mathcal{H}$-realizable distribution $P_{\varepsilon,j}$ over $\mathcal{X} \times \mathcal{Y}$ defined by $P_{\varepsilon,j}(\{(x_0, h_j(x_0))\}) = 1 - (k+1)\varepsilon$ and $P_{\varepsilon,j}(\{(x_1, h_j(x_1))\}) = (k+1)\varepsilon$. Sample $(I_1, \ldots, I_n) \sim \mathrm{Bernoulli}(\varepsilon)^n$ independent of $(I, J)$ and define $S = ((x_{I_1}, h_J(x_{I_1})), \ldots, (x_{I_n}, h_J(x_{I_n})))$ and $S' = ((x_0, h_J(x_0)), \ldots, (x_0, h_J(x_0)))$. Then, for any $k$-list learner $\mathcal{A}$, we have

$$\mathbb{P}(h_J(x_I) \notin \mathcal{A}(S, \mathcal{H})(x_I)|I_1 = 0, \ldots, I_n = 0)$$
$$\geqslant \mathbb{P}(h_J(x_1) \notin \mathcal{A}(S', \mathcal{H})(x_1), I = 1)$$
$$= \mathbb{P}(I = 1)\mathbb{P}(h_J(x_1) \notin \mathcal{A}(S', \mathcal{H})(x_1))$$
$$= (k+1)\varepsilon\mathbb{E}[\mathbb{P}(h_J(x_1) \notin \mathcal{A}(S', \mathcal{H})(x_1)|\mathcal{A}(S', \mathcal{H}))]$$
$$\geqslant \varepsilon,$$

where the last inequality follows from the facts that $h_J(x_0) = h_1(x_0)$, $|\{h_1(x_1), \ldots, h_{k+1}(x_1)\}| = k + 1$, $h_J(x_1) \sim \mathrm{Unif}(\{h_1(x_1), \ldots, h_{k+1}(x_1)\})$. Since $J \sim \mathrm{Unif}([k+1])$, there exists $j \in [k+1]$ such that

$$\mathbb{P}(h_j(x_I) \notin \mathcal{A}(S^j, \mathcal{H})(x_I)|I_1 = 0, \ldots, I_n = 0)$$
$$= \mathbb{P}(h_J(x_I) \notin \mathcal{A}(S, \mathcal{H})(x_I)|J = j, I_1 = 0, \ldots, I_n = 0) \geqslant \varepsilon,$$

where $S^j := ((x_{I_1}, h_j(x_{I_1})), \ldots, (x_{I_n}, h_j(x_{I_n}))) \sim P_{\varepsilon,j}^n$ is independent of $(x_I, h_j(x_I)) \sim P_{\varepsilon,j}$. Since $\frac{1}{2(k+1)\varepsilon} \leqslant \frac{1-(k+1)\varepsilon}{(k+1)\varepsilon} < \frac{1}{-\log(1-(k+1)\varepsilon)}$ for $\varepsilon \in (0, 1/2(k+1))$, if $n \leqslant \frac{\log(1/\delta)}{2(k+1)\varepsilon} < \frac{\log(\delta)}{\log(1-(k+1)\varepsilon)}$, we have

$$\mathbb{P}(I_1 = 0, \ldots, I_n = 0) = (1 - (k+1)\varepsilon)^n > \delta.$$

Then, with probability greater than $\delta$, we have $\mathrm{er}_{P_{\varepsilon,j}}(\mathcal{A}(S^j, \mathcal{H})) \geqslant \varepsilon$, which implies that

$$\mathcal{M}_{\mathcal{H}}^k(\varepsilon, \delta) \geqslant \frac{\log(1/\delta)}{2(k+1)\varepsilon}.$$

Next, consider an arbitrary concept class $\mathcal{H} \subseteq \mathcal{Y}^{\mathcal{X}}$ such that $\dim_k(\mathcal{H}) = d_k \in \mathbb{N}\backslash\{1\}$ for some $k \in \mathbb{N}$. Then, there exist a sequence $\mathbf{x} := (x_1, \ldots, x_{d_k}) \in \mathcal{X}^{d_k}$ and a $k$-pseudo-cube $H \subseteq \mathcal{H}|_{\mathbf{x}}$ of dimension $d_k$. Denote the elements in $H$ with $\mathbf{y}^1 = (y_1^1, \ldots, y_{d_k}^1), \ldots, \mathbf{y}^{|H|} = (y_1^{|H|}, \ldots, y_{d_k}^{|H|})$. For any $\varepsilon \in (0, 1/8(k+1))$, consider the categorical distribution $Q_\varepsilon$ over $[d_k]$ defined by $Q_\varepsilon(\{1\}) = 1 - 4(k+1)\varepsilon$ and $Q_\varepsilon(\{i\}) = \frac{4(k+1)\varepsilon}{d_k-1}$ for $i \in [d_k]\backslash\{1\}$. Let $J \sim \mathrm{Unif}([|H|])$. For any $i \in [d_k]\backslash\{1\}$ and $\mathcal{I} = (i_1, \ldots, i_m) \in ([d_k]\backslash\{i\})^m$ with $m \in [d_k-1]$ and $i_1 < \cdots < i_m$, define $\mathcal{I}' := (i_1', \ldots, i_{d_k-m}')$ such that $i_1' < \cdots < i_{d_k-m}'$ and $\{i_1, \ldots, i_m, i_1', \ldots, i_{d_k-m}'\} = [d_k]$. Then, we have that conditional on $(i_1, y_{i_1}^J), \ldots, (i_m, y_{i_m}^J)$, $\mathbf{y}^J|_{\mathcal{I}'}$ follows the uniform distribution over the set $H_{\mathbf{y}^J, \mathcal{I}}$ which is a $k$-pseudo-cube by Lemma A.2. Consequently, we can apply Lemma A.1 to conclude that

$$|(H_{\mathbf{y}^J, \mathcal{I}})_y^{i'}| \leqslant \frac{|H_{\mathbf{y}^J, \mathcal{I}}|}{k+1}$$

for any $y \in \mathcal{Y}$ and $i' \in [d_k - m]$, which immediately implies that

$$\mathbb{P}(y_i^J \in \{v_1, \ldots, v_k\}|(i_1, y_{i_1}^J), \ldots, (i_m, y_{i_m}^J)) \leqslant \frac{k}{k+1}. \tag{11}$$

for any distinct $v_1, \ldots, v_k \in \mathcal{Y}$.

Let $(I, I_1, \ldots, I_n) \sim Q_\varepsilon^{n+1}$ for $n \in \mathbb{N}$ be independent of $J$. Define $S := ((x_{I_1}, y_{I_1}^J), \ldots, (x_{I_n}, y_{I_n}^J))$. For any $k$-list leaner $\mathcal{A}$, by (11), we have

$$\mathbb{P}(y_I^J \notin \mathcal{A}(S, \mathcal{H})(x_I), I \neq 1)$$
$$\geqslant \mathbb{P}(y_I^J \notin \mathcal{A}(S, \mathcal{H})(x_I), I \neq 1, I \neq I_1, \ldots, I \neq I_n)$$
$$\geqslant \sum_{i=2}^{d_k} \mathbb{P}(y_I^J \notin \mathcal{A}(S, \mathcal{H})(x_I), I = i, I_1 \neq i, \ldots, I_n \neq i)$$
$$= \sum_{i=2}^{d_k} \mathbb{E}[\mathbb{1}_{I_1 \neq i, \ldots, I_n \neq i} \mathbb{P}(y_i^J \notin \mathcal{A}(S, \mathcal{H})(x_i), I = i | \mathcal{A}(S, \mathcal{H}), (I_1, y_{I_1}^J), \ldots, (I_n, y_{I_n}^J))]$$
$$\geqslant \sum_{i=2}^{d_k} \frac{\mathbb{P}(I = i)}{k+1} \mathbb{P}(I_1 \neq i, \ldots, I_n \neq i)$$
$$= 4\varepsilon \left(1 - \frac{4(k+1)\varepsilon}{d_k-1}\right)^n.$$

Since $J \sim \mathrm{Unif}([|H|])$, there exists $j = j(\mathcal{A}, \mathcal{H}) \in [|H|]$ such that

$$\mathbb{P}(y_I^j \notin \mathcal{A}(S^j, \mathcal{H})(x_I), I \neq 1) = \mathbb{P}(y_I^J \notin \mathcal{A}(S, \mathcal{H})(x_I), I \neq 1 | J = j) \geqslant 4\varepsilon \left(1 - \frac{4(k+1)\varepsilon}{d_k-1}\right)^n,$$

where $S^j := ((x_{I_1}, y_{I_1}^j), \ldots, (x_{I_n}, y_{I_n}^j))$. Note that if we define the distribution $P_{\varepsilon,j}$ over $\mathcal{X} \times \mathcal{Y}$ by

$$P_{\varepsilon,j}(\{x_1, y_1^j\}) = 1 - 4(k+1)\varepsilon \quad \text{and}$$
$$P_{\varepsilon,j}(\{x_i, y_i^j\}) = \frac{4(k+1)\varepsilon}{d_k-1}, \quad \forall i \in [d_k]\backslash\{1\},$$

then, we have $(S, (x_I, y_I^j)) \sim P_{\varepsilon,j}^{n+1}$. For any $n \leqslant \frac{(d_k-1)\log(2)}{8(k+1)\varepsilon} < \frac{\log(1/2)}{\log(1-4(k+1)\varepsilon/(d_k-1))}$, we have $\left(1 - \frac{4(k+1)\varepsilon}{d_k-1}\right)^n > \frac{1}{2}$ and

$$\mathbb{P}(y_I^j \notin \mathcal{A}(S^j, \mathcal{H})(x_I), I \neq 1) > 2\varepsilon.$$

Now, we define the algorithm $\mathcal{A}'$ by

$$\mathcal{A}'(\mathbf{s}, \mathcal{H})(x_1) := \{y_1^j\} \text{ and } \mathcal{A}'(\mathbf{s}, \mathcal{H})(x) := \mathcal{A}(\mathbf{s}, \mathcal{H})(x), \ \forall x \in \mathcal{X}\backslash\{x_1\}$$

for any $\mathbf{s} \in \cup_{m=0}^{\infty}(\mathcal{X} \times \mathcal{Y})^m$. Then, we have

$$
\begin{aligned}
\mathbb{E}[\mathrm{er}_{P_{\varepsilon,j}}(\mathcal{A}'(S^j, \mathcal{H}))] &\geqslant \mathbb{P}(y_I^j \notin \mathcal{A}'(S^j, \mathcal{H})(x_I), I \neq 1) \\
&= \mathbb{P}(y_I^j \notin \mathcal{A}(S^j, \mathcal{H})(x_I), I \neq 1) \\
&> 2\varepsilon.
\end{aligned}
\tag{12}
$$

On the other hand, the definition of $\mathcal{A}'$ yields that $y_1^j \in \mathcal{A}'(S^j, (x_1))$ and hence

$$\mathrm{er}_{P_{\varepsilon,j}}(\mathcal{A}'(S^j, \mathcal{H})) \leqslant P_{\varepsilon,j}(\mathcal{X}\backslash\{x_1\}) = 4(k+1)\varepsilon.$$

Suppose the following holds

$$\mathbb{P}(\mathrm{er}_{P_{\varepsilon,j}}(\mathcal{A}(S^j, \mathcal{H})) > \varepsilon) \leqslant \frac{1}{4(k+1)}.\tag{13}$$

Since $\mathrm{er}_{P_{\varepsilon,j}}(\mathcal{A}'(S^j, \mathcal{H})) \leqslant \mathrm{er}_{P_{\varepsilon,j}}(\mathcal{A}(S^j, \mathcal{H}))$, the above inequality implies that

$$\mathbb{P}(\mathrm{er}_{P_{\varepsilon,j}}(\mathcal{A}'(S^j, \mathcal{H})) > \varepsilon) \leqslant \frac{1}{4(k+1)}$$

and therefore

$$
\begin{aligned}
\mathbb{E}[\mathrm{er}_{P_{\varepsilon,j}}(\mathcal{A}'(S^j, \mathcal{H}))] &\leqslant \varepsilon + \mathbb{E}\left[\mathrm{er}_{P_{\varepsilon,j}}(\mathcal{A}'(S^j, \mathcal{H})) \mathbb{1}_{\mathrm{er}_{P_{\varepsilon,j}}(\mathcal{A}'(S^j, \mathcal{H})) > \varepsilon}\right] \\
&\leqslant \varepsilon + 4(k+1)\varepsilon \mathbb{P}(\mathrm{er}_{P_{\varepsilon,j}}(\mathcal{A}'(S^j, \mathcal{H}) > \varepsilon) \\
&\leqslant 2\varepsilon,
\end{aligned}
$$

which contradicts (12). Thus, we can conclude that (13) is false, i.e.,

$$\mathbb{P}(\mathrm{er}_{P_{\varepsilon,j}}(\mathcal{A}(S^j, \mathcal{H})) > \varepsilon) > \frac{1}{4(k+1)}$$

for $n \leqslant \frac{(d_k-1)\log(2)}{8(k+1)\varepsilon}$. For $\delta \in (0, 1/4(k+1)]$, we have $\mathbb{P}(\mathrm{er}_{P_{\varepsilon,j}}(\mathcal{A}(S^j, \mathcal{H})) > \varepsilon) > \delta$ and thus

$$\mathcal{M}_{\mathcal{H}}^k(\varepsilon, \delta) \geqslant \frac{(d_k-1)\log(2)}{8(k+1)\varepsilon}.$$

In conclusion, for $\varepsilon, \delta = O(1/k)$, we have

$$\mathcal{M}_{\mathcal{H}}^k(\varepsilon, \delta) = \Omega\left(\frac{d_k + \log(1/\delta)}{k\varepsilon}\right).$$

$\square$

## B Proof of Theorem 2.6

*Proof.* Consider an algorithm $\mathcal{A} : (\cup_{k=1}^{\infty}(\mathcal{X} \times \mathcal{Y})^k) \times 2^{\mathcal{Y}^{\mathcal{X}}} \to \mathcal{Y}^{\mathcal{X}}$ which for any hypothesis class $\mathcal{H} \subseteq \mathcal{Y}^{\mathcal{X}}$, $\mathcal{H}$-realizable distribution $\mathcal{D}$, and $n \in \mathbb{N}$, satisfies that

$$\mathbb{P}_{(T,(X,Y)) \sim \mathcal{D}^{n+1}}(\mathcal{A}(T, \mathcal{H})(X) \neq Y) \leqslant \frac{M_n}{n}\tag{14}$$

where $M_n$ is nondecreasing in the sample size $n$.

For an arbitrary sequence $S = ((x_1, y_1), \ldots, (x_n, y_n), (x_{n+1}, y_{n+1})) \in (\mathcal{X} \times \mathcal{Y})^{n+1}$, define $S_{-i}$ to be the subsequence of $S$ excluding $(x_i, y_i)$ for any $i \in [n+1]$. Let $(T_m, (X, Y)) \sim \mathrm{Unif}(S)^{m+1}$ where $m \in \mathbb{N}$, $T_m \in (\mathcal{X} \times \mathcal{Y})^m$ and $(X, Y) \in \mathcal{X} \times \mathcal{Y}$. Moreover, for any $i \in [n+1]$, let $T_m^i \sim \mathrm{Unif}(S_{-i})^m$. Then, for any algorithm $\mathcal{A} : \cup_{k=1}^\infty (\mathcal{X} \times \mathcal{Y})^k \times 2^{\mathcal{Y}^{\mathcal{X}}} \to \mathcal{Y}^{\mathcal{X}}$ and concept class $\mathcal{H} \subseteq \mathcal{Y}^{\mathcal{X}}$, we have

$$
\begin{aligned}
\mathbb{P}(\mathcal{A}(T_m, \mathcal{H})(X) \neq Y) &= \frac{1}{n+1} \sum_{i=1}^{n+1} \mathbb{P}(\mathcal{A}(T_m, \mathcal{H})(x_i) \neq y_i) \\
&\geqslant \frac{1}{n+1} \sum_{i=1}^{n+1} \mathbb{E}\left[\mathbb{1}_{(x_i, y_i) \notin T_m} \mathbb{1}_{\mathcal{A}(T_m, \mathcal{H})(x_i) \neq y_i}\right] \\
&= \frac{1}{n+1} \sum_{i=1}^{n+1} \mathbb{P}((x_i, y_i) \notin T_m) \mathbb{P}(\mathcal{A}(T_m, \mathcal{H})(x_i) \neq y_i | (x_i, y_i) \notin T_m) \\
&\geqslant \frac{n+1-m}{n+1} \frac{1}{n+1} \sum_{i=1}^{n+1} \mathbb{P}(\mathcal{A}(T_m, \mathcal{H})(x_i) \neq y_i | (x_i, y_i) \notin T_m) \\
&= \frac{n+1-m}{n+1} \frac{1}{n+1} \sum_{i=1}^{n+1} \mathbb{P}(\mathcal{A}(T_m^i, \mathcal{H})(x_i) \neq y_i). \tag{15}
\end{aligned}
$$

Note that if $S$ is consistent with $\mathcal{H}$, then $\mathrm{Unif}(S)$ is $\mathcal{H}$-realizable.

Next, define the algorithm $\mathcal{A}_{\mathrm{Maj},m} : (\cup_{k=1}^\infty (\mathcal{X} \times \mathcal{Y})^k) \times 2^{\mathcal{Y}^{\mathcal{X}}} \to \mathcal{Y}^{\mathcal{X}}$ by

$$
\mathcal{A}_{\mathrm{Maj},m}(R, \mathcal{H}) := \mathrm{Majority}((\mathcal{A}(\mathbf{r}, \mathcal{H}))_{\mathbf{r} \in R^m}), \quad \forall R \in (\mathcal{X} \times \mathcal{Y})^n, \ \forall n \in \mathbb{N}.
$$

By the definition above, for any permutation $R_\pi$ of $R \in (\mathcal{X} \times \mathcal{Y})^n$, we have

$$
\mathcal{A}_{\mathrm{Maj},m}(R_\pi, \mathcal{H}) = \mathcal{A}_{\mathrm{Maj},m}(R, \mathcal{H}). \tag{16}
$$

Moreover, for any $i \in [n+1]$, the above definition yields that

$$
\begin{aligned}
\mathbb{1}_{\mathcal{A}_{\mathrm{Maj},m}(S_{-i}, \mathcal{H})(x_i) \neq y_i} &\leqslant \mathbb{1}_{\frac{1}{n^m}|\{\mathbf{s} \in S_{-i}^m : \mathcal{A}(\mathbf{s}, \mathcal{H})(x_i) \neq y_i\}| \geqslant \frac{1}{2}} \\
&\leqslant \frac{2}{n^m} |\{\mathbf{s} \in S_{-i}^m : \mathcal{A}(\mathbf{s}, \mathcal{H})(x_i) \neq y_i\}| \\
&= 2\mathbb{P}(\mathcal{A}(T_m^i, \mathcal{H})(x_i) \neq y_i).
\end{aligned}
$$

Then, by (15), we have

$$
\frac{1}{n+1} \sum_{i=1}^{n+1} \mathbb{1}_{\mathcal{A}_{\mathrm{Maj},m}(S_{-i}, \mathcal{H})(x_i) \neq y_i} \leqslant \frac{2(n+1)}{n+1-m} \mathbb{P}(\mathcal{A}(T_m.\mathcal{H})(X) \neq Y).
$$

Choosing $m = \lfloor (n+1)/2 \rfloor$ and defining $\mathcal{A}_{\mathrm{Maj}} := \mathcal{A}_{\mathrm{Maj},\lfloor (n+1)/2 \rfloor}$, for any $\mathcal{H}$-realizable sequence $S$, by (14) and the above results, we have

$$
\sum_{i=1}^{n+1} \mathbb{1}_{\mathcal{A}_{\mathrm{Maj}}(S_{-i}, \mathcal{H})(x_i) \neq y_i} \leqslant \frac{2(n+1)^2 M_{\lfloor (n+1)/2 \rfloor}}{(n+1-\lfloor (n+1)/2 \rfloor) \lfloor (n+1)/2 \rfloor} \leqslant 8.34 M_{\lfloor (n+1)/2 \rfloor} \tag{17}
$$

for any $n \geqslant 4$. (16) and (17) imply that the algorithm $\mathcal{A}_{\mathrm{Maj}}$ satisfies Assumptions 2.1 and 2.3 in Aden-Ali et al. [2023].

Now, we can define the randomized algorithm $\mathcal{A}_{\mathrm{Ran}}$ which given a sample $S \in (\mathcal{X} \times \mathcal{Y})^n$ with $n \in \mathbb{N}$ and a concept class $\mathcal{H}$, outputs the classifier $\mathcal{A}_{\mathrm{Maj}}(S, \mathcal{H})$ if $n \leqslant 3$ and outputs a random classifier following the uniform distribution over the sequence $(\mathcal{A}_{\mathrm{Maj}}(S_{\leqslant t}, \mathcal{H}))_{\lfloor n/4 \rfloor \leqslant t \leqslant 4\lfloor n/4 \rfloor - 1}$ if $n \geqslant 4$, i.e.,

$$
\mathbb{P}(\mathcal{A}_{\mathrm{Ran}}(S, \mathcal{H}) = \mathcal{A}_{\mathrm{Maj}}(S_{\leqslant t}, \mathcal{H})) = \frac{1}{3\lfloor n/4 \rfloor}, \quad \forall t \in [\lfloor n/4 \rfloor, 4\lfloor n/4 \rfloor - 1],
$$

where $S_{\leqslant t}$ denotes the subsequence of $S$ consisting of the first $t$ elements in $S$. Then, by Aden-Ali et al. [2023, Theorem 2.1], for any $n \geqslant 4$, $\mathcal{H}$-realizable distribution $\mathcal{D}$, and confidence parameter

$\delta \in (0, 1)$, given a training sample $S \sim \mathcal{D}^n$, we have

$$\mathrm{er}_{\mathcal{D}}(\mathcal{A}_{\mathrm{Ran}}(S, \mathcal{H})) = \frac{1}{3\lfloor n/4 \rfloor} \sum_{t=\lfloor n/4 \rfloor}^{4\lfloor n/4 \rfloor - 1} \mathrm{er}_{\mathcal{D}}(\mathcal{A}_{\mathrm{Maj}}(S_{\leqslant t}, \mathcal{H}))$$

$$\leqslant 4.82 \left( \frac{8.34 M_{\lfloor n/2 \rfloor}}{n} + \frac{1}{n} \log\left(\frac{2}{\delta}\right) \right)$$

with probability at least $1 - \delta$. $\qquad\qquad\square$

## C  Proof of Theorem 2.7

The following proof requires several lemmas from Appendix H.

*Proof of Theorem 2.7.* Assume that we have access to a list learning algorithm $\mathcal{A}_{\mathrm{list}}$ which for any concept class $\mathcal{H} \subseteq \mathcal{Y}^{\mathcal{X}}$ of DS dimension $d$, any $\mathcal{H}$-realizable distribution $\mathcal{D}$, any $n \in \mathbb{N}$, and $(S, (X, Y)) \sim \mathcal{D}^{n+1}$, outputs a menu $\mathcal{A}_{\mathrm{list}}(S, \mathcal{H})$ of size $p(n) \in \mathbb{N}$ satisfying

$$\mathbb{P}(Y \notin \mathcal{A}_{\mathrm{list}}(S, \mathcal{H})(X)) \leqslant \frac{\beta(\mathcal{H}, n)}{n}$$

for some function $\beta : 2^{\mathcal{Y}^{\mathcal{X}}} \times \mathbb{N} \to [0, \infty)$.

For $n_1, n_2 \in \mathbb{N}$, let $(S^1, S^2, (X, Y)) \sim \mathcal{D}^{n_1 + 2n_2 + 1}$ where $S^1 \in (\mathcal{X} \times \mathcal{Y})^{n_1}$, $S^2 \in (\mathcal{X} \times \mathcal{Y})^{2n_2}$, and $(X, Y) \in \mathcal{X} \times \mathcal{Y}$. Let $\widehat{\mu} = \mathcal{A}_{\mathrm{list}}(S^1, \mathcal{H})$. According to the property of $\mathcal{A}_{\mathrm{list}}$, the size of $\widehat{\mu}$ is $p(n_1)$ and we have

$$\mathbb{P}(Y \notin \widehat{\mu}(X)) \leqslant \frac{\beta(\mathcal{H}, n_1)}{n_1}.$$

For notational convenience, define $S' := (S^2, (X, Y))$, $S'' := (S^1, S^2, (X, Y))$, and enumerate the elements of $S'$ as

$$((X_1, Y_1), \ldots, (X_{2n_2+1}, Y_{2n_2+1}))$$

where $(X_{2n_2+1}, Y_{2n_2+1})$ denotes $(X, Y)$. We now define

$$N := \sum_{i \in [2n_2]} \mathbb{1}_{Y_i \notin \widehat{\mu}(X_i)}$$

and

$$\mathcal{H}_{S'} := \mathcal{H}_{(X_1, \ldots, X_{2n_2+1}), \widehat{\mu}, N}$$
$$= \left\{ h|_{(X_1, \ldots, X_{2n_2+1})} : h \in \mathcal{H}, |\{ i \in [2n_2 + 1] : h(X_i) \notin \widehat{\mu}(X_i) \}| \leqslant N + 1 \right\}.$$

It follows from the property of $\widehat{\mu}$ and Lemma H.9 that

$$\mathbb{E}[N] = \frac{2n_2 \beta(\mathcal{H}, n_1)}{n_1} \tag{18}$$

and, conditional on $S'$,

$$\mathrm{dim}_G(\mathcal{H}_{S'}) \leqslant (2 \log_2 e + 4)(5d \log_2(p(n_1)) + 2N + 2).$$

Sample $(\mathbf{I}, I) \sim \mathrm{Unif}([2n_2 + 1])^{n_2 + 1}$ independent of $S''$ where $\mathbf{I} = (I_1, \ldots, I_{n_2}) \in [2n_2 + 1]^{n_2}$ and $I \in [2n_2 + 1]$. Then, we define the sequence $T(S', \mathbf{I}) := ((I_1, Y_{I_1}), \ldots, (I_{n_2}, Y_{I_{n_2}}))$ and the classifier

$$\widehat{h}_{S', \mathbf{I}} := A_G\left( T(S', \mathbf{I}), \mathcal{H}_{S'} \right)$$

for any $i \in [2n_2 + 1]$, where $A_G$ is the algorithm specified in Proposition H.5. Since conditional on $S''$, the distribution of each element in $T(S', \mathbf{I})$ is $\mathcal{H}_{S'}$-realizable, by Corollary H.6, there exists some constant $C' > 0$ such that

$$\mathbb{P}(\widehat{h}_{S', \mathbf{I}}(I) \neq Y_I | S'') \leqslant \frac{C' \mathrm{dim}_G(\mathcal{H}_{S'})}{n_2} \leqslant \frac{C_1 d \log(p(n_1)) + C_2 N + C_3}{n_2}. \tag{19}$$

for some constant $C_1, C_2, C_3 > 0$.

For $\mathbf{I}' = (I_1', \ldots, I_{n_2}') \sim \text{Unif}([2n_2])^{n_2}$ independent of all other random variables, define $T^i(S', \mathbf{I}') := ((\rho_i(I_1'), Y_{\rho_i(I_1')}), \ldots, (\rho_i(I_{n_2}'), Y_{\rho_i(I_{n_2}')}))$ where $\rho_i : [2n_2] \to [2n_2 + 1]\backslash\{i\}$, $k \mapsto k\mathbb{1}_{k<i} + (k+1)\mathbb{1}_{k \geqslant i}$ for all $i \in [2n_2 + 1]$. Consider the classifier

$$\widetilde{h}_{S', \mathbf{I}', i} := A_G(T^i(S', \mathbf{I}'), \mathcal{H}_{S'})$$

for each $i \in [2n_2 + 1]$. Since $Y_i$ is not used in the construction of $\widetilde{h}_{S', \mathbf{I}', i}$, we can also denote $\widetilde{h}_{S', \mathbf{I}', i}$ as $\widetilde{h}_{S'_{-i}, X_i, \mathbf{I}', i}$, where $S'_{-i} := ((X_1, Y_1), \ldots, (X_{i-1}, Y_{i-1}), (X_{i+1}, Y_{i+1}), \ldots, (X_{2n_2+1}, Y_{2n_2+1}))$. Thus, treating $S := (S_1, S_2) \sim \mathcal{D}^{n_1+2n_2}$ as the training sample, we can define the classifier $\widehat{h}_S \in \mathcal{Y}^\mathcal{X}$ by

$$\widehat{h}_S(x) := \widetilde{h}_{S^2, x, \mathbf{I}', 2n_2+1}(2n_2 + 1),$$

where we emphasize that the RHS depends on $S^1$ through the construction of $\widehat{\mu}$. Then, recalling that $(X, Y) \sim \mathcal{D}$ is independent of $S$, our task is to upper bound the expected error rate of $\widehat{h}_S$:

$$\mathbb{P}\left(\widehat{h}_S(X) \neq Y\right).$$

We first relate $\widehat{h}_{S', \mathbf{I}}$ to $\widetilde{h}_{S', \mathbf{I}', i}$ for $i \in [2n_2 + 1]$:

$$\mathbb{P}(\widehat{h}_{S', \mathbf{I}}(I) \neq Y_I | S'') \geqslant \frac{1}{2n_2 + 1} \sum_{i=1}^{2n_2+1} \mathbb{E}\left[\mathbb{1}_{i \notin \mathbf{I}} \mathbb{1}_{\widehat{h}_{S', \mathbf{I}}(i) \neq Y_i} | S''\right]$$

$$= \frac{1}{2n_2 + 1} \sum_{i=1}^{2n_2+1} \mathbb{P}(i \notin \mathbf{I}) \mathbb{E}\left[\mathbb{1}_{\widehat{h}_{S', \mathbf{I}}(i) \neq Y_i} | S'', i \notin \mathbf{I}\right]$$

$$\geqslant \frac{1}{2(2n_2 + 1)} \sum_{i=1}^{2n_2+1} \mathbb{E}\left[\mathbb{1}_{\widetilde{h}_{S', \mathbf{I}', i}(i) \neq Y_i} | S''\right],$$

which implies that

$$\mathbb{P}(\widehat{h}_{S', \mathbf{I}}(I) \neq Y_I | S^1) \geqslant \frac{1}{2(2n_2 + 1)} \sum_{i=1}^{2n_2+1} \mathbb{E}\left[\mathbb{1}_{\widetilde{h}_{S', \mathbf{I}', i}(i) \neq Y_i} | S^1\right]$$

$$= \frac{1}{2(2n_2 + 1)} \sum_{i=1}^{2n_2+1} \mathbb{E}\left[\mathbb{E}\left[\mathbb{1}_{\widetilde{h}_{S', \mathbf{I}', i}(i) \neq Y_i} | S^1, \mathbf{I}'\right] \Big| S^1\right].$$

Conditional on $\mathbf{I}'$ and $S^1$, for any $i \in [2n_2 + 1]$ and any sequence $\mathbf{s} \in (\mathcal{X} \times \mathcal{Y})^{2n_2+1}$, we let $\widetilde{h}_{\mathbf{s}, \mathbf{I}', i}$ denote the classifier when replacing $S'$ with $\mathbf{s}$ in $\widetilde{h}_{S', \mathbf{I}', i}$, i.e., $\widetilde{h}_{\mathbf{s}, \mathbf{I}', i} = A_G(T^i(\mathbf{s}, \mathbf{I}'), \mathcal{H}_\mathbf{s})$. For any $i \in [2n_2 + 1]$, define the set

$$B_i := \left\{\mathbf{s} = ((x_1, y_1), \ldots, (x_{2n_2+1}, y_{2n_2+1})) \in (\mathcal{X} \times \mathcal{Y})^{2n_2+1} : \widetilde{h}_{\mathbf{s}, \mathbf{I}', i}(i) \neq y_i\right\}$$

and the permutation

$$\pi^i : (\mathcal{X} \times \mathcal{Y})^{2n_2+1} \to (\mathcal{X} \times \mathcal{Y})^{2n_2+1}, \quad (z_1, \ldots, z_{2n_2+1}) \mapsto (z_1, \ldots, z_{i-1}, z_{2n_2+1}, z_i, \ldots, z_{2n_2}).$$

We also define the set

$$B := \Big\{\mathbf{s} = ((x_1, y_1), \ldots, (x_{2n_2+1}, y_{2n_2+1})) \in (\mathcal{X} \times \mathcal{Y})^{2n_2+1} :$$

$$y_{2n_2} \in \widehat{\mu}(x_{2n_2}), \ y_{2n_2+1} \in \widehat{\mu}(x_{2n_2+1}), \text{ and } \widetilde{h}_{\mathbf{s}, \mathbf{I}', 2n_2+1}(2n_2 + 1) \neq y_{2n_2+1}\Big\}.$$

We would like to show that $\pi^i(B) \subseteq B_i$ for all $i \in [2n_2]$. For any $\mathbf{s} = ((x_1, y_1), \ldots, (x_{2n_2+1}, y_{2n_2+1})) \in B$, we let $(x_j^i, y_j^i) \in \mathcal{X} \times \mathcal{Y}$ denote the $j$-th element of $\pi^i(\mathbf{s})$ for each $j \in [2n_2 + 1]$. By the definition of $\pi^i$ and $T^i$, we have

$$T^{2n_2+1}(\mathbf{s}, \mathbf{I}') = ((I_1', y_{I_1'}), \ldots, (I_{n_2}', y_{I_{n_2}'})) \text{ and}$$

$$T^i(\pi^i(\mathbf{s}), \mathbf{I}') = ((\rho_i(I_1'), y_{I_1'}), \ldots, (\rho_i(I_{n_2}'), y_{I_{n_2}'})).$$

Define
$$n_{\mathbf{s}} := \sum_{j \in [2n_2]} \mathbb{1}_{y_j \notin \widehat{\mu}(x_j)}.$$

Since $y_{2n_2} \in \widehat{\mu}(x_{2n_2})$ and $y_{2n_2+1} \in \widehat{\mu}(x_{2n_2+1})$, we have
$$n_{\pi^i(\mathbf{s})} := \sum_{j \in [2n_2]} \mathbb{1}_{y_j^i \notin \widehat{\mu}(x_j^i)} = \sum_{j \in [2n_2+1] \setminus \{2n_2\}} \mathbb{1}_{y_j \notin \widehat{\mu}(x_j)} = n_{\mathbf{s}}$$

and therefore,
$$
\begin{aligned}
\mathcal{H}_{\pi^i(\mathbf{s})} &= \mathcal{H}_{(x_1^i, \ldots, x_{2n_2+1}^i), \widehat{\mu}, n_{\pi^i(\mathbf{s})}} \\
&= \mathcal{H}_{(x_1^i, \ldots, x_{2n_2+1}^i), \widehat{\mu}, n_{\mathbf{s}}} \\
&= \left\{ h|_{(x_1, \ldots, x_{i-1}, x_{2n_2+1}, x_i, \ldots, x_{2n_2})} : h \in \mathcal{H}, |\{j \in [2n_2+1] : h(x_j) \notin \widehat{\mu}(x_j)\}| \leqslant n_{\mathbf{s}} + 1 \right\} \\
&= \{\kappa_i(h) : h \in \mathcal{H}_{\mathbf{s}}\}
\end{aligned}
$$

where for any $h \in \mathcal{Y}^{2n_2+1}$, $\kappa_i(h) \in \mathcal{Y}^{2n_2+1}$ is defined by
$$\kappa_i(h)(j) := h(\rho_i^{-1}(j)) \mathbb{1}_{j \neq i} + h(2n_2+1) \mathbb{1}_{j=i}, \quad \forall j \in [2n_2+1],$$

i.e., $h(j) = \kappa_i(h)(\rho_i(i))$ for $j \in [2n_2]$ and $h(2n_2+1) = \kappa_i(h)(i)$. Note that $\kappa_i$ is a bijection from $\mathcal{H}_{\mathbf{s}}$ to $\mathcal{H}_{\pi^i(\mathbf{s})}$. Thus, for any $h \in \mathcal{H}_{\pi^i(\mathbf{s})}$, we have
$$h(\rho_i(I_j')) = \kappa_i^{-1}(h)(I_j'), \ \forall j \in [n_2], \text{ and } h(i) = \kappa_i^{-1}(h)(2n_2+1).$$

Similarly, for any $h \in \mathcal{H}_{\mathbf{s}}$, we have
$$h(I_j') = \kappa_i(h)(\rho_i(I_j')), \ \forall j \in [n_2], \text{ and } h(2n_2+1) = \kappa_i(h)(i).$$

Given the above analysis, we have
$$
\begin{aligned}
\widetilde{h}_{\mathbf{s}, \mathbf{I}', 2n_2+1}(2n_2+1) &= A_G(T^{2n_2+1}(\mathbf{s}, \mathbf{I}'), \mathcal{H}_{\mathbf{s}})(2n_2+1) \\
&= A_G(T^i(\pi^i(\mathbf{s}), \mathbf{I}'), \mathcal{H}_{\pi^i(\mathbf{s})})(i) \\
&= \widetilde{h}_{\pi^i(\mathbf{s}), \mathbf{I}', i}(i),
\end{aligned}
$$

which immediately implies that
$$\widetilde{h}_{\pi^i(\mathbf{s}), \mathbf{I}', i}(i) = \widetilde{h}_{\mathbf{s}, \mathbf{I}', 2n_2+1}(2n_2+1) \neq y_{2n_2+1} = y_i^i$$

and thus $\pi^i(\mathbf{s}) \in B_i$. Since $i \in [2n_2]$ and $\mathbf{s} \in B$ is arbitrary, we have $\pi^i(B) \subseteq B_i$ for any $i \in [2n_2]$. Then, conditional on $\mathbf{I}'$ and $S^1$, we have
$$\mathcal{D}^{2n_2+1}(B_i) \geqslant \mathcal{D}^{2n_2+1}(\pi^i(B)) = \mathcal{D}^{2n_2+1}(B)$$

Since $B \subseteq B_{2n_2+1}$ holds trivially, we have
$$
\begin{aligned}
\mathbb{P}(\widehat{h}_{S', \mathbf{I}}(I) \neq Y_I | S^1) &\geqslant \frac{1}{2(2n_2+1)} \sum_{i=1}^{2n_2+1} \mathbb{E}\left[ \mathbb{E}\left[ \mathbb{1}_{\widetilde{h}_{S', \mathbf{I}', i}(i) \neq Y_i} \big| S^1, \mathbf{I}' \right] \Big| S^1 \right] \\
&\geqslant \frac{1}{2} \mathbb{E}\left[ \mathbb{1}_{Y_{2n_2} \in \widehat{\mu}(X_{2n_2})} \mathbb{1}_{Y_{2n_2+1} \in \widehat{\mu}(X_{2n_2+1})} \mathbb{1}_{\widetilde{h}_{S', \mathbf{I}', 2n_2+1}(2n_2+1) \neq Y_{2n_2+1}} \Big| S^1 \right].
\end{aligned}
$$

Taking expectation on both sides and applying (18) and (19), we have
$$
\begin{aligned}
&\frac{C_1 d \log p(n_1) + 2C_2 \beta(\mathcal{H}, n_1) n_2 / n_1 + C_3}{n_2} \\
&= \frac{C_1 d \log p(n_1) + C_2 \mathbb{E}[N] + C_3}{n_2} \\
&\geqslant \mathbb{P}(\widehat{h}_{S', \mathbf{I}}(I) \neq Y_I) \\
&\geqslant \frac{1}{2} \mathbb{E}\left[ \mathbb{1}_{Y_{2n_2} \in \widehat{\mu}(X_{2n_2})} \mathbb{1}_{Y_{2n_2+1} \in \widehat{\mu}(X_{2n_2+1})} \mathbb{1}_{\widetilde{h}_{S', \mathbf{I}', 2n_2+1}(2n_2+1) \neq Y_{2n_2+1}} \right]
\end{aligned}
$$

which leads to

$$\mathbb{P}\left(\widehat{h}_S(X) \neq Y\right)$$

$$=\mathbb{P}\left(\widetilde{h}_{S',\mathbf{I'},2n_2+1}(2n_2+1) \neq Y_{2n_2+1}\right)$$

$$\leqslant \mathbb{E}\left[1 - \mathbb{1}_{Y_{2n_2} \in \widehat{\mu}(X_{2n_2})} \mathbb{1}_{Y_{2n_2+1} \in \widehat{\mu}(X_{2n_2+1})}\right]$$

$$\quad + \mathbb{E}\left[\mathbb{1}_{Y_{2n_2} \in \widehat{\mu}(X_{2n_2})} \mathbb{1}_{Y_{2n_2+1} \in \widehat{\mu}(X_{2n_2+1})} \mathbb{1}_{\widetilde{h}_{S',\mathbf{I'},2n_2+1}(2n_2+1) \neq Y_{2n_2+1}}\right]$$

$$\leqslant \mathbb{P}\left(Y_{2n_2} \notin \widehat{\mu}(X_{2n_2})\right) + \mathbb{P}\left(Y_{2n_2+1} \notin \widehat{\mu}(X_{2n_2+1})\right)$$

$$\quad + \frac{2(C_1 d \log p(n_1) + 2C_2\beta(\mathcal{H},n_1)n_2/n_1 + C_3)}{n_2}$$

$$\leqslant \frac{2(\beta(\mathcal{H},n_1)n_2/n_1 + C_1 d \log p(n_1) + 2C_2\beta(\mathcal{H},n_1)n_2/n_1 + C_3)}{n_2}.$$

Now, for any $n \in \mathbb{N}$ such that $n \geqslant 3$, setting $n_1 = n - 2\lfloor n/3 \rfloor$, $n_2 = \lfloor n/3 \rfloor$, $(S,(X,Y)) \sim \mathcal{D}^{n+1}$ with $S \in (\mathcal{X} \times \mathcal{Y})^n$ and $(X,Y) \in \mathcal{X} \times \mathcal{Y}$, we have

$$\mathbb{P}\left(\widehat{h}_S(X) \neq Y\right) = O\left(\frac{\beta(\mathcal{H},n_1) + d \log p(n_1)}{n}\right).$$

Then, when $\beta(\mathcal{H},n)$ and $p(n)$ are nondecreasing in $n$, by Theorem 2.6, there exists a learner $\mathcal{A}'$ such that for any $n \geqslant 4$ and $\delta \in (0,1)$, with probability at least $1 - \delta$ over $S \sim \mathcal{D}^n$, we have

$$\mathrm{er}_{\mathcal{D}}(\mathcal{A}'(S,\mathcal{H}) = O\left(\frac{\beta(\mathcal{H},n_1) + d \log p(n_1) + \log(1/\delta)}{n}\right).$$

$\square$

## D  Proofs of the results in Section 2.3

In this section, we provide the proofs of Theorem 2.8, Corollary 2.9, and Theorem 2.10 in Section 2.3.

*Proof of Theorem 2.8.* Assume that $\mathcal{A}_{\mathrm{list}}$ satisfies that $\mathcal{M}_{\mathcal{A}_{\mathrm{list}},\mathcal{H}}(1/2 - \gamma, \nu) < \infty$ for some $\gamma \in (0, 1/2)$ and $\nu \in (0, \gamma/18]$. Define the random variable $J_k := \mathbb{1}_{\mathrm{er}_{\mathcal{D}_k}(\mu_k) > 1/2 - \gamma}$ for any $k \in [K]$. Then, we have

$$\mathbb{E}[J_k] = \mathbb{P}(J_k = 1) \leqslant \nu.$$

Define the event $\mathcal{E} := \left\{\sum_{k=1}^K J_k \geqslant 2\nu K\right\}$. By the multiplicative Chernoff bound, we have

$$\mathbb{P}(\mathcal{E}) = \mathbb{P}\left(\sum_{k=1}^K J_k \geqslant 2\nu K\right) \leqslant e^{-\nu K/3} \leqslant \delta$$

as $K \geqslant \frac{3\log(1/\delta)}{\nu}$. Define $Z_k := \sum_{i=1}^n \mathcal{D}_k(\{(x_i,y_i)\}) \exp\left(-\alpha\left(2\mathbb{1}_{y_i \in \mu_k(x_i)} - 1\right)\right)$ for all $k \in [K]$. Since $\mathcal{D}_{K+1}$ is a probability distribution over $S$, we have

$$1 = \sum_{i=1}^n \mathcal{D}_{K+1}(\{(x_i,y_i)\})$$

$$= \sum_{i=1}^n \frac{\mathcal{D}_k(\{(x_i,y_i)\}) \exp\left(-\alpha\left(2\mathbb{1}_{y_i \in \mu_k(x_i)} - 1\right)\right)}{Z_K}$$

$$= \frac{1}{n} \sum_{i=1}^n \frac{\exp\left(-\alpha \sum_{k=1}^K \left(2\mathbb{1}_{y_i \in \mu_k(x_i)} - 1\right)\right)}{\prod_{k=1}^K Z_k},$$

which implies that

$$n \prod_{k=1}^K Z_k = \sum_{i=1}^n \exp\left(-\alpha \sum_{k=1}^K \left(2\mathbb{1}_{y_i \in \mu_k(x_i)} - 1\right)\right). \tag{20}$$

For any $k \in [K]$, we have

$$Z_k = \sum_{i=1}^{n} \mathcal{D}_k(\{(x_i, y_i)\}) \exp\left(-\alpha\left(2\mathbb{1}_{y_i \in \mu_k(x_i)} - 1\right)\right)$$

$$= \sum_{i \in [n]: y_i \in \mu_k(x_i)} \mathcal{D}_k(\{(x_i, y_i)\})e^{-\alpha} + \sum_{i \in [n]: y_i \notin \mu_k(x_i)} \mathcal{D}_k(\{(x_i, y_i)\})e^{\alpha}$$

$$= (1 - \mathrm{er}_{\mathcal{D}_k}(\mu_k))e^{-\alpha} + \mathrm{er}_{\mathcal{D}_k}(\mu_k)e^{\alpha} = \mathrm{er}_{\mathcal{D}_k}(\mu_k)(e^{\alpha} - e^{-\alpha}) + e^{-\alpha}.$$

If $J_k = 0$, we have

$$Z_k \leqslant \gamma e^{\alpha} + (1 - \gamma)e^{-\alpha} = \sqrt{1 - \gamma^2}.$$

If $J_k = 1$, we have

$$Z_k \leqslant e^{\alpha} = \sqrt{1 + \frac{\gamma}{1/2 - \gamma/2}} \leqslant \sqrt{1 + 4\gamma}.$$

Then, under the event $\mathcal{E}^c$, we can upper bound

$$\prod_{k=1}^{K} Z_k \leqslant (1 - \gamma^2)^{(1/2 - \nu)K}(1 + 4\gamma)^{\nu K}$$

$$\leqslant \exp\left(-\gamma K\left(\frac{\gamma}{2} - (\gamma + 4)\nu\right)\right)$$

$$\leqslant \exp\left(-\frac{\gamma K}{4}\right) \leqslant \frac{\delta}{n}.$$

where the second last inequality follows from $\nu \leqslant \frac{\gamma}{18} \leqslant \frac{\gamma}{4(\gamma+4)}$ and the last inequality follows from $K \geqslant \frac{4\log(n/\delta)}{\gamma}$. By (20) and the above inequality, we have that on $\mathcal{E}^c$,

$$\sum_{i=1}^{n} \exp\left(-\alpha \sum_{k=1}^{K}\left(2\mathbb{1}_{y_i \in \mu_k(x_i)} - 1\right)\right) \leqslant \delta,$$

which implies that $\exp\left(-\alpha \sum_{k=1}^{K}\left(2\mathbb{1}_{y_i \in \mu_k(x_i)} - 1\right)\right) \leqslant \delta$ for all $i \in [n]$. It follows that

$$\exp(-f(x_i, y_i)) \leqslant \delta^{1/\alpha K}$$

for $f(x, y) := \frac{1}{K}\sum_{k=1}^{K}\left(2\mathbb{1}_{y \in \mu_k(x)} - 1\right)$, $\forall (x, y) \in \mathcal{X} \times \mathcal{Y}$. Thus, we have

$$f(x_i, y_i) \geqslant \frac{\log(1/\delta)}{\alpha K} = \frac{2\log(1/\delta)}{\log\left(\frac{1/2+\gamma}{1/2-\gamma}\right)\lceil 4\log(n/\delta)/\gamma\rceil} > 0, \ \forall i \in [n].$$

Since by definition,

$$f(x, y) > 0 \Leftrightarrow y \in \mu(x), \ \forall(x, y) \in \mathcal{X} \times \mathcal{Y},$$

we can conclude that on $\mathcal{E}^c$,

$$y_i \in \mu(x_i), \ \forall i \in [n],$$

i.e., with probability at least $1 - \delta$, $y_i \in \mu(x_i)$ for all $i \in [n]$. Moreover, by da Cunha et al. [2024, Lemma 3.3], the randomized compression scheme $S \mapsto (S^1, \ldots, S^K)$ is stable. Thus, we can apply da Cunha et al. [2024, Theorem 1.2] with compression size $s_n = mK = O\left(\frac{\mathcal{M}_{\mathcal{A}_{\mathrm{list}}, \mathcal{H}}(1/2 - \gamma, \nu)\log(n/\delta)}{\gamma}\right)$ to conclude the proof. $\qquad \square$

For the proofs of Corollary 2.9 and Theorem 2.10, we will need the following Lemma.

**Lemma D.1.** *If $x > 0$ satisfies $x \leqslant a\log(x/a) + b$ for some $a, b > 0$, then, we have $x \leqslant 2a + 2b$.*

*Proof.* Define $f(x) := x - a \log(x/a) - b$ for $x > 0$. Then, we have $f'(x) = \frac{x-a}{x}$, which implies that $f$ decreases with $x$ for $x \in (0, a)$ and increases with $x$ for $x > a$. Since $2a + 2b > a$, it suffices to prove that $f(2a + 2b) \geqslant 0$. Indeed,

$$f(2a + 2b) = (2 - \log 2)a + b - a \log((a + b)/a) > a\left((a + b)/a - \log((a + b)/a)\right) \geqslant 0.$$

$\square$

*Proof of Corollary 2.9.* By Brukhim et al. [2022, Theorem 36] (choosing $t = \lceil \sqrt{d} \rceil$), there exists an $n \to r$ sample compression scheme $\mathcal{A}_M$ for any $\mathcal{H} \subseteq \mathcal{Y}^{\mathcal{X}}$ of DS dimension $\dim(\mathcal{H}) = d < \infty$ where

$$r = O((d^{3/2} + d \log(p)) \log(n)) \text{ with } p = O((e\sqrt{d})^{\sqrt{d}} \log(n)). \tag{21}$$

Then, by David et al. [2016, Theorem 3.1], there exists a universal constant $C > 0$ such that for any $\mathcal{D} \in \mathrm{RE}(\mathcal{H})$, $\delta \in (0, 1)$, $n \in \mathbb{N}$ large enough, and $S \sim \mathcal{D}^n$, letting $h_S := \mathcal{A}_M(S, \mathcal{H})$ denote the output classifier of the above $n \to r$ sample compression scheme $\mathcal{A}_M$, we have that

$$\mathrm{er}_{\mathcal{D}}(h_S) \leqslant \frac{C(r \log(n/r) + \log(1/\delta))}{n}$$

with probability at least $1 - \delta$. Thus, for any $\varepsilon \in (0, 1)$, if

$$n \geqslant \frac{Cr}{\varepsilon} \log(n/r) + \frac{C \log(1/\delta)}{\varepsilon} = \frac{Cr}{\varepsilon} \log\left(\frac{n}{Cr/\varepsilon}\right) + \frac{Cr}{\varepsilon} \log\left(\frac{C}{\varepsilon}\right) + \frac{C \log(1/\delta)}{\varepsilon},$$

we have $\mathrm{er}_{\mathcal{D}}(h_S) \leqslant \varepsilon$ with probability at least $1 - \delta$. By Lemma D.1, it suffices to require

$$n \geqslant \frac{2Cr}{\varepsilon} \log\left(\frac{Ce}{\varepsilon}\right) + \frac{2C \log(1/\delta)}{\varepsilon} = \frac{2C}{\varepsilon}\left(r \log\left(\frac{Ce}{\varepsilon}\right) + \log\left(\frac{1}{\delta}\right)\right).$$

Applying the upper bound of $r$ in (21), it suffices to require

$$n \geqslant C'\left(\frac{(d^{3/2} + d \log(p)) \log(1/\varepsilon)}{\varepsilon} \log(n) + \frac{\log(1/\delta)}{\varepsilon}\right)$$

for some universal constant $C' > 0$. By Lemma D.1 again, it suffices to require

$$n \geqslant \frac{2C'(d^{3/2} + d \log(p)) \log(1/\varepsilon)}{\varepsilon} \log\left(\frac{eC'(d^{3/2} + d \log(p)) \log(1/\varepsilon)}{\varepsilon}\right) + \frac{2C' \log(1/\delta)}{\varepsilon}.$$

Since $\log(\log(1/\varepsilon)/\varepsilon) = \log(1/\varepsilon) + \log\log(1/\varepsilon) \leqslant 2 \log(1/\varepsilon)$, it suffices to require

$$n \geqslant C''\left(\frac{(d^{3/2} + d \log(p)) \log(1/\varepsilon)}{\varepsilon} \log\left(\frac{d^{3/2} + d \log(p)}{\varepsilon}\right) + \frac{\log(1/\delta)}{\varepsilon}\right)$$

for some universal constant $C'' > 0$. Applying the upper bound of $p$ in (21), it suffices to require that

$$n \geqslant C'''\left(\frac{(d^{3/2} \log(d) + d \log\log(n)) \log(1/\varepsilon)}{\varepsilon} \log\left(\frac{d^{3/2} \log(d) + d \log\log(n)}{\varepsilon}\right) + \frac{\log(1/\delta)}{\varepsilon}\right)$$

for some universal constant $C''' > 0$. For $\varepsilon = 1/6$ and $\delta = 1/54$, we require that

$$n \geqslant C_1(d^{3/2} \log(d) + d \log\log(n)) \log\left(d^{3/2} \log(d) + d \log\log(n)\right)$$

for some universal constant $C_1 > 0$. If $C_1(\sqrt{2C_1} + 1)d^{3/2} \log(d) \log\left((\sqrt{2C_1} + 1)d^{3/2} \log(d)\right) \leqslant n \leqslant e^{d\sqrt{2C_1 d}}$, we have $\sqrt{2C_1}d^{3/2} \log(d) \geqslant d \log\log(n)$ and

$$C_1(d^{3/2} \log(d) + d \log\log(n)) \log(d^{3/2} \log(d) + d \log\log(n))$$
$$\leqslant C_1(\sqrt{2C_1} + 1)d^{3/2} \log(d) \log\left((\sqrt{2C_1} + 1)d^{3/2} \log(d)\right) \leqslant n.$$

If $n > e^{d\sqrt{2C_1 d}}$, we have $\log(\log(n)) > \sqrt{2C_1 d} \log(d)$, $\log(\log(n))^2 > 2C_1 d \log^2(d) \geqslant 2C_1(d \log(2d) + d)$, and

$$C_1(d^{3/2} \log(d) + d \log\log(n)) \log(d^{3/2} \log(d) + d \log\log(n))$$
$$\leqslant 2C_1 d \log(\log(n)) \log(2d \log\log(n))$$
$$= 2C_1 d \log(2d) \log(\log(n)) + 2C_1 d \log(\log(n)) \log(\log\log(n))$$
$$\leqslant \log(\log(n))^3 \log(\log\log(n))) \leqslant n.$$

Therefore, we can conclude that if $n \geqslant C_1(\sqrt{2C_1} + 1)d^{3/2}\log(d)\log\left((\sqrt{2C_1} + 1)d^{3/2}\log(d)\right) = \Theta(d^{3/2}\log^2(d))$, we have $\mathbb{P}(\mathrm{er}_{\mathcal{D}}(h_S) > 1/6) \leqslant 1/54$, i.e.,

$$\mathcal{M}_{\mathcal{A}_M, \mathcal{H}}(1/2 - 1/3, 1/54) = O(d^{3/2}\log^2(d)).$$

Let $\mathcal{A}_B$ denote the multiclass learner output by $\mathcal{A}_{\mathrm{boost}}$ in Algorithm 2 using $\mathcal{A}_M$ as the weak list learner of size 1. Then, by Theorem 2.8, we have

$$\varepsilon_{\mathcal{A}_B, \mathcal{H}}(n) = O\left(\frac{d^{3/2}\log^2(d)\log(n)}{n}\right), \; \forall n \in \mathbb{N}.$$

Next, by Theorem 2.6, there exists a multiclass learner $\mathcal{A}$ such that for any $\mathcal{D} \in \mathrm{RE}(\mathcal{H})$, $\delta \in (0, 1)$, $n \in \mathbb{N}$, and $S \sim \mathcal{D}^n$, with probability at least $1 - \delta$, we have

$$\mathrm{er}_{\mathcal{D}}(\mathcal{A}(S, \mathcal{H})) = O\left(\frac{d^{3/2}\log^2(d)\log(n) + \log(1/\delta)}{n}\right).$$

Setting $\delta = 1/n$ and observing that $\mathrm{er}_{\mathcal{D}} \in [0, 1]$, it follows that $\varepsilon_{\mathcal{A}, \mathcal{H}}(n) = O\left(\frac{d^{3/2}\log^2(d)\log(n)}{n}\right)$. Moreover, by Lemma D.1, for any $\varepsilon \in (0, 1)$, $n \geqslant O\left(\frac{d^{3/2}\log^2(d)\log(d/\varepsilon) + \log(1/\delta)}{\varepsilon}\right)$ implies that $\mathrm{er}_{\mathcal{D}}(\mathcal{A}(S, \mathcal{H})) \leqslant \varepsilon$ with probability at least $1 - \delta$. Thus, we have

$$\mathcal{M}_{\mathcal{A}, \mathcal{H}}(\varepsilon, \delta) = O\left(\frac{d^{3/2}\log^2(d)\log(d/\varepsilon) + \log(1/\delta)}{\varepsilon}\right).$$

$\square$

*Proof of Theorem 2.10.* By Brukhim et al. [2022, Theorem 39] (choosing $t = \lceil\sqrt{d}\rceil$), there exists an $n \to r$ list sample compression scheme $\mathcal{A}_{\mathrm{list}}$ of size $p = O((e\sqrt{d})^{\sqrt{d}}\log(n))$ for any $\mathcal{H} \subseteq \mathcal{Y}^{\mathcal{X}}$ of DS dimension $\dim(\mathcal{H}) = d < \infty$ where

$$r = O(d^{3/2}\log(n)). \tag{22}$$

Define the following loss function for menus $\mu$,

$$\ell(\mu, (x, y)) := \mathbb{1}_{y \notin \mu(x)}, \; \forall (x, y) \in \mathcal{X} \times \mathcal{Y}.$$

Then, we can apply the proof of David et al. [2016, Theorem 3.1] with the loss function $\ell$ to show that there exists a universal constant $C > 0$ such that for any $n \to r$ list sample compression scheme $\mathcal{A}_{LSC}$ for $\mathcal{H}$, any $\mathcal{D} \in \mathrm{RE}(\mathcal{H})$, $\delta \in (0, 1)$, $n \in \mathbb{N}$ large enough, and $S \sim \mathcal{D}^n$, letting $\mu_S := \mathcal{A}_{LSC}(S, \mathcal{H})$ denote the output menu, we have

$$\mathrm{er}_{\mathcal{D}}(\mathcal{A}_{LSC}(S, \mathcal{H})) = \mathbb{E}_{(X,Y)\sim\mathcal{D}}[\ell(\mathcal{A}_{LSC}(S, \mathcal{H}), (X, Y)) \mid S] \leqslant \frac{C(r\log(n/r) + \log(1/\delta))}{n}$$

with probability at least $1 - \delta$. Thus, for any $\varepsilon \in (0, 1)$, if

$$n \geqslant \frac{Cr}{\varepsilon}\log(n/r) + \frac{C\log(1/\delta)}{\varepsilon} = \frac{Cr}{\varepsilon}\log\left(\frac{n}{Cr/\varepsilon}\right) + \frac{Cr}{\varepsilon}\log\left(\frac{C}{\varepsilon}\right) + \frac{C\log(1/\delta)}{\varepsilon},$$

we have $\mathrm{er}_{\mathcal{D}}(\mathcal{A}_{LSC}(S, \mathcal{H})) \leqslant \varepsilon$ with probability at least $1 - \delta$. By Lemma D.1, it suffices to require

$$n \geqslant \frac{2Cr}{\varepsilon}\log\left(\frac{Ce}{\varepsilon}\right) + \frac{2C\log(1/\delta)}{\varepsilon} = \frac{2C}{\varepsilon}\left(r\log\left(\frac{Ce}{\varepsilon}\right) + \log\left(\frac{1}{\delta}\right)\right).$$

Since $\mathcal{A}_{\mathrm{list}}$ is an $n \to r$ list sample compression scheme for $\mathcal{H}$ with $r$ bounded in (22), The above results imply that in order for $\mathbb{P}(\mathrm{er}_{\mathcal{D}}(\mathcal{A}_{\mathrm{list}}(S, \mathcal{H})) > \varepsilon) \leqslant \delta$, it suffices to require

$$n \geqslant C'\left(\frac{d^{3/2}\log(1/\varepsilon)}{\varepsilon}\log(n) + \frac{\log(1/\delta)}{\varepsilon}\right)$$

for some universal constant $C' > 0$. By Lemma D.1 again, it suffices to require

$$n \geqslant \frac{2C'd^{3/2}\log(1/\varepsilon)}{\varepsilon}\log\left(\frac{eC'd^{3/2}\log(1/\varepsilon)}{\varepsilon}\right) + \frac{2C'\log(1/\delta)}{\varepsilon}.$$

Since $\log(\log(1/\varepsilon)/\varepsilon) = \log(1/\varepsilon) + \log\log(1/\varepsilon) \leqslant 2\log(1/\varepsilon)$, it suffices to require

$$n \geqslant C'' \left( \frac{d^{3/2} \log(d/\varepsilon) \log(1/\varepsilon) + \log(1/\delta)}{\varepsilon} \right)$$

for some universal constant $C'' > 0$, which implies that

$$\mathcal{M}_{\mathcal{A}_{\mathrm{list}}, \mathcal{H}}(1/2 - 1/3, 1/54) = O\left( d^{3/2} \log(d) \right).$$

Now, we can define $\mathcal{A}_L$ to be the list learner output by $\mathcal{A}_{\mathrm{boost}}$ in Algorithm 2 using $\mathcal{A}_{\mathrm{list}}$ as the weak list learner of size $O((e\sqrt{d})^{\sqrt{d}} \log(n))$ for training sample size $n \in \mathbb{N}$. Then, by Theorem 2.8, we have that for any training sample size $n \in \mathbb{N}$,

$$\varepsilon_{\mathcal{A}_L, \mathcal{H}}(n) = O\left( \frac{d^{3/2} \log(d) \log(n)}{n} \right)$$

and the size of $\mathcal{A}_L$ is also $O((e\sqrt{d})^{\sqrt{d}} \log(n))$.

$\square$

# E  Proof of Theorem 2.11

*Proof.* By Theorem 2.10, there exists a list learner $A_L$ with $p(\mathcal{H}, n) = O\left((e\sqrt{d})^{\sqrt{d}} \log(n)\right)$ and $\beta(\mathcal{H}, n) = O\left(d^{3/2} \log(d) \log(n)\right)$ for any $n \in \mathbb{N}$ and concept class $\mathcal{H} \subseteq \mathcal{Y}^{\mathcal{X}}$ with $\dim(\mathcal{H}) = d \in \mathbb{N}$ in the context of Theorem 2.7. Thus, by Theorem 2.7, there exists a multiclass learner $\mathcal{A}_{\mathrm{multi}} = \mathcal{A}'_{\mathrm{red}}$ such that for any $\mathcal{D} \in \mathrm{RE}(\mathcal{H})$, any $\delta \in (0,1)$, $S \sim \mathcal{D}^n$, and $n_1 = n - 2\lfloor n/3 \rfloor$, with probability at least $1 - \delta$,

$$\begin{aligned}
\mathrm{er}_{\mathcal{D}}(\mathcal{A}_{\mathrm{multi}}(S, \mathcal{H})) &= O\left( \frac{\beta(\mathcal{H}, n_1) + d \log p(\mathcal{H}, n_1) + \log(1/\delta)}{n} \right) \\
&= O\left( \frac{d^{3/2} \log(d) \log(n) + d^{3/2} \log(e\sqrt{d} \log(n)) + \log(1/\delta)}{n} \right) \\
&= O\left( \frac{d^{3/2} \log(d) \log(n) + \log(1/\delta)}{n} \right).
\end{aligned}$$

For any $\varepsilon \in (0,1)$, by Lemma D.1, if

$$\begin{aligned}
n &\geqslant \frac{2d^{3/2} \log(d)}{\varepsilon} \left( 1 + \frac{5}{2} \log(d/\varepsilon) \right) + \frac{2\log(1/\delta)}{\varepsilon} \\
&\geqslant \frac{2d^{3/2} \log(d)}{\varepsilon} \left( 1 + \frac{3}{2} \log(d) + \log\log(d) + \log(1/\varepsilon) \right) + \frac{2\log(1/\delta)}{\varepsilon} \\
&\geqslant \frac{2d^{3/2} \log(d)}{\varepsilon} \log\left( \frac{ed^{3/2} \log(d)}{\varepsilon} \right) + \frac{2\log(1/\delta)}{\varepsilon},
\end{aligned}$$

then, we have

$$n \geqslant \frac{d^{3/2} \log(d)}{\varepsilon} \log(n) + \frac{\log(1/\delta)}{\varepsilon}$$

i.e.,

$$\frac{d^{3/2} \log(d) \log(n) + \log(1/\delta)}{n} \leqslant \varepsilon.$$

It follows that

$$\mathcal{M}_{\mathcal{A}_{\mathrm{multi}}, \mathcal{H}}(\varepsilon, \delta) = O\left( \frac{d^{3/2} \log(d) \log(d/\varepsilon) + \log(1/\delta)}{\varepsilon} \right).$$

(9) follows directly from (6) by plugging in $p(\mathcal{H}, n) = f_1(d)$ and $\beta(\mathcal{H}, n) = f_2(d)$ for $d = \dim(\mathcal{H})$ and any $n \in \mathbb{N}$.

$\square$

# F  Classes of DS dimension 1

In this section, we present the proof of Proposition 3.3 and Theorem 3.2.

*Proof of Proposition 3.3.* Let $E_n$ denote the edge set of the hypergraph $\mathcal{G}(V_n)$. Suppose on the contrary that there exists a cycle consisting of pairwise different vertices $\mathbf{y}^0, \dots, \mathbf{y}^{m-1} \in V_n$ and pairwise different edges $\mathbf{e}^0 = (e_{i_0, f_0}, i_0), \dots, \mathbf{e}^{m-1} = (e_{i_{m-1}, f_{m-1}}, i_{m-1}) \in E_n$ for some $m \in \{4, \dots, |V_n|\}$ such that $\mathbf{y}^j, \mathbf{y}^{(j+1) \bmod m} \in (e_{i_j, f_j}, i_j)$ for all $0 \leqslant j \leqslant m-1$. Since the edges are pairwise different, by the definition of $E_n$, we have $i_j \neq i_{(j-1) \bmod m}$ for all $0 \leqslant j \leqslant m-1$.

Define $a := i_{m-1} \in [n]$, $b := i_0 \in [n]$, $p_0 := y_a^0 = y_a^1$, $p_{-1} := y_a^{m-1} \neq p_0$, and $q_0 := y_b^0 = y_b^{m-1}$. Then, we have $a \neq b$.

For any $k \in \mathbb{N}$, we define
$$j_k := \max\{j \in \mathbb{N}_0 : j \leqslant m-1, y_a^j = p_{k-1}\},$$
$p_k := y_a^{(j_k+1) \bmod m}$, and $q_k := y_b^{j_k} = y_b^{(j_k+1) \bmod m}$ because $i_{j_k} = a$. Define
$$K := \min\{k \in \mathbb{N} : p_k = p_{-1}\}.$$

By definition, we have $p_0, \dots, p_K$ are pairwise different. There are two cases depending on the values of $q_1, \dots, q_K$ as follows.

1. Suppose that there exists some $k \in [K]$ such that $q_k \neq q_0$. Define $l_0 := 0$ and
$$l_w := \min\{k \in [K] : k > l_{w-1}, q_k \neq q_{l_{w-1}}\}$$
for all $w \in \mathbb{N}$ with the convention that $\inf \varnothing = +\infty$. Define
$$v := \max\{w \in \mathbb{N} : l_w \neq +\infty\}.$$
We have $v \in [K]$ and $q_{l_w} \neq q_{l_{w-1}}$ for all $w \in [v]$. Note that
$$p_{-1} = p_K = y_a^{m-1}, \quad q_0 = y_b^{m-1},$$
$$p_{l_0} = p_0 = y_a^0, \quad q_{l_0} = q_0 = y_b^0,$$

$$p_{l_w-1} = y_a^{j_{l_w-1}+1}, \quad q_{l_{w-1}} = y_b^{j_{l_w-1}+1},$$
$$p_{l_w-1} = y_a^{j_{l_w}}, \quad q_{l_w} = y_b^{j_{l_w}},$$
$$p_{l_w} = y_a^{j_{l_w}+1}, \quad q_{l_w} = y_b^{j_{l_w}+1}$$
for all $w \in [v]$, and
$$p_K = p_{-1} = y_a^{j_K+1}, \quad q_K = y_b^{j_K+1}.$$

Since it always holds that $q_K = q_{l_v}$, there are two cases depending on the value of $q_{l_v}$ and $q_0$ as follows.

   1.1 Suppose that $q_{l_v} \neq q_0$. Then, $V_n|_{(a,b)}$ contains the following pseudo-cube of dimension 2:
   $$(p_{l_1-1}, q_0),$$
   $$(p_{l_1-1}, q_{l_1}),$$
   $$(p_{l_2-1}, q_{l_1}),$$
   $$(p_{l_2-1}, q_{l_2}),$$
   $$\vdots$$
   $$(p_{l_v-1}, q_{l_{v-1}}),$$
   $$(p_{l_v-1}, q_{l_v}),$$
   $$(p_{-1}, q_{l_v}),$$
   $$(p_{-1}, q_0),$$
   which contradicts the assumption that $\dim(V_n) = 1$.

1.2 Suppose that $q_{l_v} = q_0$. Then, $V_n|_{(a,b)}$ contains the following pseudo-cube of dimension 2:

$$
\begin{aligned}
&(p_{l_1-1}, q_0), \\
&(p_{l_1-1}, q_{l_1}), \\
&(p_{l_2-1}, q_{l_1}), \\
&(p_{l_2-1}, q_{l_2}), \\
&\quad\vdots \\
&(p_{l_v-1}, q_{l_{v-1}}), \\
&(p_{l_v-1}, q_{l_v} = q_0),
\end{aligned}
$$

which contradicts the assumption that $\dim(V_n) = 1$.

Thus, Case 1 does not exist.

2. Suppose that $q_k = q_0$ for all $k \in [K]$. Since $y_a^0 = p_0 = p_a^{j_1}$, $y_b^0 = q_0 = q_1 = p_b^{j_1}$, and $\mathbf{y}^0 \neq \mathbf{y}^{j_1}$, there must exist some $c \in [n]\backslash\{a, b\}$ such that $y_c^{m-1} = y_c^1 = y_c^0 \neq y_c^{j_1} = y_c^{j_1+1}$. We define $r_0 := y_c^0$ and $r_k := y_c^{j_k} = y_c^{j_k+1}$ for $k \in [K]$. There are two cases depending on the value of $r_k$ for $k \in [K]$ as follows.

2.1 Suppose that there exists some $k \in [K]$ such that $r_k \neq r_1$. Similar to Case 1, we define $l_0 := 1$ and

$$l_w := \min\{k \in [K] : k > l_{w-1}, r_k \neq r_{l_{w-1}}\}$$

for $w \in \mathbb{N}$. Define $v := \max\{w \in \mathbb{N} : l_w \neq +\infty\}$. We have $v \in [K]$ and $r_{l_w} \neq r_{l_{w-1}}$ for all $w \in [v]$. Note that

$$
\begin{aligned}
p_{-1} = p_K &= y_a^{m-1}, \quad r_0 = y_c^{m-1}, \\
p_0 &= y_a^0, \quad r_0 = y_c^0, \\
p_0 &= y_a^{j_1}, \quad r_{l_0} = r_1 = y_c^{j_1},
\end{aligned}
$$

$$
\begin{aligned}
p_{l_w-1} &= y_a^{j_{l_{w-1}}+1}, \quad r_{l_{w-1}} = y_c^{j_{l_{w-1}}+1}, \\
p_{l_w-1} &= y_a^{j_{l_w}}, \quad r_{l_w} = y_c^{j_{l_w}},
\end{aligned}
$$

for all $w \in [v]$, and

$$p_K = p_{-1} = y_a^{j_K+1}, \quad r_K = y_c^{j_K+1}.$$

Since it always holds that $r_K = r_{l_v}$, there are two cases depending on the value of $q_{l_v}$ and $q_0$ as follows.

2.1.1 Suppose that $r_{l_v} \neq r_0$. Then, $V_n|_{(a,c)}$ contains the following pseudo-cube of dimension 2:

$$
\begin{aligned}
&(p_0, r_0), \\
&(p_0, r_{l_0} = r_1), \\
&(p_{l_1-1}, r_{l_0}), \\
&(p_{l_1-1}, q_{l_1}), \\
&\quad\vdots \\
&(p_{l_v-1}, r_{l_{v-1}}), \\
&(p_{l_v-1}, r_{l_v}), \\
&(p_{-1}, r_{l_v}), \\
&(p_{-1}, r_0),
\end{aligned}
$$

which contradicts the assumption that $\dim(V_n) = 1$.

2.1.2 Suppose that $r_{l_v} = r_0$. Then, $V_n|_{(a,c)}$ contains the following pseudo-cube of dimension 2:

$$(p_0, r_0),$$
$$(p_0, r_{l_0} = r_1),$$
$$(p_{l_1-1}, r_{l_0}),$$
$$(p_{l_1-1}, q_{l_1}),$$
$$\vdots$$
$$(p_{l_v-1}, r_{l_{v-1}}),$$
$$(p_{l_v-1}, r_{l_v} = r_0),$$

which contradicts the assumption that $\dim(V_n) = 1$.

2.2 Suppose that $r_k = r_1$ for all $k \in [K]$. Then, we have

$$p_{-1} = y_a^{m-1}, \quad r_0 = y_c^{m-1},$$
$$p_0 = y_a^0, \quad r_0 = y_c^0,$$
$$p_0 = y_a^{j_1}, \quad r_1 = y_c^{j_1},$$
$$p_{-1} = y_a^{j_K+1}, \quad r_1 = y_c^{j_K+1},$$

which implies that $V_n|_{(a,c)}$ contains a pseudo-cube of dimension 2.

Thus, Case 2 does not exist either.

In conclusion, there exists no cycle in the hypergraph $\mathcal{G}(V_n) = (V_n, E_n)$. $\qquad\square$

*Proof of Theorem 3.2.* It suffices to show that the for any $n \in \mathbb{N}$ and $V_n \subseteq \mathcal{Y}^n$, the average degree of $G = \mathcal{G}(V_n) = (V_n, E_n)$ with $|V_n| < \infty$ and $\dim(V_n) = 1$ is at most 2.

We prove by induction on $|V_n|$. When $|V_n| = 1$, we have $E_n = \varnothing$ and $\mathsf{avgdeg}(G) = 0 < 2$. When $|V_n| = 2$, we have $\sum_{e \in E_n : |e| \geqslant 2} |e| \leqslant 2$ and $\mathsf{avgdeg}(G) \leqslant 1 < 2$. Suppose that $\mathsf{avgdeg}(G) \leqslant 2$ for any $V_n$ of size $|V_n| \leqslant m$ with some $m \in \mathbb{N}$. When $|V_n| = m+1$, since there is no cycle in $G$ according to Proposition 3.3, the set of vertices of degree 1 ($V_n^1 := \{\mathbf{y} \in V_n : \deg(\mathbf{y}) = 1\}$) is not empty. Define $V_n^2 := V_n \backslash V_n^1$ and $E_n^2$ to be the edge set such that $(V_n^2, E_n^2)$ is the one-inclusion graph on $V_n^2$. Then, we have $|V_n^2| = |V_n| - |V_n^1| \leqslant m$ and

$$\sum_{e \in E_n : |e| \geqslant 2} |e| \leqslant \sum_{e \in E_n^2 : |e| \geqslant 2} |e| + 2|V_n^1|$$

because deleting a vertex of degree 1 decreases the total degree by at most 2. By the induction hypothesis, we have $\sum_{e \in E_n^2} |e| \leqslant 2|V_n^2|$. Thus,

$$\sum_{e \in E_n : |e| \geqslant 2} |e| \leqslant 2(|V_n^1| + |V_n^2|) = 2|V_n|$$

which implies that $\mathsf{avgdeg}(G) \leqslant 2$. By induction, $\mathsf{avgdeg}(G) \leqslant 2$ for any $V_n$ with $|V_n| < \infty$ and $\dim(V_n) = 1$. $\qquad\square$

# G  Pivot shifting

In this section, we present the proofs of Lemma 3.6 and Lemma 3.8.

*Proof of Lemma 3.6.* For notational convenience, we let $V'_{n-1}$ denote $\mathfrak{P}(V_n)$ which is defined in Definition 3.5. Define

$$V_{n-1} := \bigcup_{y \in \mathcal{Y}} \{(y_1, \ldots, y_{n-1}) \in \mathcal{Y}^{n-1} : (y_1, \ldots, y_{n-1}, y) \in V_n\}$$

and
$$V''_{n-1} := V_{n-1} \backslash V'_{n-1}.$$

Let $E_n$ denote the edge set in $\mathcal{G}(V_n)$ and $E_{n-1}$ denote the edge set in $\mathcal{G}(V_{n-1})$. For any $y \in \mathcal{Y}$, define
$$V_{n,y} := \left\{ (y_1, \ldots, y_n) \in V_n : y_n = y, (y_1, \ldots, y_{n-1}) \in V'_{n-1} \right\}.$$

By the assumption on $a \in \mathcal{Y}$, we have $|V_n| = |V''_{n-1}| + \sum_{y \in \mathcal{Y}} |V_{n,y}|$, $|V'_{n-1}| = |V_{n,a}|$, and $|V_n| - |V_{n-1}| = \sum_{y \in \mathcal{Y} \backslash \{a\}} |V_{n,y}|$. Defining
$$E_n^n := \{(e_{i,f}, i) \in E_n : i = n\},$$

we have
$$\sum_{e \in E_n^n} (|e| - 1) = |V_n| - |V_{n-1}|.$$

For any $y \in \mathcal{Y}$, we define
$$E_{n,y} := \{(e_{i,f}, i) \in E_n : f(n) = y, i \in [n-1]\}$$

and let $E'_{n,y}$ denote the edge set in $\mathcal{G}(V_{n,y})$; for any $e = (e_{i,f}, i) \in E_{n,y}$, we define
$$s_1(e) := |\{\mathbf{y} \in e_{i,f} : \mathbf{y} \in V_{n,y}\}| \text{ and } s_2(e) := |\{\mathbf{y} \in e_{i,f} : \mathbf{y} \notin V_{n,y}\}|.$$

Then, we have $|e| = s_1(e) + s_2(e)$, $E_n = (\cup_{y \in \mathcal{Y}} E_{n,y}) \cup E_n^n$, and
$$\sum_{e \in E_{n-1}} (|e| - 1)$$
$$\geqslant \sum_{e \in E_{n,a}} (|e| - 1) + \sum_{y \in \mathcal{Y} \backslash \{a\}} \left( \sum_{e \in E_{n,y} : s_1(e) = 0} (s_2(e) - 1) + \sum_{e \in E_{n,y} : s_1(e) \geqslant 1} s_2(e) \right).$$

Note that by the induction hypothesis, we have
$$\sum_{e \in E_{n-1}} (|e| - 1) \leqslant d|V_{n-1}|$$

and by definition, we also have
$$\sum_{e \in E'_{n,y}} (|e| - 1) = \sum_{e \in E_{n,y} : s_1(e) \geqslant 2} (s_1(e) - 1).$$

We claim that $\dim(V_{n,y}) \leqslant d - 1$ for all $y \in \mathcal{Y} \backslash \{a\}$. Suppose on the contrary that $\dim(V_{n,y}) > d - 1$. Since $\dim(V_n) = d$, we have $\dim(V_{n,y}) = d$ and there exists a set $\mathbf{i} := \{i_1, \ldots, i_d\} \subseteq [n]$ with $i_1 < i_2 < \cdots < i_d$ such that $V_{n,y}|_{\mathbf{i}}$ contains a pseudo-cube $H_y$ of dimension $d$. Since for any $(y_1, \ldots, y_n) \in V_{n,y}$, we have $y_n = y$, it must hold that $i_d \leqslant n - 1$. Now, we can define
$$H_{y,a} := \{(y_{i_1}, \ldots, y_{i_d}, a) \in \mathcal{Y}^{d+1} : (y_{i_1}, \ldots, y_{i_d}) \in H_y\}$$
$$\bigcup \{(y_{i_1}, \ldots, y_{i_d}, y) \in \mathcal{Y}^{d+1} : (y_{i_1}, \ldots, y_{i_d}) \in H_y\}.$$

By the assumption that $|V'_{n-1}| = |V_{n,a}|$, we have $|H_{y,a}| = 2|H_y|$ and $H_{y,a} \subseteq V_n|_{(i_1, \ldots, i_d, n)}$. For any $k \in [d]$ and any $(y_{i_1}, \ldots, y_{i_d}, y), (y_{i_1}, \ldots, y_{i_d}, a) \in H_{y,a}$, since there exists an $k$-neighbor of $(y_{i_1}, \ldots, y_{i_d})$ denoted by $(y_{i_1}, \ldots, y'_{i_k}, \ldots, y_{i_d})$ in $H_y$, $(y_{i_1}, \ldots, y'_{i_k}, \ldots, y_{i_d}, y') \in H_{y,a}$ is a $k$-neighbor of $(y_{i_1}, \ldots, y_{i_d}, y')$ in $H_{y,a}$ for $y' = y, a$. Moreover, $(y_{i_1}, \ldots, y_{i_d}, y)$ is a $(d+1)$-neighbor of $(y_{i_1}, \ldots, y_{i_d}, a)$ in $H_{y,a}$ and vise-versa. Thus, $H_{y,a}$ is a pseudo-cube of dimension $d + 1$, which contradicts the assumption that $\dim(V_n) \leqslant d$. Therefore, we must have $\dim(V_{n,y}) \leqslant d - 1$. Then, by the induction hypothesis, we have for any $y \in \mathcal{Y} \backslash \{a\}$,
$$\sum_{e \in E'_{n,y}} (|e| - 1) \leqslant (d - 1)|V_{n,y}|.$$

Summarizing the results above, we have

$$\sum_{e \in E_n} (|e| - 1)$$

$$= \sum_{e \in E_n^n} (|e| - 1) + \sum_{y \in \mathcal{Y}} \sum_{e \in E_{n,y}} (|e| - 1)$$

$$= \sum_{e \in E_n^n} (|e| - 1) + \sum_{e \in E_{n,a}} (|e| - 1) +$$

$$\sum_{y \in \mathcal{Y} \setminus \{a\}} \left( \sum_{e \in E_{n,y}: s_1(e)=0} (s_2(e) - 1) + \sum_{e \in E_{n,y}: s_1(e)=1} s_2(e) + \sum_{e \in E_{n,y}: s_1(e) \geqslant 2} (s_2(e) + (s_1(e) - 1)) \right)$$

$$= \sum_{e \in E_n^n} (|e| - 1) + \sum_{e \in E_{n,a}} (|e| - 1) + \sum_{y \in \mathcal{Y} \setminus \{a\}} \left( \sum_{e \in E_{n,y}: s_1(e)=0} (s_2(e) - 1) + \sum_{e \in E_{n,y}: s_1(e) \geqslant 1} s_2(e) \right)$$

$$+ \sum_{y \in \mathcal{Y} \setminus \{a\}} \sum_{e \in E_{n,y}: s_1(e) \geqslant 2} (s_1(e) - 1)$$

$$\leqslant |V_n| - |V_{n-1}| + \sum_{e \in E_{n-1}} (|e| - 1) + \sum_{y \in \mathcal{Y} \setminus \{a\}} \sum_{e \in E_{n,y}'} (|e| - 1)$$

$$\leqslant |V_n| - |V_{n-1}| + d|V_{n-1}| + \sum_{y \in \mathcal{Y} \setminus \{a\}} (d - 1)|V_{n,y}|$$

$$= |V_n| - |V_{n-1}| + d|V_{n-1}| + (d - 1)(|V_n| - |V_{n-1}|)$$

$$= d|V_n|.$$

$\square$

*Proof of Lemma 3.8.* Consider arbitrary $n \in \mathbb{N} \setminus \{1\}$ and $V_n \subseteq \mathcal{Y}^n$. By the definition of $\mathfrak{P}_a(V_n)$, we have $|V_n^\gamma| = |V_n|$. Let $E_n$ denote the edge set of $\mathcal{G}(V_n)$ and $E_{n,\gamma}$ denote the edge set of $\mathcal{G}(V_n^\gamma)$. It suffices to prove that $\sum_{e \in E_n} (|e| - 1) \leqslant \sum_{e \in E_{n,\gamma}} (|e| - 1)$. Define

$$E_n^i := \{(e_{k,f}, k) \in E_n : k = i\} \quad \text{and} \quad E_{n,\gamma}^i := \{(e_{k,f}, k) \in E_{n,\gamma} : k = i\}$$

for all $i \in [n]$. By the definition of $V_n^\gamma$, we have

$$\sum_{e \in E_n^n} (|e| - 1) = \sum_{e \in E_{n,\gamma}^n} (|e| - 1). \tag{23}$$

For any $i \in [n]$ and $f : [n] \setminus \{i\} \to \mathcal{Y}$, we define

$$e_{i,f} := \{(y_1, \ldots, y_n) \in V_n : y_k = f(k) \; \forall k \in [n] \setminus \{i\}\} \quad \text{and}$$
$$e_{i,f}^\gamma := \{(y_1, \ldots, y_n) \in V_n^\gamma : y_k = f(k) \; \forall k \in [n] \setminus \{i\}\}$$

to distinguish edges in $E_n$ and $E_{n,\gamma}$. For any $y \in \mathcal{Y}$, $i \in [n-1]$, and $f : [n-1] \setminus \{i\} \to \mathcal{Y}$, define $f_y : [n] \setminus \{i\} \to \mathcal{Y}$ such that $f_y|_{[n-1] \setminus \{i\}} = f$ and $f_y(n) = y$. Then, we have

$$\sum_{y \in \mathcal{Y} \setminus \{a\} : (e_{i,f_y}, i) \in E_n} (|e_{i,f_y}| - 1) - \sum_{y \in \mathcal{Y} \setminus \{a\} : (e_{i,f_y}^\gamma, i) \in E_{n,\gamma}} (|e_{i,f_y}^\gamma| - 1)$$

$$\leqslant \mathbb{1}_{(e_{i,f_a}^\gamma, i) \in E_{n,\gamma}} (|e_{i,f_a}^\gamma| - 1) - \mathbb{1}_{(e_{i,f_a}, i) \in E_n} (|e_{i,f_a}| - 1)$$

which implies that

$$\sum_{i=1}^{n-1} \sum_{e \in E_n^i} (|e| - 1) \leqslant \sum_{i=1}^{n-1} \sum_{e \in E_{n,\gamma}^i} (|e| - 1).$$

and from (23),

$$\sum_{e \in E_n} (|e| - 1) \leqslant \sum_{e \in E_{n,\gamma}} (|e| - 1).$$

Thus, we can conclude that $\mathsf{avgoutdeg}(\mathcal{G}(V_n^\gamma)) \geqslant \mathsf{avgoutdeg}(\mathcal{G}(V_n))$ for any $\gamma \in \Gamma_{a,V_n}$. $\square$

# H Lemmas regarding graph dimension

In this section, we provide the technical lemmas on learning with finite graph dimension and bounding the graph dimension of certain classes. Those lemmas are used in the proof of Theorem 2.7.

## H.1 Learning algorithm for classes with finite graph dimension

We first provide the definition of graph dimension.

**Definition H.1** (Graph dimension)**.** *For $\mathcal{H} \subseteq \mathcal{Y}^{\mathcal{X}}$ and $n \in \mathbb{N}$, $\mathbf{x} = (x_1, \dots, x_n) \in \mathcal{X}^n$ is said to be* ***G-shattered*** *by $\mathcal{H}$ is there exists $f : [n] \to \mathcal{Y}$ such that for any $\mathbf{i} \subseteq [n]$, there exists $g \in \mathcal{H}$ satisfying $g(x_i) = f(i)$ for all $i \in \mathbf{i}$ and $g(x_i) \neq f(i)$ for all $i \in [n] \backslash \mathbf{i}$. The* ***graph dimension*** *of $\mathcal{H}$, denoted as $\dim_G(\mathcal{H})$ is the maximum size of a G-shattered sequence.*

Define $\mathrm{Log} : [0, \infty) \to [1, \infty)$, $x \mapsto \log(x \vee e)$ where $x \vee e = \max\{x, e\}$. For any $\mathcal{H} \subseteq \mathcal{Y}^{\mathcal{X}}$, $n \in [\dim_G(\mathcal{H})]$, and $\mathbf{x} = \{x_1, \dots, x_n\} \subseteq \mathcal{X}$ that is G-shattered by $\mathcal{H}$, there exists $f : [n] \to \mathcal{Y}$ such that for any $\mathbf{i} \subseteq [n]$, there exists $g \in \mathcal{H}$ satisfying $g(x_i) = f(i)$ for all $i \in \mathbf{i}$ and $g(x_i) \neq f(i)$ for all $i \in [n] \backslash \mathbf{i}$. Thus, we can define

$$\mathcal{H}(\mathbf{x}) := \left\{ h : [n] \to \{0, 1\}, i \mapsto \mathbb{1}_{\widetilde{h}(x_i) = f(i)} \big| \, \widetilde{h} \in \mathcal{H} \right\}.$$

For general $n \in \mathbb{N}$, $\mathbf{x} \subseteq \mathcal{X}$ with $|\mathbf{x}| = n$, and $f \in \mathcal{Y}^n$, we define

$$\mathcal{H}_f(\mathbf{x}) := \left\{ h : [n] \to \{0, 1\}, i \mapsto \mathbb{1}_{\widetilde{h}(x_i) = f(i)} \big| \, \widetilde{h} \in \mathcal{H} \right\} \text{ and}$$

$$\tau_{\mathcal{H}}(n) := \sup_{\mathbf{x} \subseteq \mathcal{X} : |\mathbf{x}| = n} \sup_{f \in \mathcal{Y}^n} |\mathcal{H}_f(\mathbf{x})|.$$

Note that $\tau_{\mathcal{H}}(n) = |\mathcal{H}(\mathbf{x})| = 2^n$ for any $n \in [\dim_G(\mathcal{H})]$. We have the following lemma.

**Lemma H.2.** *For any $\mathcal{H} \subseteq \mathcal{Y}^{\mathcal{X}}$ with $\dim_G(\mathcal{H}) = d$, we have $\tau_{\mathcal{H}}(n) \leq \sum_{i=0}^{d} \binom{n}{i}$. In particular, if $n \geq d$, then $\tau_{\mathcal{H}}(n) \leq (en/d)^d$.*

*Proof.* We first prove by induction on $n$ that for any $\mathbf{x} = \{x_1, \dots, x_n\} \subseteq \mathcal{X}$ and $f \in \mathcal{Y}^n$,

$$|\mathcal{H}_f(\mathbf{x})| \leq |\{\mathbf{x}' \subseteq \mathbf{x} : \mathcal{H} \text{ G-shatters } \mathbf{x}'\}|. \tag{24}$$

For $n = 1$, it is obviously that $|\mathcal{H}_f(\mathbf{x})| \leq 2$ and $|\{\mathbf{x}' \subseteq \mathbf{x} : \mathcal{H} \text{ G-shatters } \mathbf{x}'\}| \geq 1$. If $|\{\mathbf{x}' \subseteq \mathbf{x} : \mathcal{H} \text{ G-shatters } \mathbf{x}'\}| = 1$, then $\mathbf{x}$ is not G-shattered by $\mathcal{H}$, which implies that $|\mathcal{H}_f(\mathbf{x})| \leq 1$. Thus, (24) holds. Now, suppose that (24) holds for any $k < n$. Consider $\bar{\mathbf{x}} := \{x_2, \dots, x_n\}$,

$$Y^0 := \{(y_2, \dots, y_n) \in \{0, 1\}^{n-1} : (0, y_2, \dots, y_n) \in \mathcal{H}_f(\mathbf{x}) \text{ or } (1, y_2, \dots, y_n) \in \mathcal{H}_f(\mathbf{x})\}, \text{ and}$$

$$Y^1 := \{(y_2, \dots, y_n) \in \{0, 1\}^{n-1} : (0, y_2, \dots, y_n) \in \mathcal{H}_f(\mathbf{x}) \text{ and } (1, y_2, \dots, y_n) \in \mathcal{H}_f(\mathbf{x})\}.$$

We have $|\mathcal{H}_f(\mathbf{x})| = |Y^0| + |Y^1|$ and $|Y^0| = |\mathcal{H}_f(\bar{\mathbf{x}})|$. Then, by the induction hypothesis, we have

$$|Y^0| \leq |\{\mathbf{x}' \subseteq \bar{\mathbf{x}} : \mathcal{H} \text{ G-shatters } \mathbf{x}'\}| = |\{\mathbf{x}' \subseteq \mathbf{x} : x_1 \notin \mathbf{x}' \text{ and } \mathcal{H} \text{ G-shatters } \mathbf{x}'\}|.$$

For any $y \in \mathcal{Y}$, define

$$\mathcal{H}^y := \{h \in \mathcal{H} : \exists h' \in \mathcal{H} \text{ s.t. } h|_{\mathbf{x}} \text{ and } h'|_{\mathbf{x}} \text{ differs only at } 1 \text{ and } y \in \{h(x_1), h'(x_1)\}\}.$$

Then, $\mathcal{H}^y$ G-shatters $\mathbf{x}' \subseteq \bar{\mathbf{x}}$ implies that $\mathcal{H}$ G-shatters $\mathbf{x}' \cup \{x_1\}$. We also have $Y^1 = \mathcal{H}_f^{f(1)}(\bar{\mathbf{x}})$. It follows from the induction hypothesis that

$$|Y^1| = |\mathcal{H}_f^{f(1)}(\bar{\mathbf{x}})| \leq |\{\mathbf{x}' \subseteq \bar{\mathbf{x}} : \mathcal{H}^{f(1)} \text{ G-shatters } \mathbf{x}'\}| \leq |\{\mathbf{x}' \subseteq \mathbf{x} : x_1 \in \mathbf{x}' \text{ and } \mathcal{H} \text{ G-shatters } \mathbf{x}'\}|.$$

In conclusion, we have

$$\begin{aligned}
&|\mathcal{H}_f(\mathbf{x})| \\
=&|Y^0| + |Y^1| \\
\leq&|\{\mathbf{x}' \subseteq \mathbf{x} : x_1 \notin \mathbf{x}' \text{ and } \mathcal{H} \text{ G-shatters } \mathbf{x}'\}| + |\{\mathbf{x}' \subseteq \mathbf{x} : x_1 \in \mathbf{x}' \text{ and } \mathcal{H} \text{ G-shatters } \mathbf{x}'\}| \\
=&|\{\mathbf{x}' \subseteq \mathbf{x} : \mathcal{H} \text{ G-shatters } \mathbf{x}'\}|,
\end{aligned}$$

which is exactly (24). By (24), we have

$$\tau_{\mathcal{H}}(n) \leqslant |\{\mathbf{x}' \subseteq \mathbf{x} : \mathcal{H} \text{ G-shatters } \mathbf{x}'\}| \leqslant \sum_{i=0}^{d} \binom{n}{i}.$$

$\square$

For any (measurable) classifier $h : \mathcal{X} \to \mathcal{Y}$, define

$$\mathrm{ER}(h) := \{(x, y) \in \mathcal{X} \times \mathcal{Y} : h(x) \neq y\}.$$

Then, for any probability measure $\mathcal{D}$ over $\mathcal{X} \times \mathcal{Y}$, we can define

$$\mathrm{er}_{\mathcal{D}}(h) := \mathcal{D}(\mathrm{ER}(h)) = \mathbb{P}_{(X,Y)\sim\mathcal{D}}(h(X) \neq Y).$$

**Definition H.3.** $S \subseteq \mathcal{X} \times \mathcal{Y}$ *is said to be a an $\varepsilon$-net ($\varepsilon \in (0,1)$) for $\mathcal{H} \subseteq \mathcal{Y}^{\mathcal{X}}$ with respect to a distribution $\mathcal{D}$ over $\mathcal{X} \times \mathcal{Y}$ if for any $h \in \mathcal{H}$,*

$$\mathrm{er}_{\mathcal{D}}(h) \geqslant \varepsilon \implies \mathrm{ER}(h) \cap S = \{(x, y) \in S : h(x) \neq y\} \neq \varnothing.$$

For any integer $n \in \mathbb{N}$, set $\mathcal{Z}$, and $T = (z_1, \ldots, z_n) \in \mathcal{Z}^n$, we say $z \in T$ if $z = z_i$ for some $i \in [n]$ and use $|T|$ to denote the length of the sequence $T$. For notational convenience, we use $\varnothing$ to also denote an empty sequence (a sequence of length 0). For any subset $E \subseteq \mathcal{Z}$, we use $T \cap E = E \cap T$ to denote the subsequence of $T$ consisting of all elements in $E$, i.e., for $I := \{i \in [n] : z_i \in E\}$,

$$T \cap E = E \cap T = (z_i)_{i \in I}.$$

Then, we have $|T \cap E| = |E \cap T| = |I|$.

**Proposition H.4.** *For any $\mathcal{H} \subseteq \mathcal{Y}^{\mathcal{X}}$ with $\dim_G(\mathcal{H}) = d$, any $\mathcal{H}$-realizable distribution $\mathcal{D}$, any $\delta \in (0,1]$, any $n \in \mathbb{N}$, and any ERM algorithm $A$, consider $S_n \sim \mathcal{D}^n$. With probability at least $1 - \delta$, we have*

$$\mathrm{er}_{\mathcal{D}}(A(S_n, \mathcal{H})) \leqslant \frac{2}{n} \left[ d \vee \left( d \log_2 \left( \frac{2en}{d} \right) \right) + \log_2 \left( \frac{2}{\delta} \right) \right].$$

*Proof.* For any $n \geqslant 2$ and $\varepsilon \in [2/n, 1]$, define

$$B := \{S \in (\mathcal{X} \times \mathcal{Y})^n : \exists h \in \mathcal{H} \text{ s.t. } \mathrm{er}_{\mathcal{D}}(h) \geqslant \varepsilon \text{ and } \mathrm{ER}(h) \cap S = \varnothing\} \quad \text{and}$$

$$B' := \big\{(S, T) \in (\mathcal{X} \times \mathcal{Y})^{2n} : |S| = |T| = |n|, \exists h \in \mathcal{H} \text{ s.t. }$$
$$\mathrm{er}_{\mathcal{D}}(h) \geqslant \varepsilon, \ \mathrm{ER}(h) \cap S = \varnothing, \text{ and } |\mathrm{ER}(h) \cap T| > \varepsilon n/2\big\}.$$

Let $(S, T) \sim \mathcal{D}^{2n}$ with $S, T \in (\mathcal{X} \times \mathcal{Y})^n$. Since $(S, T) \in B'$ implies that $S \in B$, we have

$$\mathbb{P}((S, T) \in B') = \mathbb{E}[\mathbb{1}_{(S,T)\in B'} \mathbb{1}_{S\in B}] = \mathbb{E}[\mathbb{1}_{S\in B} \mathbb{P}((S, T) \in B'|S)].$$

On $S \in B$, there exists $h \in \mathcal{H}$ such that $\mathrm{er}_{\mathcal{D}}(h) \geqslant \varepsilon$ and $\mathrm{ER}(h) \cap S = \varnothing$. Then, $|\mathrm{ER}(h) \cap T| > \varepsilon n/2$ implies that $(S, T) \in B'$. It follows that

$$\mathbb{1}_{S\in B} \mathbb{P}((S, T) \in B'|S) \geqslant \mathbb{1}_{S\in B} \mathbb{P}(|\mathrm{ER}(h) \cap T| > \varepsilon n/2|S).$$

Since $T$ is independent of $S$, $h$ is determined by $S$, and $\mathcal{D}(\mathrm{ER}(h)) = \mathrm{er}_{\mathcal{D}}(h) \geqslant \varepsilon$ on $S \in B$, we know that on $S \in B$, $|\mathrm{ER}(h) \cap T|$ follows the Binomial distribution $B(n, \mathrm{er}_{\mathcal{D}}(h))$ conditional on $S$, and

$$\mathbb{1}_{S\in B} \mathbb{E}[|\mathrm{ER}(h) \cap T||S] = \mathrm{er}_{\mathcal{D}}(h) n \mathbb{1}_{S\in B} \geqslant \varepsilon n \mathbb{1}_{S\in B}.$$

Thus, by Lemma H.8, since $n\varepsilon \geqslant 2$, we have

$$\mathbb{1}_{S\in B} \mathbb{P}(|\mathrm{ER}(h) \cap T| \leqslant \varepsilon n/2|S) \leqslant \mathbb{P}(|\mathrm{ER}(h) \cap T| \leqslant \mathrm{er}_{\mathcal{D}}(h) n/2|S) \mathbb{1}_{S\in B} < \frac{1}{2} \mathbb{1}_{S\in B},$$

which implies that

$$\mathbb{P}((S, T) \in B') > \frac{1}{2} \mathbb{P}(S \in B).$$

By the definition of $B'$, we have

$$\mathbb{P}((S,T) \in B') = \mathbb{E}\left[\sup_{h\in\mathcal{H}} \mathbb{1}_{\mathrm{er}_{\mathcal{D}}(h)\geqslant\varepsilon} \mathbb{1}_{\mathrm{ER}(h)\cap S=\varnothing} \mathbb{1}_{|\mathrm{ER}(h)\cap T|>\varepsilon n/2}\right]$$

$$= \mathbb{E}\left[\sup_{h\in\mathcal{H}} \mathbb{1}_{\mathrm{er}_{\mathcal{D}}(h)\geqslant\varepsilon} \mathbb{1}_{\mathrm{ER}(h)\cap S=\varnothing} \mathbb{1}_{|\mathrm{ER}(h)\cap(S,T)|>\varepsilon n/2}\right]$$

$$\leqslant \mathbb{E}\left[\sup_{h\in\mathcal{H}} \mathbb{1}_{\mathrm{ER}(h)\cap S=\varnothing} \mathbb{1}_{|\mathrm{ER}(h)\cap(S,T)|>\varepsilon n/2}\right].$$

For any $m \in \mathbb{N}$, $\mathbf{r} = ((x_1,y_1),\dots,(x_m,y_m)) \in (\mathcal{X}\times\mathcal{Y})^m$, and $h \in \mathcal{Y}^{\mathcal{X}}$, we define

$$h_{\mathbf{r}} : [m] \to \{0,1\},\ i \mapsto \mathbb{1}_{h(x_i)=y_i}$$

and $\mathcal{H}_{\mathbf{r}} := \{h_{\mathbf{r}} : h \in \mathcal{H}\}$. Note that $\mathcal{H}_{\mathbf{r}} = \mathcal{H}_{(y_1,\dots,y_m)}((x_1,\dots,x_m))$, which implies that

$$|\mathcal{H}_{\mathbf{r}}| \leqslant \tau_{\mathcal{H}}(m). \tag{25}$$

For any $k \in [m]$ and $1 \leqslant i_1 < \cdots < i_k \leqslant m$, we use $\mathbf{r}_{(i_1,\dots,i_k)}$ to denote a permutation of $\mathbf{r}$ where $(x_j,y_j)$ appears in the $i_j$-th position for all $j \in [k]$, specifically, $\mathbf{r}_{\{i_1,\dots,i_k\}} = (x_{\sigma(i)}, y_{\sigma(i)})_{i\in[n]}$ where $\sigma(i_j) := j$ for all $j \in [k]$ and $(\sigma(l))_{l\in[n]\setminus\{i_1,\dots,i_k\}} := (k+1,\dots,m)$. Then, for any $\mathbf{i} \subseteq [2n]$ with $|\mathbf{i}| = n$, we have

$$\sup_{h\in\mathcal{H}} \mathbb{1}_{\mathrm{ER}(h)\cap S=\varnothing} \mathbb{1}_{|\mathrm{ER}(h)\cap(S,T)|>\varepsilon n/2} \leqslant \sum_{h\in\mathcal{H}_{(S,T)}} \mathbb{1}_{h(i)=1,\ \forall i\in[n]} \mathbb{1}_{\sum_{i\in[2n]} h(i)<(2-\varepsilon/2)n}$$

$$= \sum_{h\in\mathcal{H}_{(S,T)_{\mathbf{i}}}} \mathbb{1}_{h(i)=1,\ \forall i\in\mathbf{i}} \mathbb{1}_{\sum_{i\in[2n]} h(i)<(2-\varepsilon/2)n}.$$

Since $(S,T) \sim \mathcal{D}^{2n}$, we also have

$$\mathbb{E}\left[\sum_{h\in\mathcal{H}_{(S,T)_{\mathbf{i}}}} \mathbb{1}_{h(i)=1,\ \forall i\in\mathbf{i}} \mathbb{1}_{\sum_{i\in[2n]} h(i)<(2-\varepsilon/2)n}\right] = \mathbb{E}\left[\sum_{h\in\mathcal{H}_{(S,T)}} \mathbb{1}_{h(i)=1,\ \forall i\in\mathbf{i}} \mathbb{1}_{\sum_{i\in[2n]} h(i)<(2-\varepsilon/2)n}\right].$$

Thus,

$$\mathbb{E}\left[\sup_{h\in\mathcal{H}} \mathbb{1}_{\mathrm{ER}(h)\cap S=\varnothing} \mathbb{1}_{|\mathrm{ER}(h)\cap(S,T)|>\varepsilon n/2}\right]$$

$$\leqslant \frac{1}{\binom{2n}{n}} \sum_{\mathbf{i}\subseteq[2n]:|\mathbf{i}|=n} \mathbb{E}\left[\sum_{h\in\mathcal{H}_{(S,T)}} \mathbb{1}_{h(i)=1,\ \forall i\in\mathbf{i}} \mathbb{1}_{\sum_{i\in[2n]} h(i)<(2-\varepsilon/2)n}\right]$$

$$= \mathbb{E}\left[\sum_{h\in\mathcal{H}_{(S,T)}} \mathbb{1}_{\sum_{i\in[2n]} h(i)<(2-\varepsilon/2)n} \frac{1}{\binom{2n}{n}} \sum_{\mathbf{i}\subseteq[2n]:|\mathbf{i}|=n} \mathbb{1}_{h(i)=1,\ \forall i\in\mathbf{i}}\right]$$

$$\leqslant \mathbb{E}\left[\sum_{h\in\mathcal{H}_{(S,T)}} \mathbb{1}_{\sum_{i\in[2n]} h(i)<(2-\varepsilon/2)n} \frac{\binom{\lfloor(2-\varepsilon/2)n\rfloor}{n}}{\binom{2n}{n}}\right]$$

$$\leqslant 2^{-\varepsilon n/2}\mathbb{E}\left[\sum_{h\in\mathcal{H}_{(S,T)}} \mathbb{1}_{\sum_{i\in[2n]} h(i)<(2-\varepsilon/2)n}\right] \tag{26}$$

$$\leqslant 2^{-\varepsilon n/2}\mathbb{E}\left[|\mathcal{H}_{(S,T)}|\right]$$

$$\leqslant 2^{-\varepsilon n/2}\tau_{\mathcal{H}}(2n), \tag{27}$$

where (26) follows from Lemma H.7 and (27) follows from (25). Finally, we have proved that

$$\mathcal{D}^n(B) = \mathbb{P}(S \in B) < 2\mathbb{P}((S,T) \in B') \leqslant 2\tau_{\mathcal{H}}(2n)2^{-\varepsilon n/2}. \tag{28}$$

Since $\mathcal{D}$ is $\mathcal{H}$-realizable and $A$ is an ERM algorithm, we must have $A(S_n,\mathcal{H}) \in \mathcal{H}$ and

$$\mathrm{ER}(A(S_n,\mathcal{H})) \cap S_n = \varnothing$$

almost surely. Moreover, by the definition of $B$, if $S_n \notin B$, then $\mathrm{ER}(A(S_n, \mathcal{H})) \cap S_n = \varnothing$ implies that $\mathrm{er}_{\mathcal{D}}(A(S_n, \mathcal{H})) < \varepsilon$. Thus, we have

$$\mathbb{P}\left(\mathrm{er}_{\mathcal{D}}(A(S_n, \mathcal{H})) < \varepsilon\right) \geqslant \mathbb{P}(S_n \notin B) = 1 - \mathcal{D}^n(B).$$

Solving $2\tau_{\mathcal{H}}(2n)2^{-\varepsilon n/2} = \delta$, we get

$$\varepsilon = 1 \wedge \left[\frac{2}{n}\left(\log_2(\tau_{\mathcal{H}}(2n)) + \log_2\left(\frac{2}{\delta}\right)\right)\right] \leqslant \frac{2}{n}\left[d \vee \left(d\log_2\left(\frac{2en}{d}\right)\right) + \log_2\left(\frac{2}{\delta}\right)\right]$$

where the last inequality follows from Lemma H.2. Note that $1 \wedge \left[\frac{2}{n}\left(\log_2(\tau_{\mathcal{H}}(2n)) + \log_2\left(\frac{2}{\delta}\right)\right)\right] \geqslant \frac{2}{n}$, which implies that the above choice of $\varepsilon$ is legitimate. Applying (28), we can conclude that with probability at least $1 - \delta$,

$$\mathrm{er}_{\mathcal{D}}(A(S_n, \mathcal{H})) < \frac{2}{n}\left[d \vee \left(d\log_2\left(\frac{2en}{d}\right)\right) + \log_2\left(\frac{2}{\delta}\right)\right].$$

$\square$

**Proposition H.5.** *There exists a learning algorithm $A_G$ such that for any $\mathcal{H} \subseteq \mathcal{Y}^{\mathcal{X}}$ with $\dim_G(\mathcal{H}) = d$, any $\mathcal{H}$-realizable distribution $\mathcal{D}$, any $\delta \in (0, 1]$, and any $n \in \mathbb{N}$, given $S_n \sim \mathcal{D}^n$, it holds with probability at least $1 - \delta$ that*

$$\mathrm{er}_{\mathcal{D}}(A_G(S_n, \mathcal{H})) = O\left(\frac{1}{n}\left(d + \mathrm{Log}\left(\frac{1}{\delta}\right)\right)\right). \tag{29}$$

*Proof.* The algorithm $A_G$ is the algorithm $S_n \mapsto \mathrm{Majority}(\mathrm{ERM}_{\mathcal{H}}(\mathbb{A}(S_n; \varnothing)))$ defined in Hanneke [2016], where $\mathrm{ERM}_{\mathcal{H}}$ denotes an ERM algorithm on the concept class $\mathcal{H}$. Applying the error rate of ERM algorithms proved in Proposition H.4 in the proof of Hanneke [2016, Theorem 2], we establish (29). $\square$

The above proposition immediately implies the following corollary on the expected error rate of the learning algorithm $A_G$.

**Corollary H.6.** *There exists a learning algorithm $A_G$ such that for any $\mathcal{H} \subseteq \mathcal{Y}^{\mathcal{X}}$ with $\dim_G(\mathcal{H}) = d$, any $\mathcal{H}$-realizable distribution $\mathcal{D}$, and any $n \in \mathbb{N}$ it holds that*

$$\mathbb{E}_{S_n \sim \mathcal{D}^n}\left[\mathrm{er}_{\mathcal{D}}(A_G(S_n, \mathcal{H}))\right] = \mathbb{E}_{((S_n, (X, Y)) \sim \mathcal{D}^{n+1}}\left[A_G(S_n, \mathcal{H})(X) \neq Y\right] = O\left(\frac{d}{n}\right). \tag{30}$$

*Proof.* According to Proposition H.5, there exists some constant $C > 0$ such that for any $\delta \in (0, 1]$, it holds with probability at least $1 - \delta$ that

$$\mathrm{er}_{\mathcal{D}}(A_G(S_n, \mathcal{H})) \leqslant \frac{C}{n}\left(d + \log\left(\frac{1}{\delta} \vee e\right)\right),$$

which implies that for any $t \geqslant \frac{C(d+1)}{n}$,

$$\mathbb{P}\left(\mathrm{er}_{\mathcal{D}}(A_G(S_n, \mathcal{H})) > t\right) \leqslant e^{-\frac{nt}{C} + d}.$$

Since $\mathrm{er}_{\mathcal{D}}(A_G(S_n, \mathcal{H}))$ is nonnegative, we have

$$\begin{aligned}
\mathbb{E}\left[\mathrm{er}_{\mathcal{D}}(A_G(S_n, \mathcal{H}))\right] &= \int_0^{\infty} \mathbb{P}\left(\mathrm{er}_{\mathcal{D}}(A_G(S_n, \mathcal{H})) > t\right) dt \\
&\leqslant \frac{C(d+1)}{n} + \int_{\frac{C(d+1)}{n}}^{\infty} e^{-\frac{nt}{C} + d} dt \\
&= \frac{C(d + 1 + e^{-1})}{n} \\
&= O\left(\frac{d}{n}\right).
\end{aligned}$$

$\square$

**Lemma H.7.** *For any $n \in \mathbb{N}$ and $m \in \mathbb{N} \cap [n, 2n]$, we have*

$$\frac{\binom{m}{n}}{\binom{2n}{n}} \leqslant 2^{m-2n}. \tag{31}$$

*Proof.* Note that

$$\frac{\binom{m}{n}}{\binom{2n}{n}} = \frac{m(m-1)\cdots(m-n+1)}{2n(2n-1)\cdots(n+1)}.$$

We prove by induction on $m$. When $m = 2n$, we have

$$\frac{\binom{m}{n}}{\binom{2n}{n}} = 1 = 2^{m-2n}.$$

Suppose that (31) holds for some $m \in \mathbb{N} \cap [n+1, 2n]$. Then, we have

$$\frac{\binom{m-1}{n}}{\binom{2n}{n}} = \frac{m-n}{m} \frac{\binom{m}{n}}{\binom{2n}{n}} \leqslant \frac{1}{2} \cdot 2^{m-2n} = 2^{m-1-2n}.$$

Thus, by induction, (31) holds for any $m \in \mathbb{N} \cap [n, 2n]$. $\qquad\square$

**Lemma H.8.** *For $X \sim B(n, p)$, if $np \geqslant 2$, then*

$$\mathbb{P}(X \leqslant np/2) < 1/2.$$

*Proof.* If $np > 8$, since $\mathbb{E}[X] = np$, by the multiplicative Chernoff bound, we have

$$\mathbb{P}(X \leqslant np/2) \leqslant e^{-np/8} < e^{-1} \leqslant 1/2.$$

For $2 \leqslant np \leqslant 8$, we have

$$\mathbb{P}(X \leqslant np/2) = \sum_{i=0}^{\lfloor np/2 \rfloor} \binom{n}{i} p^i (1-p)^{n-i}.$$

For $6 \leqslant pn \leqslant 8$, we have $6/n \leqslant p \leqslant 8/n$ and $n \geqslant 6$. Thus,

$$\mathbb{P}(X \leqslant np/2) = \sum_{i=0}^{3} \binom{n}{i} p^i (1-p)^{n-i}.$$

Consider

$$f_3(x, p) := \log(x(x-1)(x-2)) + (x-3)\log(1-p), \ x \geqslant 6, \ 1 \geqslant p \geqslant 6/x.$$

Fixing $p \in (0, 1]$, we have $x \geqslant 6/p$ and for $x \geqslant 6/p$,

$$\frac{\partial}{\partial x} f_3(x, p) = \frac{1}{x} + \frac{1}{x-1} + \frac{1}{x-2} + \log(1-p) \leqslant \frac{1}{x} + \frac{1}{x-1} + \frac{1}{x-2} - p$$

$$\leqslant \frac{p}{6} + \frac{p}{6-p} + \frac{p}{6-2p} - p < 0.$$

Thus, fixing $p \in (0, 1]$, $f_3(\cdot, p)$ is a decreasing function on $[6/p, \infty)$. Therefore, we have

$$f_3(x, p) \leqslant f_3(6/p, p),$$

which implies that

$$g_3(x, p) \leqslant g_3(6/p, p)$$

for

$$g_3(x, p) := \frac{x(x-1)(x-2)}{6} p^3 (1-p)^{x-3}, \ p \in (0, 1], \ x \geqslant 6/p.$$

Since $\log z - z + 1 \leqslant 0$ for $z \in (0, 1]$ and $g'(z) = \log z - z + 1$ for $g(z) := z \log(z) + 1 - z - \frac{1}{2}(1-z)^2$ defined on $z \in (0, 1]$, we have

$$0 = g(1) \leqslant g(z) \leqslant \lim_{z \to 0_+} g(z) = 1/2$$

for $z \in (0, 1]$. Thus, we have

$$\log(z) + \frac{(1-z)(1 - \frac{1-z}{2})}{z} \geqslant 0.$$

Plugging in $z = 1 - 6/t$ for $t > 6$, we have

$$\log(1 - 6/t) + \frac{6/t(1 - 3/t)}{1 - 6/t} \geqslant 0.$$

Then, defining

$$f(t) := \log(t) + \log(t-1) + \log(t-2) - 3\log(t) + (t-3)\log(1 - 6/t), \quad t > 6,$$

we have

$$f'(t) = \frac{1}{t} + \frac{1}{t-1} + \frac{1}{t} - \frac{3}{t} + \log(1 - 6/t) + \frac{6/t(1 - 3/t)}{1 - 6/t} > 0,$$

which implies that $f(t)$ increases with $t$ for $t > 6$. Since

$$g_3(6/p, p) = \frac{6/p(6/p - 1)(6/p - 2)}{6} p^3 (1-p)^{6/p-3}$$

and

$$e^{f(t)} = t(t-1)(t-2)t^{-3}(1 - 6/t)^{t-3},$$

we know that $g_3(6/p, p)$ decreases with $p \in (0, 1]$. Thus,

$$f_3(6/p, p) \leqslant \lim_{p \to 0_+} g_3(6/p, p) = \lim_{t \to \infty} \frac{t(t-1)(t-2)}{6}(6/t)^3(1 - 6/t)^{t-3} = 36e^{-6}.$$

Following the above steps, for

$$g_2(x, p) := \frac{x(x-1)}{2} p^2 (1-p)^{x-2},$$
$$g_1(x, p) := xp(1-p)^{x-1}, \quad \text{and}$$
$$g_0(x, p) := (1-p)^x,$$
$$\text{where } p \in (0, 1], \ x \geqslant 6/p,$$

it is easy to prove that for any $i = 0, 1, 2$, and $p \in (0, 1]$,

$$g_i(x, p) \leqslant g_i(6/p, p) \leqslant \lim_{p \to 0_+} g_i(6/p, p).$$

Specifically, we have

$$g_2(x, p) \leqslant \lim_{t \to \infty} \frac{t(t-1)}{2}(6/t)^2(1 - 6/t)^{t-2} = 18e^{-6},$$
$$g_1(x, p) \leqslant \lim_{t \to \infty} t(6/t)(1 - 6/t)^{t-1} = 6e^{-6}, \quad \text{and}$$
$$g_0(x, p) \leqslant \lim_{t \to \infty}(1 - 6/t)^t = e^{-6}.$$

Then, we can conclude that

$$\mathbb{P}(X \leqslant np/2) \leqslant \sum_{i=0}^{3} \sup_{p \in (0,1], x \geqslant 6/p} g_i(x, p) = (36 + 18 + 6 + 1)e^{-6} < \frac{1}{2}.$$

Next, we consider the regime that $4 \leqslant np < 6$. Now, we have

$$\mathbb{P}(X \leqslant np/2) = \sum_{i=0}^{2} \binom{n}{i} p^i (1-p)^{n-i} \leqslant \sum_{i=0}^{2} \sup_{p \in (0,1], x \geqslant 4/n} g_i(x, p).$$

Following the procedures in the previous case, it is not hard to verify that for $i = 0, 1, 2$,

$$\sup_{p \in (0,1], x \geqslant 4/n} g_i(x, p) = \lim_{p \to 0} g_i(4/p, p) = \lim_{t \to \infty} \frac{4^i}{i!} (1 - 4/t)^{t-i} = \frac{4^i}{i!} e^{-4}.$$

It implies that

$$\mathbb{P}(X \leqslant np/2) = \sum_{i=0}^{2} \binom{n}{i} p^i (1-p)^{n-i} \leqslant \sum_{i=0}^{2} \frac{4^i}{i!} e^{-4} < \frac{1}{2}.$$

Finally, we consider the regime that $2 \leqslant np < 4$. Now, we have

$$\mathbb{P}(X \leqslant np/2) = (1-p)^n + np(1-p)^{n-1} \leqslant \sum_{i=0}^{1} \sup_{p \in (0,1], x \geqslant 2/n} g_i(x, p).$$

Following the previous procedures, it is not hard to verify that for $i = 0, 1$,

$$\sup_{p \in (0,1], x \geqslant 2/n} g_i(x, p) = \lim_{p \to 0} g_i(2/p, p) = \lim_{t \to \infty} \frac{2^i}{i!} (1 - 2/t)^{t-i} = \frac{2^i}{i!} e^{-2}.$$

It implies that

$$\mathbb{P}(X \leqslant np/2) \leqslant \sum_{i=0}^{1} \frac{2^i}{i!} e^{-2} < \frac{1}{2}.$$

In conclusion, if $np \geqslant 2$, then

$$\mathbb{P}(X \leqslant np/2) < \frac{1}{2}.$$

$\square$

## H.2 Bounding the graph dimension

For any $\mathcal{H} \subseteq \mathcal{Y}^{\mathcal{X}}$ of DS dimension $\dim(\mathcal{H}) = d < \infty$, sequence $S = (x_1, \ldots, x_n) \in \mathcal{X}^n$, menu $\mu : \mathcal{X} \to \{Y \subseteq \mathcal{Y} : |Y| \leqslant p\}$ of size $p \in \mathbb{N}$ with $n \in \mathbb{N}$, and $d' \in [n]$, define

$$\mathcal{H}_{S,\mu,d'} := \left\{ h|_S : h \in \mathcal{H}, |\{i \in [n] : h(x_i) \notin \mu(x_i)\}| \leqslant d' \right\},$$
$$\mathcal{H}_{S,\mu,\mathbf{i}} := \left\{ h|_S : h \in \mathcal{H}, \{i \in [n] : h(x_i) \notin \mu(x_i)\} \subseteq \mathbf{i} \right\}, \text{ and}$$
$$\mathcal{H}'_{S,\mu,\mathbf{i}} := \left\{ h : [n] \backslash \mathbf{i} \to \mathcal{Y}, i \mapsto \tilde{h}(i) \middle| \tilde{h} \in \mathcal{H}_{S,\mu,\mathbf{i}} \right\}$$

for all $\mathbf{i} \in 2^{[n]}_{d'}$ which denotes the collection of all subsets of $[n]$ of size $d'$. We have the following lemma.

**Lemma H.9.** $\dim_G(\mathcal{H}_{S,\mu,d'}) \leqslant (2 \log_2(e) + 4)(5d \log_2(p) + 2d')$ for any $d' \in [n]$.

*Proof.* For any $\mathbf{i} \in 2^{[n]}_{d'}$, by Bendavid et al. [1995], Daniely and Shalev-Shwartz [2014], we have

$$\dim_G(\mathcal{H}'_{S,\mu,\mathbf{i}}) \leqslant 5 \log_2(p) \dim_N(\mathcal{H}'_{S,\mu,\mathbf{i}}) \leqslant 5 \log_2(p) \dim(\mathcal{H}) = 5d \log_2(p).$$

For any $\mathbf{j} \subseteq [n]$ that is G-shattered by $\mathcal{H}_{S,\mu,\mathbf{i}}$, define $\mathbf{j}' := \mathbf{j} \backslash \mathbf{i}$. We have that $|\mathbf{j}'| \geqslant |\mathbf{j}| - d'$ and $\mathbf{j}'$ is G-shattered by $\mathcal{H}'_{S,\mu,\mathbf{i}}$, which immediately implies that $|\mathbf{j}'| \leqslant \dim_G(\mathcal{H}'_{S,\mu,\mathbf{i}}) \leqslant 5d \log_2(p)$ and $|\mathbf{j}| \leqslant |\mathbf{j}'| + d' \leqslant 5d \log_2(p) + d'$. It follows that

$$\dim_G(\mathcal{H}_{S,\mu,\mathbf{i}}) \leqslant 5d \log_2(p) + d' =: d_1.$$

For any $m \in \mathbb{N}$ and $\mathbf{j} = \{j_1, \ldots, j_m\} \subseteq [n]$ that is G-shattered by $\mathcal{H}_{S,\mu,d'}$, there exists $f : [m] \to \mathcal{Y}$ such that for any $K \subseteq [m]$, there exists $g \in \mathcal{H}_{S,\mu,d'}$ satisfying $g(j_k) = f(k)$ for all $k \in K$ and $g(j_k) \neq f(k)$ for all $k \in [m]\backslash K$. It follows that $|\mathcal{H}_{S,\mu,d'}(\mathbf{j})| = 2^m$. If $m > d'$, by Lemma H.2, we have

$$2^m = |\mathcal{H}_{S,\mu,d'}(\mathbf{j})| \leqslant \sum_{\mathbf{i} \in 2^{\mathbf{j}}_{d'}} |(\mathcal{H}_{S,\mu,\mathbf{i}})_f(\mathbf{j})| \leqslant \sum_{\mathbf{i} \in 2^{\mathbf{j}}_{d'}} \tau_{\mathcal{H}_{S,\mu,\mathbf{i}}}(m) \leqslant \binom{m}{d'} \left( 2^{d_1} \vee \left( \frac{em}{d_1} \right)^{d_1} \right),$$

which implies that

$$m \leqslant d_1 \left( 1 \vee \log_2 \left( \frac{em}{d_1} \right) \right) + \log_2 \binom{m}{d'} \leqslant d_1 \left( 1 \vee \log_2 \left( \frac{em}{d_1} \right) \right) + d' \log_2 \left( \frac{em}{d'} \right)$$

By Lemma H.11, we have

$$m \leqslant (2 \log_2(e) + 4)(d_1 + d') = (2 \log_2(e) + 4)(5d \log_2(p) + 2d'),$$

which implies that

$$\mathrm{dim}_G(\mathcal{H}_{S,\mu,d'}) \leqslant (2 \log_2(e) + 4)(5d \log_2(p) + 2d').$$

$\square$

**Lemma H.10.** *If $x > 0$ satisfies $x \leqslant a \log_2(x/a) + b$ for some $a, b > 0$, then, we have $x \leqslant 2a + 2b$.*

*Proof.* Define $f(x) := x - a \log_2(x/a) - b$ for $x > 0$. Then, we have $f'(x) = 1 - \frac{a}{x \log(2)}$, which implies that $f$ decreases with $x$ for $x \in (0, a/\log(2))$ and increases with $x$ for $x > a/\log(2)$. Since $2a + 2b > a/\log(2)$, it suffices to prove that $f(2a + 2b) \geqslant 0$. Indeed,

$$f(2a + 2b) = a + b - a \log_2((a + b)/a) = a \left( (a + b)/a - \log_2((a + b)/a) \right) \geqslant 0.$$

$\square$

**Lemma H.11.** *If $x > 0$ satisfies $x \leqslant a \log_2(x/a) + b \log_2(x/b) + c$ for some $a, b, c > 0$, then, we have $x \leqslant 4a + 4b + 2c$.*

*Proof.* Since

$$\begin{aligned}
x &\leqslant a \log_2(x/a) + b \log_2(x/b) + c \\
&= (a + b) \log_2 \frac{x}{a + b} + c - (a + b) \left[ \frac{a}{a + b} \log_2 \frac{a}{a + b} + \frac{b}{a + b} \log_2 \frac{b}{a + b} \right] \\
&\leqslant (a + b) \log_2 \frac{x}{a + b} + c + a + b,
\end{aligned}$$

by Lemma H.10, we have

$$x \leqslant 2(a + b) + 2(a + b + c) = 4a + 4b + 2c.$$

$\square$

