# OpenReview forum: "Improved Sample Complexity for Multiclass PAC Learning"
_NeurIPS.cc/2024/Conference — NeurIPS 2024 poster_

### Official Review · Reviewer_aJcM · 2024-07-11

**Soundness:** 3
**Presentation:** 1
**Contribution:** 3
**Rating:** 5
**Confidence:** 3

**Summary:**

This paper presents improved sample complexity upper and lower bounds for multiclass classification, shaving previous upper bounds to within a factor of $\log(1/\epsilon)$ from the conjectured optimal dependence on $\epsilon$, and adding a dependence of $\log(1/\delta)$ to previous best lower bounds. To do so, the authors study multiclass classification through the lens of the more genearal problem of list-learning -- this generality allows for a theorem stating that the conjectured optimal dependence of $1/\epsilon$ in the sample complexity holds given a conjecture about the sample complexity of list-learning. They further discuss the potential of technique they term ``pivot shifting'' to remove the extra log factor of $1/\epsilon$ in the multi class complexity, showing that whenever pivoting shifting does not increase a characteristic combinatorial complexity measure by more than a constant factor, pivot shifting eliminates the extra log factor.

**Strengths:**

The paper introduces a novel boosting algorithm for list-learning which enables the log improvement in $1/\epsilon$, allowing for the effective application of list-learning machinery to the multiclass problem. It improves on previous best lower bounds via the addition of the usual dependence on the confidence parameter $\log(1/\delta)$.

**Weaknesses:**

Outside of Section 1, I found the presentation of the paper to be very weak. Certain paragraphs present hard to follow proof outlines (e.g. line 236 "the standard leave-one-out argument" is not something I am familiar with and is not explained in the main body). Section 3, which introduces the concept of pivot shifting, provides little intuition into the notation-heavy definitions provided, and cannot be understood without reference to definitions which only appear in the appendix.

It is also a bit unclear what the upside of introducing the generality of list-learning is. The second half of Theorem 2.11 (line 281) does not feel like particularly compelling evidence for the conjectured $1/\epsilon$ dependence, given that it relies on an analogous intuitive conjecture on the sample complexity of the more general problem of list learning.

**Questions:**

1) Is there a particularly compelling reason to believe in the existence of some $\mathcal{A}_{goodlist}$ in Theorem 2.11? What do see as the advantage of thinking about multi-class learning through this extra layer of generality?

2) Could you elaborate on the $\log(1/\delta)$ factor in the lower bound (Theorem 2.5) -- does the proof technique follow that of Charikar and Pabbaraju [2023], or is this a different construction?

**Limitations:**

Yes

---

> ### Author Rebuttal · Authors · 2024-08-06
>
> Thank you for reviewing our paper. Below, $d$ denotes DS dimension and $d_k$ denotes $k$-DS dimension.
>
> ## List learning
>
> * The advantage of reducing to list learning is that it reduces the original problem to an easier problem: list learning is easier than multiclass learning. Indeed, a multiclass learner is a list learner of any size. If a concept class is PAC learnable with some sample complexity, then it is also $k$-list PAC learnable with the same sample complexity for any size $k$.
> On the contrary, for any $k\ge2$, we cannot directly obtain a multiclass learner from a general $k$-list learner.
> If a concept class is $k$-list PAC learnable with some sample complexity, it is unclear whether it is multiclass PAC learnable and what its multiclass sample complexity is. As mentioned in line 285 (there is a typo, it should be "$\mathsf{dim}_k(\mathcal{H})\ge$'' instead of "$\mathsf{dim}_k(\mathcal{H})>$''), $d_k(\mathcal{H})$ is nonincreasing with $k$. Consequently, a $k$-list PAC learnable concept class (i.e., $d_k(\mathcal{H})<\infty$) is not necessarily multiclass PAC learnable.
> Thus, Theorem 2.7 and 2.11 are highly non-trivial because they reduce the harder problem of multiclass learning to the easier problem of list learning: quantitatively, if a concept class is $k$-list learnable with some error rate $r(n)$, it is also multiclass learnable with an error rate bounded by $r(n)+d\log(k)/n$.
>
> * Regarding $\mathcal A_{\text{goodlist}}$, if there is a multiclass learner $A$ with error rate bounded by $f(d)/n$, then $A$ is also a $k$-list learner with $\epsilon_{A,\mathcal{H}}(n)\le f(d)/n$ for any $k\in\mathbb{N}$. Thus, our conjecture on list learners holds if the conjecture of the same rate on multiclass learners holds, indicating that the conjecture for such a list learner is weaker than that for a multiclass learner, hence more approachable.
> Moreover, in Theorem 2.10, our list learner already saves a log factor compared to the previous best list learner in [1]. It is reasonable to conjecture the removal of the remaining log factor.
>
> * The fact that we reduced a log factor in the upper bound of multiclass sample complexity by reducing to list learning also shows its advantage.
> The improvement of log factors in PAC sample complexity is significant; e.g., in binary classification, though the upper bound with a $\log(1/\epsilon)$ factor was established in 1982 [2], the log factor was not removed until 2016 [3].
> Given that we reduced multiclass learning to an easier problem of list learning and thus removed a log factor, we believe our paper has made a significant contribution to multiclass learning.
>
> * Finally, with a large or infinite label space, it is natural to consider reducing the size of the label space by proposing a list learner. This idea has been adopted in [1], the first paper to establish the equivalence of multiclass PAC learnability and the finiteness of DS dimension. [1] proposed a list learner to construct a list sample compression scheme and built their multiclass learner using the labels in the list.
>
>
> ## Lower bound
>
> One cannot derive the $\Omega((d_k-\log\delta)/k\epsilon)$ bound using the $\Omega(d_k/kn)$ expected error lower bound in [4] as a black box since one must stick to the distribution constructed for the lower bound when applying it as a black box.
> Thus, for the PAC lower bound, we have to construct **different** hard distributions from that in [4] for both the $\log(1/\delta)/k\epsilon$ (line 486) and the $d_k/k\epsilon$ (line 511) terms. The detailed constructions are provided in the proof of Theorem 2.5 in Appendix B. To list one difference between our constructions and that in [4], our distributions are not uniform over the $\mathcal{X}$ sequence selected and depends on the parameter $\epsilon$, while theirs are uniform over the $\mathcal{X}$ sequence selected.
>
> Finally, our major contribution is the upper bound. We include the lower bound and its proof for comparison and completeness as it is missing in the literature (see line 37 and 124).
>
> ## Presentation
>
> * We provided the reference [1, Fact 14] of the leave-one-out argument in line 102 where it first appears. We have added the reference in line 236 in the revision. The current proof outlines are concise due to the space limitation, but we believe that they convey the key points of the proof. If the reviewer has other questions regarding the proof outlines, we are glad to answer in the discussion. We will extend the proof outlines to improve the readability in the revision.
>
> * Due to the space limitation, we have to put some definitions in the appendix, but we included the references of the definitions when they appear in the main text. We will move the important definitions in Appendix A to the main text with the additional page if the paper is accepted.
> We believe that we have provided intuition for the definitions. We summarized the existing results in Proposition 3.1 to motivate the idea of upper bounding the density.
> In Theorem 3.2, this idea is supported by its success for classes of DS dimension 1.
> In line 329-335, we explained the reason of considering the pivot of a concept class. In line 346-352, we provided the intuition of pivot shifting and compared it to the standard shifting in the literature. Lemma 3.7 explained the partial validity of pivot shifting, which naturally leads to Open Question 2 about the remaining validity of pivot shifting in upper bounding the density using DS dimension.
> We have added more explanations in words for the technical definitions to improve the readability in the revision.
>
>
> [1] Brukhim et al. A characterization of multiclass learnability. 2022.
>
> [2] Vapnik. Estimation of Dependencies Based on Empirical Data. 1982.
>
> [3] Hanneke. The Optimal Sample Complexity of PAC Learning. 2016.
>
> [4] Charikar and Pabbaraju. A characterization of list learnability. 2023.

---

> > ### Comment · Reviewer_aJcM · 2024-08-10
> >
> > Thank you for your reply, and apologies for not fully understanding your contribution on the first read. On account of this, I raise my rating to a 5.
> >
> > In particular, I thank the authors prompting me to absorb the advatange of the reduction analyzed in Theorem 2.7. This makes clear to me the utility of considering the conjectured $\mathcal{A}_{goodlist}$, and eliminates the concerns in the second half of the above "weaknesses" section of my initial review.
> >
> > I still feel the presentation leaves a fair amount to be desired. While I understand that the contribution is rather technical,
> > the sheer density and abruptness of the writing in certain parts do make for a particularly good reading experience in my opinion. While it's of course possible improve such things in the next version, it does limit my revised rating.
> >
> > Whether the technical contribution outweighs this downside is probably not something I'm qualified to comment on, as the paper is part of a line of work that I'm not familiar with.

---

> ### Author Response · Authors · 2024-08-13
>
> Thank you for raising your rating. We will further elaborate on the technical aspects of the paper and improve its readability.

---

### Official Review · Reviewer_AhgM · 2024-07-12

**Soundness:** 3
**Presentation:** 3
**Contribution:** 3
**Rating:** 6
**Confidence:** 4

**Summary:**

For the problem of analyzing the optimal PAC sample complexity for multi-class learning, two possible routes to induce the improved sample complexity and error rate are proposed. On the one hand, benefiting from the reduction from multi-class learning to list learning and the boosting technique, the dependence of the corresponding upper bounds on the sample size is improved by a logarithmic factor. On the other hand, the introduction of "pivot shifting" shows its potential ability to improve the sample complexity of multi-class learning.

**Strengths:**

1. Based on the Daniely-Shalev-Shwartz (DS) dimension, using the reduction and the boosting technique, the explicit lower bound on the sample complexity of multi-class learning is derived, and the upper bound of the error rate for multi-class learning is improved by a logarithmic factor.
2. In order to obtain the corresponding optimal theoretical bounds, the implementation limitations of some relevant assumptions are formalized as open problems and discussed in detail, i.e., the construction of the specific list learners and the impact of pivot shifting on DS dimension.

**Weaknesses:**

The theoretical analysis in this paper improves the existing theoretical results for multi-class learning on the DS dimension, but the relationship between the obtained theoretical bounds and other data-independent complexity bounds based on combinatorial dimensions, e.g., the graph dimension, the Natarajan dimension and its scale-sensitive analog, is not discussed and compared, which is not conducive to demonstrating the improvement of these theoretical results from a macro perspective. In addition, existing theoretical results on multi-class learning show that data-independent generalization bounds often depend on the number of classes, and the best (weakest) known dependency is logarithmic. The results in this paper focus on improving the dependency on the number of samples, and do not explicitly discuss and analyze the dependency on the number of classes. These related issues require further analysis and explanation.

**Questions:**

Please refer to the Weaknesses for details.

**Limitations:**

This work does not seem to have any potential negative societal impact.

---

> ### Author Rebuttal · Authors · 2024-08-06
>
> Thank you for reviewing our paper. We appreciate your positive feedback.
> First, we need to emphasize that in this paper, we study multiclass PAC sample complexity for general concept classes which can have **infinite** number of labels ($|\mathcal{Y}|=\infty$). Our results hold for general concept classes independent of the number of labels.
> Below, $d$ denotes DS dimension, $d_N$ denotes Natarajan dimension, and $d_G$ denotes graph dimension.
>
> ## Graph dimension and Natarajan dimension
>
> * As we have stated in line 26-28, [1] showed that a concept class is PAC learnable if and only if its DS dimension is finite. Moreover, as is detailed below, it has been shown in the literature that finite graph dimension or Natarajan dimension does **not** characterize multiclass PAC learnability.
>
> * For the graph dimension, [2, Section 6] showed that finite graph dimension does not characterize multiclass PAC learnability by identifying a PAC learnable class with infinite graph dimension and infinite label space.
> [3, Example 1] provided a family of concept classes whose graph dimensions can be any positive integers or infinite, and all those concept classes are PAC learnable with sample complexity $\log(1/\delta)/\epsilon$.
> [1, Example 8] is a concept class with DS dimension 1 (thus it is PAC learnable), infinite graph dimension, and infinite label space.
>
> * For the Natarajan dimension, [1, Theorem 2] provided a concept class with Natarajan dimension 1 and infinite DS dimension (thus it is not PAC learnable).
>
> * Thus, graph dimension and Natarajan dimension cannot appear in the optimal PAC sample complexity.
> In general, for $\mathcal{H}\subseteq \mathcal{Y}^{\mathcal{X}}$, we have $d_N(\mathcal{H})\le d(\mathcal{H})\le d_G(\mathcal{H})$ [4] and for the special case of $|\mathcal{Y}|<\infty$, $d_G(\mathcal{H})\le 5\log_2(|\mathcal{Y}|)d_N(\mathcal{H})$ [3]. Quantitative scaling between those dimensions for general concept classes is still missing, and as is discussed above, there exist $\mathcal{H}$ and $\mathcal{H}'$ such that $d_N(\mathcal{H})=1$, $d(\mathcal{H})=\infty$, $d(\mathcal{H}')=1$, and $d_G(\mathcal{H}')=\infty$.
>
> * For detailed comparisons, our lower bound in eq. (2) is better than the best lower bound in terms of Natarajan dimension in [3, Theorem 5], and the Natarajan dimension cannot upper bound the PAC sample complexity of general concept classes by [1, Theorem 2] discussed before.
> The graph dimension cannot lower bound the PAC sample complexity of general concept classes by the above examples of PAC learnable concept classes with infinite graph dimensions.
> The best upper bound in terms of the graph dimension is $O((d_G(\mathcal{H})+\log(1/\delta))/\epsilon)$ by Proposition I.5 of our paper (see also line 226), which can be infinite for PAC learnable concept classes by the examples discussed before.
> Moreover, by [3, Example 1], there exists $\mathcal H_i$ for all $1\le i\le\infty$ such that $d_G(\mathcal H_i)=i$ and $\mathcal{M}_{\mathcal{H}_i}(\epsilon,\delta)\le \log(1/\delta)/\epsilon$, indicating that their DS dimensions are uniformly bounded in terms of $i$. Thus, the optimal upper bound cannot depend on graph dimension, and the upper bound using graph dimension can be arbitrarily worse than our upper bound using DS dimension as our upper bound is always non-trivial with a finite gap of at most $\sqrt{d}\log(d)\log(d/\epsilon)$ from the optimal bound for PAC learnable concept classes.
>
> ## Number of classes
>
> * As mentioned before, our upper and lower bounds on multiclass PAC sample complexity are independent of the number of classes which can be infinite. We actually prove the following result. There exists an algorithm $A$ and universal constants $C_1,C_2>0$ such that for any feature space $\mathcal{X}$, label space $\mathcal{Y}$ ($|\mathcal{Y}|$ can be arbitrarily large or infinite), and concept class $\mathcal{H}\subseteq\mathcal{Y}^{\mathcal{X}}$, we have $\mathcal M_{\mathcal{H}}(\epsilon,\delta)\ge C_1({d+\log(1/\delta)})/{\epsilon}$ and $\mathcal{M}_{A,\mathcal{H}}(\epsilon,\delta)
> \le C_2({d^{3/2}\log(d)\log(d/\epsilon)+\log(1/\delta)})/\epsilon$.
> As we have listed above, there are many concept classes with infinite label space and finite DS dimension. Actually, by taking finite levels in the tree example in [1, Figure 3], we can construct a series of concept classes with DS dimension 1 and number of labels increasing to infinity. Thus, any bound that increases to infinity with the number of labels cannot be optimal and is worse than our upper bound for general concept classes. It follows that the optimal bound must be uniformly bounded with respect to the number of labels and the number of labels should not exist in the optimal bound.
>
> We will incorporate the key points in the above discussion in the revised version of the paper.
>
>
> [1] Brukhim et al. A characterization of multiclass learnability. FOCS 2022.
>
> [2] Natarajan. Some results on learning. 1989.
>
> [3] Daniely et al. Multiclass learnability and the ERM principle. JMLR, 2015.
>
> [4] Daniely and Shalev-Shwartz. Optimal learners for multiclass problems. COLT 2014.

---

> > ### Comment · Reviewer_AhgM · 2024-08-10
> >
> > Thanks for your detailed feedback. It has partially addressed my concerns.

---

> > > ### Author Response · Authors · 2024-08-13
> > >
> > > Thank you for your response and we are glad to hear that.

---

### Official Review · Reviewer_s6iC · 2024-07-13

**Soundness:** 4
**Presentation:** 3
**Contribution:** 3
**Rating:** 7
**Confidence:** 3

**Summary:**

This work is focused on improving sample complexity upper and lower bounds on multiclass classifcation for a general hypothesis class. Their bounds are given in terms of the so-called DS-dimension of the hypothesis class, which can be viewed as a generalization of the VC dimension to the multiclass setting. Their bounds improve on the best known upper bounds and leave a factor of $O(\sqrt{d})$ between their upper and lower bounds on the sample complexity.

Their main idea is to apply a reduction to list learning, where the learner is allowed to output a set of $k$ potential labels for each $x$ and is evaluated based on whether any of the outputted labels are correct. In particular, they provide a multiclass algorithm that works by calling a list learning algorithm as a subroutine. The subsequent guaranteed loss then grows logarithmically with $k$, the number of used labels.

Their full algorithm consequently requires a list learner, and they provide one that adapts a previously known boosting technique for binary classification to multiclass list learners. To apply boosting, they define the majority vote of a set of lists as the set of labels are included in at least half of the lists. Note that the resulting list has size $2k$ if each of the lists have size $k$.

Combining this with their above reduction results in their upper bound.

**Strengths:**

This paper is a highly technical paper that makes significant progress on a classic and important open problem in learning theory. In addition to improving the current best bounds, it provides an approach for further improvements. In particular, their reduction implies that improvements on list learning will result in improvements on multiclass classification.

**Weaknesses:**

Many of the proof ideas are differed to the appendix. I think it would improve the presentation to include small "proof intution" sections following the theorems. While this would cost more space, I think this can be accounted for by moving the highly technical algorithm blocks into the appendix and merely giving summaries of what the algorithms do in English.

Of course, this paper is "incremental" in that it merely improves existing bounds to an preexisting problem, but I do not see this as cause at all for rejection. The improvements are clearly non-trivial, and the problem being studied is of fundamental importance.

**Questions:**

See weaknesses.

**Limitations:**

Yes.

---

> ### Author Rebuttal · Authors · 2024-08-06
>
> Thank you for reviewing our paper. We appreciate your positive feedback.
> We provided a proof sketch for Theorem 2.7 in line 229-241. We will elaborate more on it and include the proof intuition for other major theorems in the revision. We believe that the extra space can be accounted for by the additional page allowed if the paper is accepted.

---

> > ### Comment · Reviewer_s6iC · 2024-08-12
> > **Response to rebuttal**
> >
> > Thank you for your response, my score will remain the same, and I think this paper should be accepted.

---

> > > ### Author Response · Authors · 2024-08-13
> > >
> > > Thank you again for your support.

---

### Official Review · Reviewer_jEdN · 2024-07-15

**Soundness:** 3
**Presentation:** 3
**Contribution:** 3
**Rating:** 7
**Confidence:** 3

**Summary:**

The paper considers the problem of analyzing the sample complexity of Multiclass PAC Learning. The key contributions of this paper are two-fold: (1) Give an improved upper bound for the sample complexity by a poly-log factor, and (2) give the first (formal) lower bound for the Multiclass PAC Learning problem, which matches that of binary concept class. The key idea is to use a reduction from multiclass learning to list learning, which can then be further analyzed using recent works on boosting algorithms for list learners.

Besides, the paper also explores a (potential) alternative approach to analyze the sample complexity for a concept class in Multiclass PAC Learning by alternatively analyzing its corresponding one-inclusion graph. Following this route, if we can give an upper bound for the density of any concept class (defined by the average degree of its corresponding one-inclusion graph) by multiple of its DS dimensions, we can have a matching lower bound for the problem of Multi-class PAC Learning. They argue that this approach is promising by giving proof for the case where the DS dimension of the concept class is 1.

**Strengths:**

The paper is well-written and easy to follow. Motivation and key results in the main paper are presented clearly and concisely, though I would also want the proofs in the Appendix to be discussed and commented on in the same way as in the main paper. I did not go through the proof very carefully, but it looks good at first glance. I would try to go through that in the details in the rebuttal phase.

I like the presentation of this paper: first presenting the improvement on the sample complexity of Multiclass PAC Learning using a technique, and then proposing an alternative view that can potentially lead to a matching upper bound. This gives good insights to the readers, not simply about the sample complexity improvements, but also about the problem itself.

Overall, I think this is a good paper and vote to accept it.

**Weaknesses:**

There is not much to comment on the weaknesses of this paper. A fastidious reviewer might argue that this paper is a collection of many good but not-so-strong results (in the sense that the list learner techniques used for the proofs of lower and upper-bound using are not completely new), but I think it is good enough for me.

Some minor comments:
1. It is good to include a proof sketch for every result presented in the Appendix, as well as the high-level intuitions behind the results. Though the statements of key results are clear, readers like us would like to see what is actually going on behind the proofs and would appreciate it if the authors could break down the steps for better readability.

**Questions:**

See above.

**Limitations:**

See above.

---

> ### Author Rebuttal · Authors · 2024-08-06
>
> Thank you for reviewing our paper. We appreciate your positive feedback.
> We will include the proof sketches and high-level intuitions of the results presented in the Appendix in the revised version of the paper.

---

### Official Review · Reviewer_wmES · 2024-07-23

**Soundness:** 3
**Presentation:** 2
**Contribution:** 2
**Rating:** 5
**Confidence:** 4

**Summary:**

This paper gives better bounds on the complexity of multiclass learning using DS-dimension and provides a lower bound. The improvement is relatively minor.

**Strengths:**

The main strength of this paper is that it addresses one of the most fundamental problems in learning theory -- the complexity of multiclass learning and provides an improved bound on the sample complexity.

**Weaknesses:**

The writing of the paper and the presentation style can be improved substantially. The technical summary is way too dense and the comparison with the previous work is very narrow.

The technical improvement is rather weak (some log factors) and doesn't address the main gap (polynomial gap in d).

**Questions:**

-- I am not convinced that DS-dimension is the right quantity to look at here. Don't we already have optimal bounds for the sample complexity of multiclass learning in terms or Graph Dimension (see Daniely and Shalev-Shwartz)? Can you please explain how your results compare to the Graph Dimension results?

---

> ### Author Rebuttal · Authors · 2024-08-06
>
> Thank you for reviewing our paper. Below, $d$ denotes DS dimension and $d_G$ denotes graph dimension.
>
> ## Graph dimension
>
> * The DS dimension is the right quantity to look at here.
> The optimal multiclass PAC sample complexity is described by DS dimension and **not** by graph dimension.
> As is stated in line 26-28, [1] showed that a concept class is PAC learnable if and only if its DS dimension is finite.
> [2, Sec. 6] showed that finite graph dimension does not characterize PAC learnability by proposing a PAC learnable class with **infinite** graph dimension.
> [3, Example 1] provided a family of concept classes $\mathcal H_i$ for all $1\le i\le\infty$ such that $d_G(\mathcal H_i)=i$ and $\mathcal M_{\mathcal{H}_i}(\epsilon,\delta)\le \log(1/\delta)/\epsilon$, indicating that their DS dimensions are uniformly bounded.
> [1, Example 8] is a concept class with DS dimension 1 (thus it is PAC learnable) and infinite graph dimension.
> Thus, graph dimension cannot appear in the optimal multiclass PAC sample complexity.
>
> * By the above examples, graph dimension cannot lower bound the PAC sample complexity of general concept classes.
> The best upper bound using graph dimension is $O((d_G(\mathcal{H})+\log(1/\delta))/\epsilon)$ by our Proposition I.5 (see also line 226), which can be infinite for PAC learnable classes.
> By [3, Example 1], the optimal upper bound cannot depend on graph dimension, and the above upper bound can be arbitrarily worse than our upper bound using DS dimension as our upper bound is always non-trivial with a finite gap of at most $\sqrt{d}\log(d)\log(d/\epsilon)$ from the optimal bound for PAC learnable classes.
>
> * Moreover, Daniely and Shalev-Shwartz [4] did not provide optimal bound on PAC sample complexity in terms of graph dimension. Instead, in [4, Conjecture 11], they conjectured that the optimal bound is given by DS dimension. In the last paragraph, they wrote
> > it is known (Daniely et al., 2011) that the graph dimension does not characterize the sample complexity, since it can be substantially larger than the sample complexity in several cases.
>
> We will include the key points in the above discussion in our revision.
>
>
> ## Technical summary
>
> We believe that it is necessary to include enough technical summaries to introduce results and ideas. We made efforts on providing sufficient motivation of the technical summaries. For example, in Sec. 1, we reviewed existing results in multiclass PAC learning in the first two paragraphs, motivating the introduction of the concepts in PAC learning in Sec. 1.1. In paragraph 3 of Sec. 1, we previewed our reduction to list learning, motivating the technical summary of list learning. In paragraph 4 of Sec. 1, we provided the motivation of the pivot shifting in Sec. 3. In Sec. 2.2, after the main result, we provided a technical overview in line 229-241 to explain the key points of the proof. In Sec. 2.3, we provided the intuition of Alg. 2 in the first paragraph. In Sec. 3, we built the technical summary on existing results to explain the motivation of upper bounding the density and pivot shifting.
> We will elaborate more on technical summary in the revision.
>
> ## Comparison with previous work
>
> We believe that our paper makes adequate comparisons with previous work. We compared our upper bound to the previous best bound (line 39 and 120) in line 49 and 288. We mentioned that a potentially sharp lower bound on PAC sample complexity is still missing in the literature in line 37 and 124. In line 244-251, we compared to the list sample compression scheme in [1] to explain how we saved a $\log(n)$ factor using sampled boosting.
> In line 268-271, we compared to the bound of directly invoking the learner in [1] to Alg. 2 to explain how we avoided a $\log(d)$ factor using list learning.
> In Prop. 3.1, we reviewed known results on density to explain the reason of bounding the density.
> In line 329-334, we referred to the induction technique in Haussler et al. (1994) to motivate pivot shifting.
> In line 348-352, we compared pivot shifting to the standard shifting technique.
>
>
> ## Technical improvement
>
> * We do not think that our improvement is weak. The optimal PAC sample complexity is a fundamental and difficult problem in statistical learning theory. The characterization of multiclass PAC learnability remained open for decades until recently solved by [1].
> The improvement of log factors in sample complexity is highly non-trivial: e.g., in binary classification, though the upper bound with a $\log(1/\epsilon)$ factor was established in 1982 [5], the log factor was not removed until 2016 [6].
>
> * In this paper, we focus on improving the dependence of PAC sample complexity on $\epsilon$ (equivalently, the dependence of the error rate on $n$). In learning theory, it is important to study the dependence of error rate on $n$ with $d$ fixed because a learner is typically applied to a fixed concept class with increasing $n$ so that $n$ is much larger than $d$. Moreover, by [7], our upper bound improves the universal learning rate from $\log^2(n)/n$ to $(\log n)/n$ for the concept classes studied in [7].
> Ideally, we want tight dependence on each parameter. However, as tight dependence on either $d$ or $\epsilon$ is unknown, we believe it is meaningful to explore better dependence on either parameter.
>
> * Besides the removal of a log factor, we also proposed two possible routes toward removing the remaining log factor of $\epsilon$.
>
>
> [1] Brukhim et al. A characterization of multiclass learnability. 2022.
>
> [2] Natarajan. Some results on learning. 1989.
>
> [3] Daniely et al. Multiclass learnability and the ERM principle. 2015.
>
> [4] Daniely and Shalev-Shwartz. Optimal learners for multiclass problems. 2014.
>
> [5] Vapnik. Estimation of Dependencies Based on Empirical Data. 1982.
>
> [6] Hanneke. The Optimal Sample Complexity of PAC Learning. 2016.
>
> [7] Hanneke et al. Universal rates for multiclass learning. 2023.

---

> > ### Comment · Reviewer_wmES · 2024-08-12
> >
> > Thanks, sounds good. I can increase the score to 5.

---

> > > ### Author Response · Authors · 2024-08-13
> > >
> > > Thank you for increasing the rating.

---

### Decision · Program_Chairs · 2024-09-25

**Decision:**

Accept (poster)

**Comment:**

The authors make progress in answering a fundamental result in learning theory: the sample complexity of multi-class learning. The authors improve the existing gap by poly-log factors, and outlining directions to potentially get a tight bound. The reviewers appreciated the results. The authors are encouraged to incorporate the comments specially those about the presentation of the technical results.